# Direct activation of a bacterial innate immune system by a viral capsid protein

Tong Zhang[1], Hedvig Tamman[2], Kyo Coppieters 't Wallant[3], Tatsuaki Kurata[4], Michele LeRoux[1], Sriram Srikant[1], Tetiana Brodiazhenko[5], Albinas Cepauskas[2], Ariel Talavera[2], Chloe Martens[3], Gemma C. Atkinson[4], Vasili Hauryliuk[4,5 ✉], Abel Garcia-Pino[2,6 ✉] & Michael T. Laub[1,7 ✉]

Bacteria have evolved diverse immunity mechanisms to protect themselves against the constant onslaught of bacteriophages[1–3]. Similar to how eukaryotic innate immune systems sense foreign invaders through pathogen-associated molecular patterns[4] (PAMPs), many bacterial immune systems that respond to bacteriophage infection require phage-specific triggers to be activated. However, the identities of such triggers and the sensing mechanisms remain largely unknown. Here we identify and investigate the anti-phage function of CapRel[SJ46], a fused toxin–antitoxin system that protects *Escherichia coli* against diverse phages. Using genetic, biochemical and structural analyses, we demonstrate that the C-terminal domain of CapRel[SJ46] regulates the toxic N-terminal region, serving as both antitoxin and phage infection sensor. Following infection by certain phages, newly synthesized major capsid protein binds directly to the C-terminal domain of CapRel[SJ46] to relieve autoinhibition, enabling the toxin domain to pyrophosphorylate tRNAs, which blocks translation to restrict viral infection. Collectively, our results reveal the molecular mechanism by which a bacterial immune system directly senses a conserved, essential component of phages, suggesting a PAMP-like sensing model for toxin–antitoxin-mediated innate immunity in bacteria. We provide evidence that CapRels and their phage-encoded triggers are engaged in a 'Red Queen conflict'[5], revealing a new front in the intense coevolutionary battle between phages and bacteria. Given that capsid proteins of some eukaryotic viruses are known to stimulate innate immune signalling in mammalian hosts[6–10], our results reveal a deeply conserved facet of immunity.

Innate immunity in eukaryotes relies on pattern recognition receptors that directly sense PAMPs, which are conserved molecules like bacterial lipopolysaccharide and flagellin, or viral RNA or DNA[4]. These innate immune signalling pathways must remain silent before infection, but be poised for rapid activation to defend against foreign invaders. Bacteria also encode innate immune systems to protect themselves against diverse invading bacteriophages, but how they sense infection is poorly understood. One exception is restriction-modification systems, but these distinguish self from non-self using DNA methylation patterns and thus do not need the nuclease effector to be specifically activated during phage infection. Similarly, for CRISPR–Cas systems, the adaptive immune system of some bacteria, guide RNAs enable a cell to specifically target foreign DNA without a need for infection-triggered activation. Dozens of bacterial defence systems have been identified in recent years[11–15], but unlike restriction modification and CRISPR–Cas, many of them must be specifically activated upon phage infection. This is particularly critical for abortive infection systems, in which a defence system uses a lethal effector to kill an infected cell and

prevent propagation of the virus through a population[16]. The phage-encoded triggers for such bacterial immunity mechanisms are largely unknown.

Toxin–antitoxin systems are prevalent genetic elements in bacteria that are emerging as key components of anti-phage innate immunity[13,14,17,18], often serving as abortive infection modules that kill infected cells to prevent spread of phages through a population. How toxin–antitoxin systems sense and respond to phage infection remains poorly understood. For the *toxIN* system, toxin (ToxN) activation relies on efficient, phage-induced shutoff of host transcription coupled to the intrinsically fast turnover of the antitoxin *toxI*[19–21]. However, *toxI* is an RNA, whereas most toxin–antitoxin systems feature a protein antitoxin. For systems with a protein antitoxin, the mechanism of activation is often assumed to arise through antitoxin degradation. Although protein antitoxins are often more proteolytically unstable than their cognate toxins, their turnover may not be fast enough to enable toxin activation on the timescale of a phage infection[22], suggesting the existence of alternative mechanisms for toxin–antitoxin system activation. Bacterial retrons

[1]Department of Biology, Massachusetts Institute of Technology, Cambridge, MA, USA. [2]Cellular and Molecular Microbiology, Faculté des Sciences, Université Libre de Bruxelles, (ULB), Brussels, Belgium. [3]Centre for Structural Biology and Bioinformatics, Université Libre de Bruxelles (ULB), Bruxelles, Belgium. [4]Department of Experimental Medical Science, Lund University, Lund, Sweden. [5]Institute of Technology, University of Tartu, Tartu, Estonia. [6]WELBIO, Brussels, Belgium. [7]Howard Hughes Medical Institute, Massachusetts Institute of Technology, Cambridge, MA, USA. ✉e-mail: vasili.hauryliuk@med.lu.se; abel.garcia-pino@ulb.be; laub@mit.edu

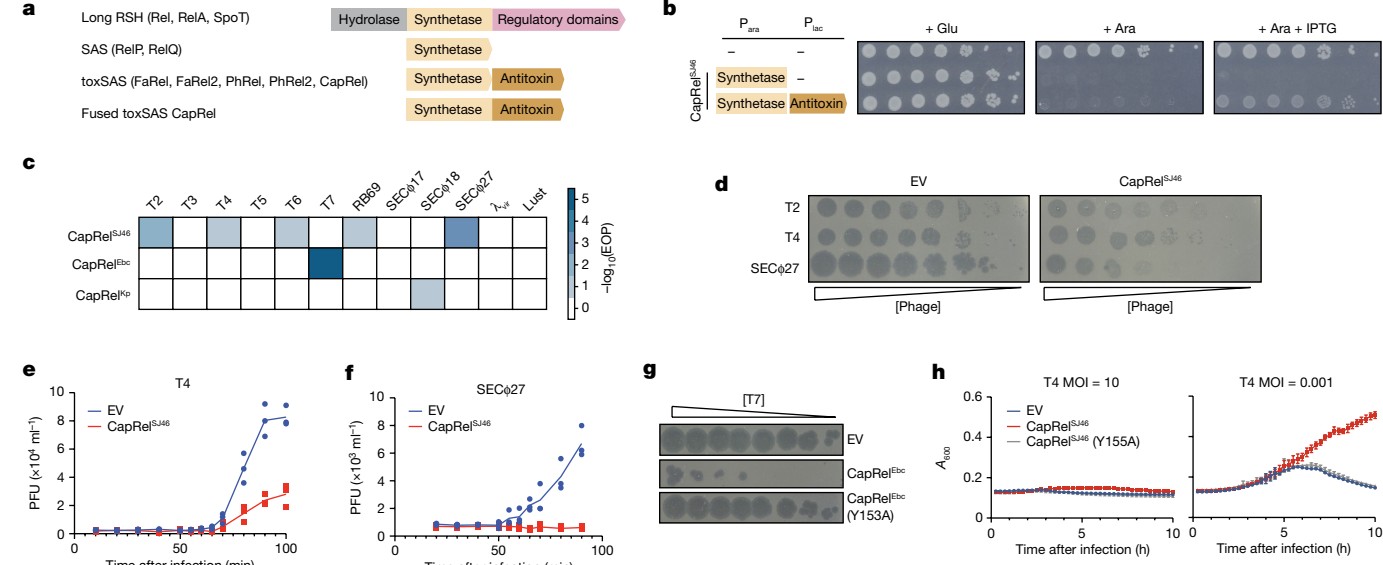

**Fig. 1 | Fused CapRel homologues are toxin–antitoxin systems that can provide *E. coli* with robust defence against phages. a**, Domain organization of long RelA-SpoT homologues (RSH), SAS, toxSAS and the fused subclass of toxSAS toxin–antitoxin systems including CapRel[SJ46]. **b**, Cell viability assessed by serial dilutions for strains expressing the N-terminal toxin domain of CapRel[SJ46] alone or with the C-terminal antitoxin domain. Ara, arabinose; glu, glucose; $P_{ara}$, arabinose-inducible, glucose-repressible promoter; $P_{lac}$, IPTG-inducible promoter. **c**, EOP data for the indicated phages when infecting cells producing CapRel[SJ46], CapRel[Ebc] or CapRel[Kp]. **d**, Serial, tenfold dilutions of the indicated phages spotted on lawns of cells harbouring plasmid expressing CapRel[SJ46] or an empty vector (EV). Relative phage concentration is indicated by the height of the wedge. **e**,**f**, One-step growth curves measuring plaque-forming units (PFU) during the first round of infection by T4 (**e**) or SECΦ27 (**f**) in cells harbouring plasmid expressing CapRel[SJ46] or an empty vector. **g**, Serial dilutions of T7 phage spotted on lawns of cells harbouring plasmids expressing CapRel[Ebc], CapRel[Ebc](Y153A) or an empty vector. **h**, Growth of cells producing CapRel[SJ46] or CapRel[SJ46](Y155A) or harbouring an empty vector, following infection with T4 at a MOI of 10 or 0.001. Data are mean ± s.d. of eight plate replicates and representative of three independent experiments.

function as tripartite toxin–antitoxin systems and can be activated by overexpressing various prophage genes[23], but whether these activators function as such during phage infection is unknown.

## CapRel[SJ46] is an anti-phage fused toxin–antitoxin system

To investigate the molecular basis of phage-induced activation of bacterial immunity, we focused on toxSAS toxin–antitoxin systems, which feature toxins homologous to bacterial small alarmone synthetases (SAS) that pyrophosphorylate purine nucleotides[24]. Whereas most housekeeping alarmone synthetases produce the growth regulator (p) ppGpp[25,26], toxSAS toxins can synthesize (p)ppApp to deplete ATP[24,27] or pyrophosphorylate tRNAs to inhibit translation[24,28]. Their cognate antitoxins can either bind and neutralize the toxin or act as hydrolases to reverse toxin-catalysed pyrophosphorylation[24,28]. One subfamily of translation-inhibiting toxSAS is called CapRel, on the basis of their prevalence in Cyanobacteria, Actinobacteria, and Proteobacteria and sequence similarity to the (p)ppGpp synthetase/hydrolase Rel. The CapRel subfamily is the most broadly distributed subfamily of toxSAS, being found across multiple phyla of Gram-positive and Gram-negative bacteria. In addition to its prevalence in Cyanobacteria, Actinobacteria, and Proteobacteria, CapRel representatives are found in Spirochetes, Bacteroidetes, and Firmicutes[24]. This subfamily often features systems that are—in contrast to canonical bicistronic toxin–antitoxin systems—encoded by a single open reading frame, with an N-terminal domain homologous to toxSAS toxins and a C-terminal domain homologous to the corresponding antitoxins[29] (Fig. 1a and Extended Data Figs. 1 and 2a). This fused architecture is the predominant form of CapRel, except in Actinobacteria.

Prophages in bacterial genomes often encode anti-phage systems, helping both the temperate phage and its host to defend against other phages[11,30,31]. Here, we selected a fused CapRel encoded by the *Salmonella* temperate phage SJ46 and also encoded (with 100% amino acid

sequence identity) in prophages of several *E. coli* strains (Extended Data Fig. 2b). The toxin and antitoxin-like regions of CapRel[SJ46] are related to the PhRel toxSAS toxin and its antitoxin ATphRel, respectively, from the mycobacterial temperate phage Phrann[30] (Extended Data Fig. 1 and 2a). This Phrann-encoded system can inhibit superinfection by other temperate mycophages[30], although the molecular basis of PhRel activation is not known. To test whether CapRel[SJ46] is a fused toxin–antitoxin system, we cloned the N-terminal region containing the conserved alarmone synthetase domain and the C-terminal region containing the putative antitoxin domain under the control of separate inducible promoters. Expression of the N-terminal fragment alone was toxic, and its toxicity was rescued in *trans* by co-expression with the C-terminal fragment (Fig. 1b and Extended Data Fig. 2c), suggesting that CapRel[SJ46] is a fused toxin–antitoxin system.

To determine whether fused CapRels can defend against phages, we transformed *E. coli* MG1655 with three different systems expressed from their native promoters on low copy number plasmids, and then tested whether each conferred protection against a panel of 12 diverse coliphages. In addition to CapRel[SJ46], we also tested CapRel[Ebc] from *Enterobacter chengduensis* and CapRel[Kp] from *Klebsiella pneumoniae* (Fig. 1c and Extended Data Fig. 2b,d). CapRel[SJ46] expressed from a low copy number plasmid or the *E. coli* genome decreased the efficiency of plaquing (EOP) for T2, T4, T6, RB69 and SECΦ27 by 10- to 1,000-fold (Fig. 1c,d and Extended Data Fig. 2e,f), indicating that this system provides strong protection against phages. T4 phage formed smaller plaques when plated onto CapRel[SJ46]-containing cells, and one-step growth curves confirmed that CapRel[SJ46] reduced the burst size of T4 by around 70% (Fig. 1e) and prevented bursting of SECΦ27 (Fig. 1f). CapRel[Ebc] protected strongly against T7 and CapRel[Kp] protected, albeit less efficiently, against SECΦ18 (Fig. 1g and Extended Data Fig. 2d,g).

Next, we tested whether CapRel[SJ46] provides direct immunity or functions through abortive infection, in which an infected cell dies but

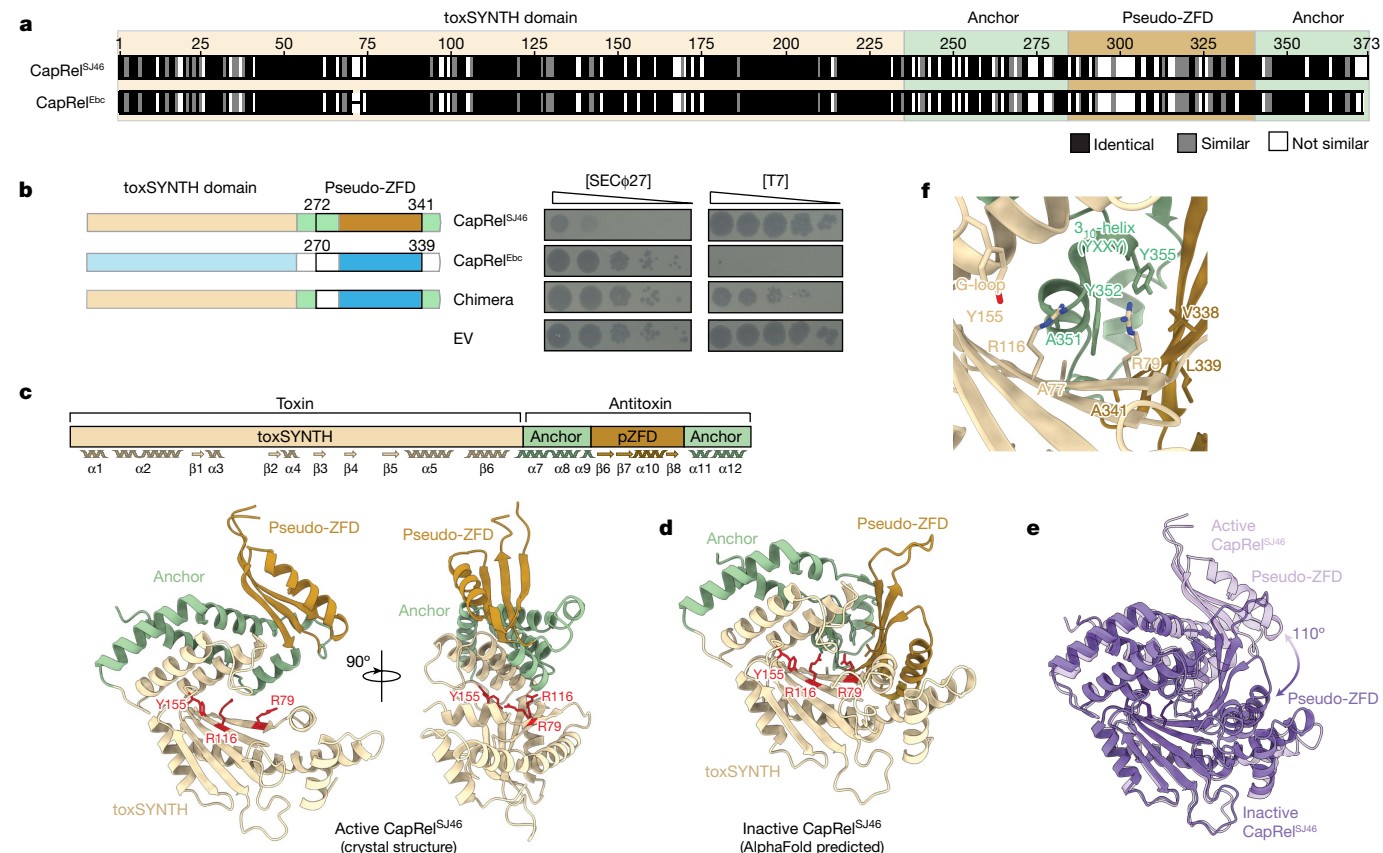

**Fig. 2 | The pseudo-ZFD of CapRel confers phage specificity. a**, Sequence alignment of CapRel[SJ46] and CapRel[Ebc], with the more variable pseudo-ZFD labelled. **b**, Serial dilutions of the indicated phages spotted on lawns of cells harbouring the indicated CapRel constructs. Left, schematic of the CapRel constructs. **c**, Cartoon representation of the crystal structure of CapRel[SJ46] with active site G-loop Y155 and the ATP-coordination residues R79 and R116 highlighted in red. Structural elements (toxSYNTH, pseudo-ZFD and the anchors) are coloured as in **a**. Wavy lines and arrows indicate alpha helices and beta strands, respectively. **d**, The closed conformation of CapRel[SJ46] predicted by AlphaFold and coloured as in **c**. **e**, Superposition of the active (open, light purple) and inactive (closed, dark purple) states of CapRel[SJ46] as observed in the crystal structure and predicted by AlphaFold. **f**, Details of the autoinhibited active site of CapRel[SJ46] in the closed state. In this conformation, the YXXY neutralization motif blocks the adenine coordination site, preventing catalysis.

prevents the production of mature virions, thereby sparing uninfected cells in a population. To this end, we infected cells containing CapRel[SJ46] with T4 at a multiplicity of infection (MOI) of either 10 or 0.001, and found that defence only manifested at the low MOI, indicating that CapRel[SJ46] probably functions through abortive infection (Fig. 1h). Phage protection by CapRel[SJ46] depended on the predicted enzymatic activity of the N-terminal synthetase domain, as substituting the conserved tyrosine (Y155A) in the G-loop that is critical for substrate binding abolished phage protection[32] (Fig. 1h). A similar catalysis-compromising substitution, Y153A in CapRel[Ebc], also abolished phage protection (Fig. 1g). Collectively, our results established that fused CapRels can provide anti-phage defence, with variable phage specificity.

To understand what determines the specificity of phage protection by fused CapRels, we compared CapRel[SJ46] and CapRel[Ebc]. These two proteins share 70% amino acid identity overall, but differ substantially in their C-terminal regions, which are only 47% identical (Fig. 2a). In addition, this region is the least conserved when we compared a more diverse set of fused CapRel homologues (Extended Data Fig. 3a). Because CapRel[SJ46] and CapRel[Ebc] protected against different phages, we made a chimera in which the C-terminal region of CapRel[SJ46] was replaced by the corresponding region of CapRel[Ebc]. This chimeric CapRel no longer protected against SECΦ27 and gained protection against T7, manifesting as a tenfold decrease in EOP and smaller plaques (Fig. 2b and Extended Data Fig. 3b). The chimera protected against T7 less efficiently than CapRel[Ebc], despite similar expression levels (Extended Data Fig. 3c), probably owing to a non-optimal interaction interface between the N and C termini. Nevertheless, this result indicates that the C-terminal region of CapRel is critical for phage specificity.

## Structural analysis of CapRel[SJ46]

To further understand the mechanistic basis of anti-phage defence by CapRel[SJ46], we solved a crystal structure to 2.3 Å resolution (Fig. 2c and Extended Data Table 1). CapRel[SJ46] contains a conserved, N-terminal nucleotide pyrophosphokinase domain (toxSYNTH) that is also present in alarmone synthetases and tRNA-pyrophosphotransferase enzymes, which mediates toxicity. The smaller C-terminal antitoxin domain consists of a central antiparallel three-stranded β-sheet with an α-helix connecting β-strands β7 and β8 (Fig. 2c and Extended Data Fig. 3d,e). The antitoxin domain is topologically analogous to the classical zinc-finger domain (ZFD), but lacks the conserved cysteines (Extended Data Fig. 3f); we refer to this domain as a pseudo-ZFD. The pseudo-ZFD is connected to the toxSYNTH domain via α-helices α7, α8 and α9, and has a C-terminal α-helical extension that anchors the domain to α8 and α9 (Extended Data Fig. 3d). In this structure, the toxSYNTH and pseudo-ZFD are distant from each other, revealing the ATP donor nucleotide binding pocket and the conserved G-loop Y155 of toxSYNTH (Fig. 2c), indicating that this state may represent the active, toxic conformation of CapRel[SJ46]. This open state captured by crystallography is probably stabilized by crystal lattice contacts that are incompatible with a closing of the active site owing to clashes with symmetry-related partners (Extended Data Fig. 3g).

To explore the conformational dynamics of the enzyme, we used AlphaFold[33] to predict possible alternative structures of CapRel[SJ46]. In addition to predicting the open conformation observed in the crystal structure (Extended Data Fig. 3h), AlphaFold also predicted a closed conformational state in which the C-terminal domain folds back 110° onto the toxSYNTH central β-sheet and blocks the ATP-binding site (Fig. 2d). Comparison of the two states suggested that a conserved YXXY motif (Extended Data Fig. 3a) located in the hinge connecting the two C-terminal α-helices in the open state morphs into a short $3_{10}$-helix in the closed state (Fig. 2e,f). This $3_{10}$-helix projects into the toxSYNTH active site and intercalates between β1 R79 and β2 R116 to block the adenine coordination site (Fig. 2e,f).

We hypothesized that this closed-to-open switch underlies the activation of CapRel[SJ46], with the docking of the pseudo-ZFD onto toxSYNTH precluding substrate binding in the absence of phage infection (Fig. 2f). To test this hypothesis, we made single substitutions to the YXXY motif (Y352A and Y355A) and residues from the predicted interface that serves as a scaffold to orient and stabilize the $3_{10}$-helix (A77K, R116A, V338A, L339A, A341K and A351K), which are highly conserved among diverse CapRel homologues (Extended Data Fig. 3a,i). Whereas wild-type CapRel[SJ46] was not toxic when expressed in cells, each of the substitutions predicted to disrupt the intramolecular recognition interface, on either the N- or the C-terminal domain, rendered CapRel[SJ46] toxic (Extended Data Fig. 3j,k). These substitutions probably lead to constitutive activation of CapRel[SJ46] by disrupting an autoinhibited state. As a control, we showed that substitutions in different structural elements of the pseudo-ZFD but not pointing towards the interface did not lead to constitutive activation (Extended Data Fig. 3j,k). Collectively, our results indicate that the pseudo-ZFD docks onto the ATP-binding site of CapRel[SJ46] to prevent switching to the open state captured in our crystal structure. Conservation of the YXXY motif and the interface residues suggests that this auto-inhibitory regulation is probably retained in other CapRels.

## SECΦ27 capsid protein triggers CapRel[SJ46]

Because full-length, wild-type CapRel[SJ46] was not toxic when expressed in the absence of phage infection, we inferred that it must somehow be activated by phages. The toxins of some toxin–antitoxin systems are activated by the degradation of the more labile antitoxin[19,34,35]. To test whether the C-terminal antitoxin of CapRel[SJ46] is proteolytically cleaved off and degraded upon phage infection, we N-terminally tagged CapRel[SJ46] and first verified that the tagged protein still defends against phage (Extended Data Fig. 4a). We then tracked the size of CapRel[SJ46] by immunoblotting following infection with SECΦ27. The overall protein levels of CapRel[SJ46] remained constant and we observed only the full-length product, suggesting that CapRel[SJ46] was not proteolytically processed (Fig. 3a and Extended Data Fig. 4b,c). Thus, we hypothesized that a specific phage product regulates the C-terminal domain of CapRel[SJ46] to relieve autoinhibition. To identify such a factor, we sought to identify SECΦ27 mutants that escape CapRel[SJ46] defence. As no spontaneous escape mutants could be isolated, we used an experimental evolution approach (Fig. 3b). In brief, we infected cells containing an empty vector or CapRel[SJ46] with serial dilutions of phage in microtitre plates. After overnight incubation, we collected and pooled the phages from cleared wells, which indicated successful infection, and used these to seed the next round of infections. Initially, cells with the empty vector were infected more efficiently, but after 13 rounds, each phage population had evolved to infect cells containing empty vector or CapRel[SJ46] to a similar degree (Fig. 3c and Extended Data Fig. 4d). We isolated ten mutant SECΦ27 clones from five independently evolved populations and sequenced their genomes. Remarkably, all ten clones contained a point mutation in the same gene that encodes a hypothetical protein, Gp57, with nine clones producing the same L114P substitution and one clone yielding an I115F substitution (Fig. 3d and Supplementary Table 1).

The structure of the hypothetical protein Gp57 predicted by Alpha-Fold[33] is highly similar (DALI z-score of approximately 17) to the HK97 fold commonly adopted by major capsid proteins of dsDNA viruses including bacteriophages and herpesviruses[36] (Fig. 3e). By performing mass spectrometry on wild-type and escape-mutant SECΦ27 phages, we identified this hypothetical protein as the most abundant protein in mature virions, consistent with it being the major capsid protein of SECΦ27 (Fig. 3f and Extended Data Table 2).

Our results suggested that wild-type Gp57 from SECΦ27 activates CapRel[SJ46], with the escape mutants preventing activation while retaining the ability to form a capsid. To test this hypothesis, we first examined whether Gp57 alone is sufficient to activate CapRel[SJ46]. Indeed, co-producing wild-type Gp57 with wild-type CapRel[SJ46] was highly toxic to cells in the absence of phage infection, whereas neither evolved variant (L114P or I115F) of Gp57 had a measurable effect on growth when co-produced with CapRel[SJ46] (Fig. 3g and Extended Data Fig. 4e,f). As controls, we confirmed that expressing the wild-type or either Gp57 variant was not toxic on its own or if co-produced with a catalytically compromised CapRel[SJ46] (Extended Data Fig. 4g).

To examine the basis of CapRel[SJ46] toxicity, we first co-produced it with wild-type or the L114P variant of Gp57 and then measured the effects on bulk transcription or translation by pulse-labelling with $^3$H-uridine or with $^{35}$S-methionine and $^{35}$S-cysteine, respectively. Active CapRel[SJ46] produced with wild-type Gp57 robustly inhibited translation but not transcription (Fig. 3h and Extended Data Fig. 4h), whereas no effect was seen with Gp57(L114P). Similar effects were seen when over-expressing just the N-terminal domain of CapRel[SJ46] (Extended Data Fig. 4i). We also measured bulk translation and transcription following SECΦ27 infection of CapRel[SJ46]-containing cells and observed a decrease in translation but not transcription with wild-type SECΦ27 (Fig. 3i and Extended Data Fig. 4j), with a less pronounced decrease in translation compared to overproducing Gp57 likely because poor adsorption of SECΦ27 phage leads to substantial asynchrony of infection dynamics (Extended Data Fig. 4k). No effect on translation was seen with the evolved mutant phage producing Gp57(L114P) (Fig. 3i). To compare the timing of CapRel[SJ46] activation during infection to the timing of Gp57 production, we expressed a haemagglutinin (HA)-tagged version of Gp57 from its native promoter in the bacterial genome and infected cells with SECΦ27. Immunoblotting indicated substantial accumulation of Gp57 by around 30 min after infection (Extended Data Fig. 4l), which coincides with the detected inhibition of translation (Fig. 3i).

Next, we measured the ability of full-length CapRel[SJ46] to affect translation in vitro using the reconstituted in vitro transcription–translation system. Purified CapRel[SJ46] inhibited synthesis of a control DHFR protein after pre-synthesizing SECΦ27 major capsid protein Gp57 (using a template encoding it), whereas no inhibition was seen for either the L114P or I115F variants of Gp57 (Fig. 3j). We also incubated wild-type or an evolved variant (L114P or I115F) of Gp57 with [γ-$^{32}$P]ATP and bulk E. coli tRNAs in the presence and absence of purified CapRel[SJ46]. Wild-type Gp57 strongly stimulated the pyrophosphorylation of tRNAs by CapRel[SJ46], similar to the previously characterized toxSAS enzymes FaRel2 and PhRel2[28] (Fig. 3k). With the L114P or I115F variant of Gp57, tRNA pyrophosphorylation was reduced to the background levels seen with CapRel[SJ46] alone. Together, our results demonstrate that Gp57, the major capsid protein of SECΦ27, is both necessary and sufficient to activate CapRel[SJ46], enabling it to pyrophosphorylate tRNAs and inhibit translation.

## CapRel[SJ46] binds SECΦ27 capsid protein

To test whether the SECΦ27 major capsid protein directly binds CapRel[SJ46], we first immunoprecipitated CapRel[SJ46]–Flag from cells infected with wild-type phage or the mutant that produces Gp57(L114P) after verifying the tag does not affect CapRel[SJ46] function (Extended Data Fig. 4a). We detected Gp57 that had co-precipitated with CapRel[SJ46] by mass

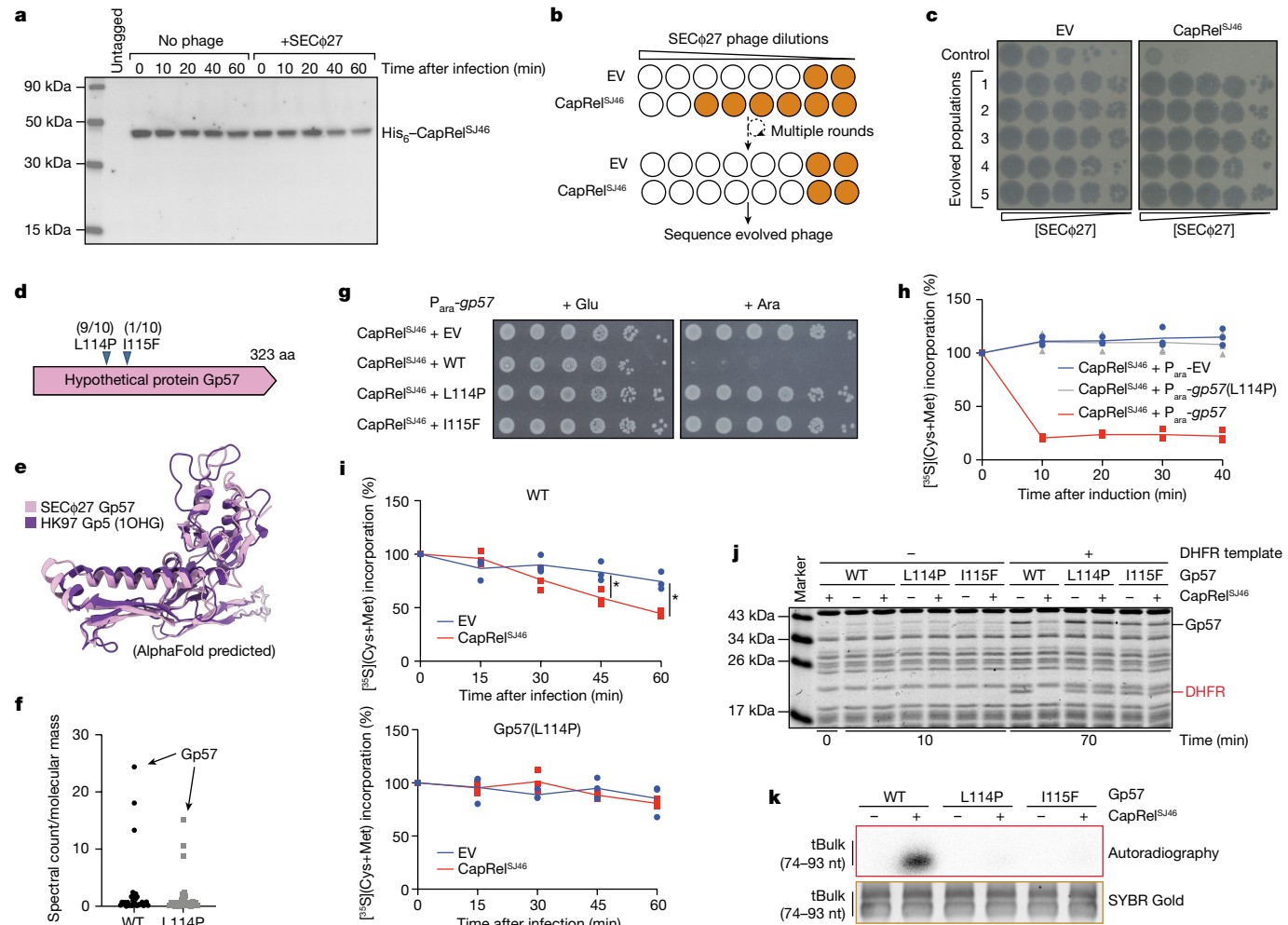

**Fig. 3 | CapRel$^{SJ46}$ is activated by the major capsid protein of SECΦ27 to pyrophosphorylate tRNAs and block translation. a**, Immunoblot of His$_6$–CapRel$^{SJ46}$ following infection with SECΦ27 compared with an uninfected control. Representative of two biological replicates. **b**, Schematic of the experimental evolution approach to identify SECΦ27 escape mutants that can infect CapRel$^{SJ46}$-containing cells. White wells indicate clearing by phages and brown wells indicate bacterial growth. **c**, Serial dilutions of five independently evolved populations of SECΦ27 and a control population spotted on cells harbouring an empty vector or a CapRel$^{SJ46}$ expression vector. **d**, Summary of identified escape mutants, all of which map to a hypothetical protein encoded by gene 57 of SECΦ27. The fraction of each escape mutant is indicated. **e**, AlphaFold-predicted structure of Gp57 compared with the major capsid protein Gp5 from phage HK97. **f**, Mass spectrometry analysis of SECΦ27 phage lysates (wild type or a mutant producing Gp57(L114P)). Spectrum count normalized to molecular mass is shown for individual phage proteins. **g**, Serial

dilutions of cells expressing CapRel$^{SJ46}$ and the indicated variant of Gp57 from an arabinose-inducible promoter on media containing glucose or arabinose. **h**, Cells expressing CapRel$^{SJ46}$ and Gp57 (wild type (WT) or L114P) from an arabinose-inducible promoter or harbouring empty vector were pulse-labelled with $^{35}$S-cysteine and $^{35}$S-methionine after addition of arabinose. **i**, As in **h**, but for cells harbouring a CapRel$^{SJ46}$ expression vector or empty vector and after infection with SECΦ27 (top) or the escape mutant expressing Gp57(L114P) (bottom). Asterisks indicate $P = 0.022$ (45 min) or $P = 0.004$ (60 min) (unpaired two-tailed $t$-test). **j**, In vitro transcription–translation assays using DHFR production from a DNA template as the readout of expression activity. Purified CapRel$^{SJ46}$ was added along with a template for producing Gp57. Representative of two biological replicates. **k**, Autoradiography of reactions in which purified CapRel$^{SJ46}$ was incubated with [γ-$^{32}$P]ATP, bulk *E. coli* tRNAs and Gp57. SYBR Gold staining of bulk tRNAs serves as a loading control. Representative of two biological replicates.

spectrometry when cells were infected with wild-type phage, with a significant reduction in the mutant phage (Extended Data Fig. 5a,b). In addition, we co-produced CapRel$^{SJ46}$–Flag and Gp57–HA and found that wild-type, but not the L114P or I115F variant of the capsid protein, co-precipitated with CapRel$^{SJ46}$–Flag (Fig. 4a and Extended Data Fig. 5c). Finally, we purified both full-length CapRel$^{SJ46}$ and Gp57, and used isothermal titration calorimetry (ITC) to show that they interact directly with an affinity of 350 nM in a 1:1 ratio (Fig. 4b). We confirmed that CapRel$^{SJ46}$ and Gp57 form a complex with 1:1 stoichiometry using size-exclusion chromatography coupled to multi-angle light scattering (SEC–MALS) (Extended Data Fig. 5d). Introducing the escape substitutions L114P and I115F into Gp57 decreased affinity more than 60-fold (Extended Data Fig. 5e).

Consistent with this tight binding interaction, the ab initio AlphaFold prediction of the CapRel$^{SJ46}$–Gp57 complex has a large contact interface of around 1,800 Å$^2$ (Fig. 4c). Although further structural studies are needed to fully validate this complex structure, the AlphaFold prediction has CapRel$^{SJ46}$ adopting the same open state seen in our crystal structure (Fig. 2c), with the pseudo-ZFD making extensive contacts with the β-sheet and spine α-helix of the peripheral (P)-domain of Gp57 (Fig. 4c and Extended Data Fig. 5f). Notably, this region of Gp57 contains the residues L114 and I115 identified in our escape mutants. The complex predicted further interactions of pseudo-ZFD β6–β7 loop with the β6–α5 and β8–β9 loops of the axial (A)-domain of Gp57. In this arrangement Gp57 prevents the recoil of pseudo-ZFD to block

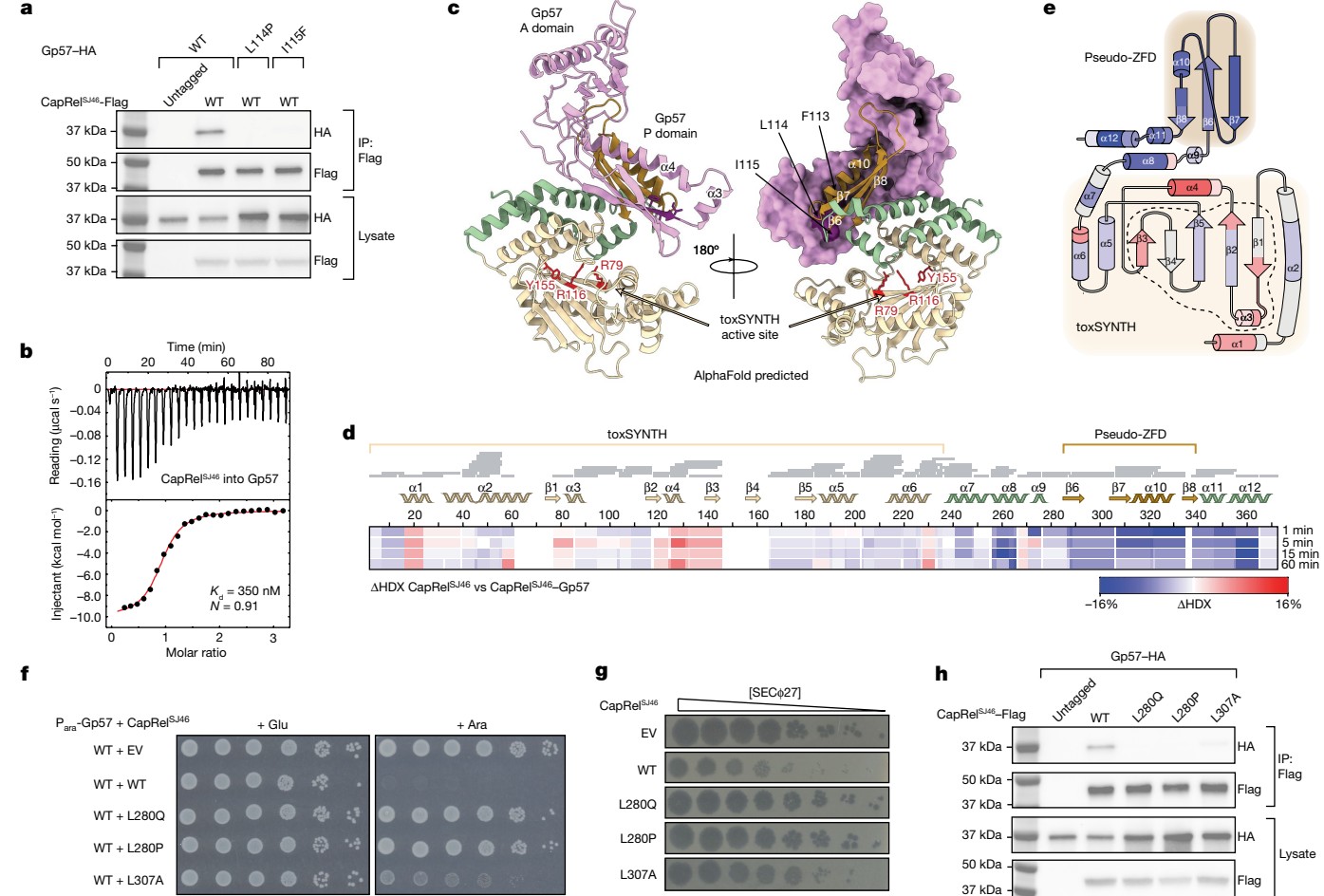

**Fig. 4 | The SECΦ27 major capsid protein Gp57 binds directly to the pseudo-ZFD of CapRel^SJ46. a**, CapRel^SJ46–Flag was immunoprecipitated from cells expressing CapRel^SJ46–Flag and Gp57–HA (wild-type or indicated mutant) and probed for the presence of the indicated Gp57 variant via the HA tag. Lysates used as input for the immunoprecipitation (IP) were probed as controls for expression levels. Images are representatives of three biological replicates. **b**, Binding of CapRel^SJ46 to Gp57 monitored by ITC. $K_d$, binding affinity; $N$, stoichiometry. **c**, Structural model of the CapRel^SJ46–Gp57 complex predicted by AlphaFold. According to the model, the P-domain of Gp57 (pink) recognizes the pseudo-ZFD (orange) and anchor regions (green) of CapRel^SJ46. This interaction prevents the recoil of pseudo-ZFD to the active site and activates the enzyme. **d**, Differential HDX (ΔHDX) between CapRel^SJ46 and CapRel^SJ46–Gp57 displayed as a difference heat map. Red indicates increased

deuteration of CapRel^SJ46 in the presence of Gp57; blue indicates lower deuteration. Grey bars indicate peptides identified in mass spectrometry analysis. **e**, Topological representation of CapRel^SJ46, coloured according to the ΔHDX. The active site of the enzyme is marked by a black dashed outline and the catalytic toxSYNTH domain and the phage-recognition pseudo-ZFD are shadowed in light yellow and light orange, respectively. **f**, Serial dilutions on media containing glucose or arabinose of cells expressing the indicated mutant of CapRel^SJ46 from its native promoter and the wild-type Gp57 from an arabinose-inducible promoter. **g**, Serial dilutions of SECΦ27 phage spotted on cells expressing the indicated mutant of CapRel^SJ46 or an empty vector. **h**, As in **a**, but with the indicated mutants of CapRel^SJ46–Flag. Images shown are representatives of three independent biological replicates.

the active site of the enzyme while stabilizing the YXXY motif in the non-neutralizing hinge conformation.

Results from hydrogen–deuterium exchange (HDX) monitored by mass spectrometry strongly supported the AlphaFold predictions. In the presence of Gp57, the pseudo-ZFD of CapRel^SJ46 became more protected with the strongest protection mapping to α10, β8, and the C-terminal α-helical extension (Fig. 4d,e and Extended Data Fig. 5g,h). This overlaps the same region critical for phage specificity (Fig. 2b). The HDX data also confirmed the interface formed between Gp57 P-domain β5 and CapRel^SJ46 pseudo-ZFD as well as the Gp57 A-domain β8–β9 loop and CapRel^SJ46 β6–β7 loop. Finally, we observed increased deuterium uptake in CapRel^SJ46 in residues 110–124 of β2 and 125–130 of α4, which are part of the adenine coordination pocket of toxSYNTH, thus confirming that interaction with Gp57 exposes the active site of the enzyme (Fig. 4d,e and Extended Data Fig. 5g,h).

To further validate the role of the pseudo-ZFD in binding and activating CapRel^SJ46, we performed error-prone PCR-based mutagenesis on

this domain and screened for mutations that disrupted activation of CapRel^SJ46 when it was co-produced with the capsid protein Gp57. The substitutions L280Q and L280P drastically reduced the toxicity of CapRel^SJ46 in the presence of wild-type Gp57 (Fig. 4f and Extended Data Fig. 5i), and prevented CapRel^SJ46 from protecting against SECΦ27 infection (Fig. 4g and Extended Data Fig. 5j). Importantly, these CapRel^SJ46 variants still protected *E. coli* against phage T2 and T4, indicating that these variants retained structural integrity (Extended Data Fig. 5k). The substitution L307A had similar, but less pronounced, effects on CapRel^SJ46 activity (Fig. 4f,g).

The crystal structure of CapRel^SJ46 suggested that L280 and L307 in the wild-type protein promote the open, active state, with L280 stabilizing one of the hinge regions involving the pseudo-ZFD and L307 structuring the β6–β7 loop that interacts with Gp57 A-domain. The L280Q and L280P variants of CapRel^SJ46 were unable to co-precipitate the major capsid protein of SECΦ27, and the L307A substitution significantly reduced binding in this assay (Fig. 4h). Using ITC, we also

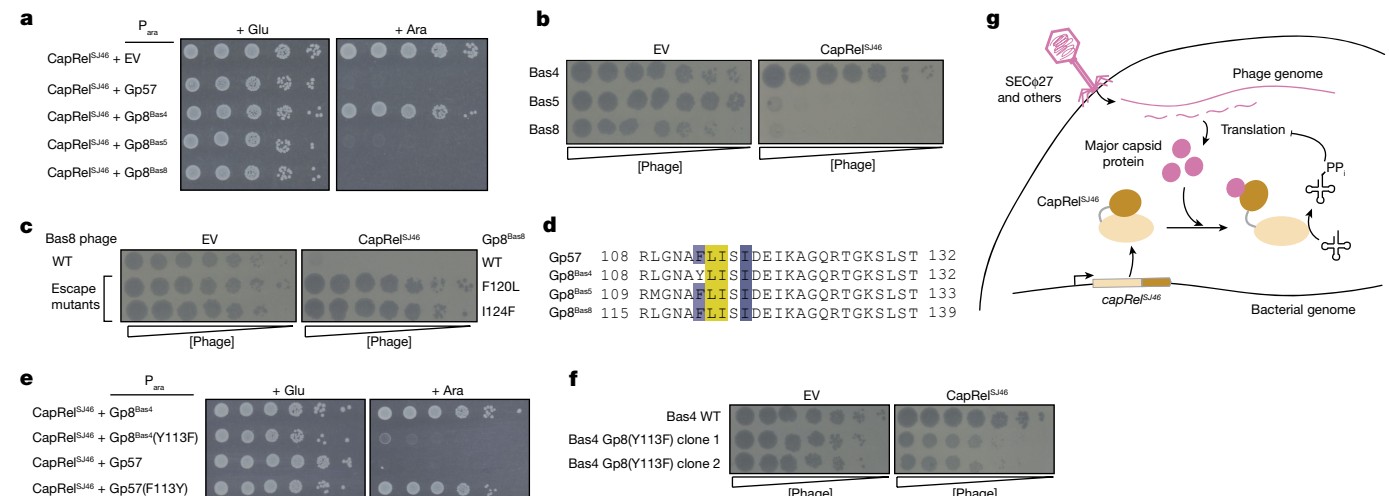

**Fig. 5 | Evidence for the coevolution of CapRel^SJ46 and the major capsid protein of SECΦ27 and related phages. a**, Serial dilutions on media containing glucose or arabinose of cells expressing CapRel^SJ46 from its native promoter and the major capsid protein homologue from the phage indicated from an arabinose-inducible promoter. **b**, Serial dilutions of the phages indicated spotted on lawns of cells expressing CapRel^SJ46 or an empty vector. **c**, Serial dilutions of wild-type Bas8 phage or the escape mutants bearing the major capsid mutations indicated spotted on lawns of cells harbouring CapRel^SJ46 or an empty vector. **d**, Alignment of the region of the major capsid protein in SECΦ27, Bas5 and Bas8 that triggers CapRel^SJ46, along with Bas4, which has a tyrosine at position 113 instead of phenylalanine. **e**, Serial dilutions showed on media containing glucose or arabinose of cells expressing CapRel^SJ46 from its native promoter and the Bas4 or SECΦ27 major capsid protein variant indicated from an arabinose-inducible promoter. **f**, Serial dilutions of wild-type Bas4 or two mutant clones containing Y113F in the major capsid protein Gp8 spotted on lawns of cells expressing CapRel^SJ46 or an empty vector. **g**, Model for the direct activation of CapRel^SJ46 by the major capsid protein of SECΦ27 and related phages. After genome injection, the production of the major capsid protein triggers relief of autoinhibition by the C-terminal antitoxin of CapRel^SJ46, leading to pyrophosphorylation (PP$_i$) of tRNAs by activated CapRel^SJ46, which inhibits translation and restricts viral infection.

showed that the L280P variant of CapRel^SJ46 could no longer bind to wild-type Gp57 (Extended Data Fig. 5e). In sum, our findings strongly support a model in which the C-terminal pseudo-ZFD of CapRel^SJ46 directly recognizes the major capsid protein of SECΦ27, thereby triggering a relief of autoinhibition of the N-terminal toxSYNTH domain.

## Other phage capsids activate CapRel^SJ46

The pseudo-ZFD of CapRel^SJ46, including residues L280 and L307, is the least well conserved region of the protein (Extended Data Fig. 3a,i). This variability may reflect a 'Red Queen dynamic', a hallmark of many host–pathogen interfaces that arises from cycles of selective pressure on pathogens to evade host immunity followed by selection on host immune factors to restore recognition of a pathogen[5]. As triggers of the CapRel defence system, phage capsid proteins are likely to be under pressure to diversify while retaining the ability to form a capsid, leading to a selective pressure on the pseudo-ZFD of CapRel to diversify and retain its interaction with the capsid proteins. To test this hypothesis, we examined three phages from the BASEL collection[37] (Bas4, Bas5 and Bas8) that are closely related to SECΦ27 and contain Gp8, a close homologue of Gp57 (Extended Data Fig. 6a). We first found that co-expressing the major capsid homologues from Bas5 and Bas8—but not that of Bas4—with CapRel^SJ46 rendered CapRel^SJ46 toxic, as with the SECΦ27 capsid protein (Fig. 5a and Extended Data Fig. 6b). We then tested whether CapRel^SJ46 protects against these phages and found that it protected strongly against Bas5 and Bas8, but not Bas4 (Fig. 5b and Extended Data Fig. 6c).

To validate that defence against Bas8 requires activation of CapRel^SJ46 by the capsid protein homologue of this phage, we isolated spontaneous mutants of Bas8 that escaped defence. Two mutant clones of Bas8 were no longer defended against by CapRel^SJ46 and contained either an F120L or I124F substitution in the major capsid homologue Gp8^Bas8 (Fig. 5c and Extended Data Fig. 6d). Each substitution significantly reduced the ability of the capsid protein to activate CapRel^SJ46 when co-produced (Extended Data Fig. 6e,f). As these Gp8^Bas8 variants were not toxic when expressed on their own (Extended Data Fig. 6g), the residual toxicity seen during co-production with CapRel^SJ46 probably arises from their accumulation over extended periods of time, despite their reduced binding affinity for CapRel^SJ46. By contrast, each escape mutant almost completely evaded CapRel^SJ46 defence (Fig. 5c), given the short timescale of phage infection. Notably, the two positions identified in Bas8 escape mutants were close to the positions of the escape mutants identified in SECΦ27 Gp57, further confirming that this region in the major capsid protein is important for activating CapRel^SJ46 (Fig. 5d). In addition, we identified another CapRel homologue from *E. coli* strain HT2012018, called CapRel^EcHT, that defends against Bas8 (Extended Data Fig. 6h). Three different substitutions were identified in the same region of the major capsid protein (G111S, L116F and I124N) that enabled Bas8 to escape defence (Extended Data Fig. 6h), suggesting that CapRel^EcHT—like CapRel^SJ46—is activated by the major capsid protein.

Unlike Bas8, Bas4 was not defended against by CapRel^SJ46, and its capsid homologue did not activate CapRel^SJ46 despite being 98% identical to SECΦ27 Gp57, with just 5 amino acid differences between the two. However, one difference is at position 113, near the region that is likely to bind to CapRel^SJ46. This residue is a phenylalanine in the capsid proteins of SECΦ27, Bas5 and Bas8, but a tyrosine in the Bas4 capsid homologue (Fig. 5d). We tested whether this residue is critical for activation by making a Y113F substitution in the Bas4 capsid homologue and found that it gained the ability to activate CapRel^SJ46 when co-produced (Fig. 5e and Extended Data Fig. 6i). By contrast, a F113Y substitution in the SECΦ27 capsid protein abolished its ability to activate CapRel^SJ46. Additionally, we mutated Bas4 phage such that it produces major capsid protein with the Y113F substitution. This mutant phage could still produce mature virions, but was defended against by CapRel^SJ46 (Fig. 5f and Extended Data Fig. 6j). These results support the notion of a Red Queen dynamic between the pseudo-ZFD of CapRel and the phage capsid proteins that directly bind and activate CapRel.

## Conclusions

We propose the following model for CapRel[SJ46] activation by SECΦ27 (Fig. 5g). Without phage infection, CapRel[SJ46] adopts an inactive, closed conformation in cells with its C-terminal antitoxin domain autoinhibiting the N-terminal toxin domain. Upon infection, the phage's major capsid protein is produced and directly binds to CapRel[SJ46] to stabilize the active, open state. This open state enables CapRel[SJ46] to pyrophosphorylate tRNAs and inhibit translation, which is likely to prevent mature virion production, leading to an abortive infection that prevents propagation of phage through a population of cells. Notably, our results imply that type II toxin–antitoxin systems, which feature protein antitoxins, can be activated without proteolysis of the antitoxin, which is often asserted as their primary means of activation. Although our work focused on a fused type II toxin–antitoxin system, activation by the direct binding of phage proteins could also be a common mechanism for canonical, non-fused type II toxin–antitoxin systems. As noted above, although antitoxins are frequently less stable than their cognate toxins, their turnover may not occur quickly enough to respond to phage infection[22], whereas direct binding to phage-encoded triggers could allow rapid activation.

Major capsid proteins, such as Gp57 from SECΦ27, may be a common trigger for toxin–antitoxin systems and other anti-phage defence systems. Prior studies found that a short peptide called Gol within the major capsid protein Gp23 of T4 can activate the Lit protease in *E. coli*, which cleaves EF-Tu, if both Gol and Lit protease are overproduced[38,39]. For PifA, which allows the F plasmid to exclude T7, escape mutants mapped to the major capsid protein, but this interaction has not been studied biochemically[40]. Recent work reported that mutations in the major capsid protein of T5 allow it to overcome Pycsar-mediated defence, but the capsid protein alone is insufficient to activate Pycsar[41]. Finally, mutations in the gene encoding a major capsid protein of a *Pseudomonas aeruginosa* phage enables escape from CBASS-mediated defence, but whether the capsid protein is an activator of CBASS is not yet known[42]. Nevertheless, we anticipate that major capsid proteins may be common direct triggers for a diverse range of anti-phage defence systems. Other structural components of phages may also serve as triggers; for example, antiviral STAND defence systems were recently found to be directly triggered by terminase or portal proteins[43]. Although structural proteins may be effective triggers, the intense coevolution of phages and bacteria may also drive defence systems to rely on yet other triggers[44].

As with PAMPs in eukaryotes, relying on essential, abundant components of phages for activation may help ensure that an immune response is mounted only following an infection. Notably, the capsid proteins of some eukaryotic viruses stimulate mammalian innate immune pathways. For instance, HIV capsid protein is directly detected in the host cell cytoplasm and nucleus by TRIM5 and NONO, respectively, to trigger innate immune activation[7,9]. Thus, our results suggest that similar principles of pathogen detection underlie the function and molecular basis of innate immunity in both bacteria and eukaryotes.

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

# Methods

## Strains and growth conditions

All bacterial and phage strains used in this study are listed in Supplementary Table 2. *Escherichia coli* strains were routinely grown at 37 °C in Luria broth (LB) medium for cloning and maintenance. Phages were propagated by infecting a culture of *E. coli* MG1655 at an $A_{600}$ of 0.1–0.2 with a MOI of 0.1. Cleared cultures were pelleted by centrifugation to remove residual bacteria and filtered through a 0.2-µm filter. Chloroform was then added to phage lysates to prevent bacterial growth. All phage infection experiments in liquid media and phage spotting experiments were performed in LB medium at 25 °C, except for spotting of T2 and T4 on strains producing CapRel[SJ46] open reading frames were (6.4 g l$^{-1}$ Na$_2$HPO$_4$-7H$_2$O, 1.5 g l$^{-1}$ KH$_2$PO$_4$, 0.25 g l$^{-1}$ NaCl, 0.5 g l$^{-1}$ NH$_4$Cl medium supplemented with 0.1% casamino acids, 0.4% glycerol, 0.4% glucose, 2 mM MgSO$_4$, and 0.1 mM CaCl$_2$) at 30 °C. For liquid induction experiments from pBAD33 vectors, bacterial cells were grown in M9 medium. Antibiotics were used at the following concentrations (liquid; plates): carbenicillin (50 µg ml$^{-1}$; 100 µg ml$^{-1}$) and chloramphenicol (20 µg ml$^{-1}$; 30 µg ml$^{-1}$).

## Plasmid construction

All plasmids are listed in Supplementary Table 3. All primers and synthesized gene sequences are listed in Supplementary Table 4.

pBR322-capRel constructs: DNA encoding CapRel[SJ46], CapRel[Ebc], CapRel[Kp] or CapRel[EcHT] open reading frames were expressed in *E. coli* and 100–200 bp of the upstream region from the source organism was added in each case for native expression (TZ-1 to TZ-5, TZ-92 and TZ-93). DNA was commercially synthesized by Integrated DNA Technology as gBlocks and assembled into a promoter-less backbone of pBR322 amplified with TZ-6 and TZ-7 by Gibson assembly. Mutations that produce the single amino acid substitutions CapRel[SJ46](Y155A), CapRel[Ebc](Y153A), CapRel[SJ46](L280Q), CapRel[SJ46](L280P) and CapRel[SJ46](L307A) were generated by site-directed mutagenesis using primers TZ-8 to TZ-11 and TZ-49 to TZ-54. To add an N-terminal His$_6$-tag or a C-terminal Flag tag to CapRel[SJ46], primers TZ-41 and TZ-42 or TZ-45 and TZ-46 were used to PCR-amplify pBR322-capRel[SJ46] followed by Gibson assembly. pBR322-capRel-chimera was constructed by inserting *capRel[Ebc] (270–339)* that had been PCR-amplified with TZ-22 and TZ-23 into pBR322-capRel[SJ46] linearized with TZ-20 and TZ-21 using Gibson assembly. To add a C-terminal Flag tag to CapRel[Ebc] or the CapRel chimera, primers TZ-90 and TZ-91, or TZ-45 and TZ-46 were used to PCR-amplify corresponding pBR322 vectors followed by Gibson assembly.

pBAD33-capRel[SJ46] constructs: *capRel[SJ46](1–272)* or full-length *capRel[SJ46]* was PCR-amplified with TZ-14 and TZ-15, or TZ-14 and TZ-24, respectively, and inserted into pBAD33 linearized with TZ-12 and TZ-13 using Gibson assembly. pBAD33-capRel[SJ46] variants (A77K, R116A, V338A, L339A, A341K, A351K, Y352A or Y355A) were constructed by site-directed mutagenesis using primers TZ-25 to TZ-40. pBAD33-capRel[SJ46] variants (R78A, K311A, R314A, E319A and K346A) were constructed by site-directed mutagenesis using primers TZ-80 to TZ-89.

pEXT20-capRel[SJ46] construct: *capRel[SJ46](273–373)* was PCR-amplified with primers TZ-18 and TZ-19, and then inserted into linearized pEXT20 with TZ-16 and TZ-17 using Gibson assembly.

pBAD33-gp57 constructs: wild-type or mutant variant (L114P or I115F) *gp57* was PCR-amplified from the corresponding wild-type or escape-mutant SECΦ27 phage using primers TZ-43 and TZ-44, and inserted into linearized pBAD33 using Gibson assembly. A C-terminal HA tag was added to wild-type or mutant *gp57* using primers TZ-47 and TZ-48 to PCR-amplify the corresponding construct followed by Gibson assembly. The F113Y variant of *gp57* was generated by site-directed mutagenesis using primers TZ-63 and TZ-64.

pBAD33-gp8: the genes encoding the major capsid protein homologues Gp8[Bas4], Gp8[Bas5] and Gp8[Bas8] were PCR-amplified from the corresponding phage using primers TZ-55 to TZ-60 and inserted into linearized pBAD33 by Gibson assembly. The Y113F variant of *gp8[Bas4]* was generated by site-directed mutagenesis using primers TZ-61 and TZ-62. The F120L and I124F variants of *gp8[Bas8]* were cloned from the corresponding phage escape mutants using primers TZ-59 and TZ-60.

pET-gp57 constructs: a *gp57* fragment was PCR-amplified with primers TZ-65 and TZ-66 and the TZ-67 template using Gibson assembly, the resulting linear DNA fragment was inserted into linearized pET24d (without tag) using TZ-68 and TZ-69. Template TZ-67 was synthesized as a gBlock by Integrated DNA Technology. To construct L114P and I115F-substituted *gp57* variants, DNA fragments were PCR-amplified with primer pairs TZ-70 + TZ-71 and TZ-72 + TZ-73 (for L114P) or with pairs TZ-74 + TZ-71 and TZ75 + TZ-73 (for I115F) and pET-gp57 template. The resulting linear DNA fragments were assembled using Gibson assembly.

pET24d-His$_{10}$-SUMO-capRel[SJ46] constructs: the *capRel[SJ46]* open reading frame was PCR-amplified using primers TZ-76 and TZ-77 as well as pBAD-capRel[SJ46] as template, and, using Gibson assembly, inserted into a linearized pET24d-His$_{10}$-SUMO plasmid using primers TZ-78 and TZ-79.

## Strain construction

Plasmids described above were introduced into *E. coli* MG1655 or BW27783 by TSS transformation or electroporation.

A single copy of *capRel[SJ46]*, *capRel[SJ46](Y155A)*, *His$_6$-capRel[SJ46]*, *gp57* or *gp57-HA* with its native promoter was inserted onto the MG1655 chromosome at the HK022 attachment site using the CRIM system[45], using the pAH69 helper plasmid with the pAH144 vector containing the desired insert.

Bas4 mutant phage were generated using a CRISPR–Cas system for targeted mutagenesis as described[46]. In brief, sequences for RNA guides to target Cas9-mediated cleavage were designed using the toolbox in Geneious Prime 2021.2.2 and selected for targeting of *gp8[Bas4]* but nowhere else in the Bas4 genome. The guides were inserted into the pCas9 plasmid and tested for their ability to restrict Bas4. An efficient guide was selected and the pCas9 guide plasmid was co-transformed into *E. coli* MG1655 with a high copy number repair plasmid containing *gp8[Bas4](Y113F)* with the guide mutated to prevent self-cutting. The wild-type Bas4 phage was plated onto a strain containing both the pCas9 guide and the repair plasmid, and single plaques were screened by Sanger Sequencing. Two clones that produce the Y113F substituted Gp8 were propagated twice on strains containing only pCas9 guide for further selection and genomes were sequence verified by Illumina sequencing as described below.

## External data

Previously published structures are available from the Protein Data Bank (PDB) with IDs 5DEC, 6S2T and 2LVH. The UniRef90 database is publicly available. Reference phage genomes are publicly available: SECΦ27 (NC_047938.1), Bas04 (MZ501069.1), Bas08 (MZ501059.1).

## Toxicity assays on solid media

For producing the CapRel[SJ46] N- and C-terminal domains, single colonies of *E. coli* MG1655 containing pBAD33-capRel[SJ46](1–272) and pEXT20-capRel[SJ46](273–373) or the corresponding empty vectors were grown for 6 h at 37 °C in LB-glucose to saturation. 200 µl of each saturated culture was then pelleted by centrifugation at 4,000$g$ for 10 min, washed once in 1× phosphate-buffered saline (PBS), and resuspended in 400 µl 1× PBS. Cultures were then serially diluted 10-fold in 1× PBS and spotted on M9L plates (M9 medium supplemented with 5% LB (v/v)) further supplemented with 0.4% glucose, 0.2% arabinose or 0.2% arabinose and 100 µM IPTG. Plates were then incubated at 37 °C overnight before imaging.

For producing full-length CapRel[SJ46], *E. coli* MG1655 containing pBAD33-capRel[SJ46] or a mutant form of capRel[SJ46] were grown to

saturation and processed as above. Cultures were plated onto 0.4% glucose and 0.2% arabinose and incubated at 37 °C overnight.

For co-producing CapRel$^{SJ46}$ and the major capsid proteins from SECΦ27, Bas4, Bas5, or Bas8, *E. coli* MG1655 harbouring pBR322-capRel$^{SJ46}$ or MG1655 containing genomic *capRel$^{SJ46}$* and pBAD33-capsid protein were grown to saturation and processed as above. Cultures were plated onto 0.4% glucose and 0.2% arabinose and incubated at 37 °C overnight.

For co-producing CapRel$^{SJ46}$ and variants of the major capsid protein from Bas8, *E. coli* BW27783 harbouring pBR322-capRel$^{SJ46}$ and pBAD33-gp8$^{Bas8}$ (wild-type or a mutant variant) were grown to saturation and processed as above. Cultures were plated onto 0.4% glucose and 0.0002% arabinose and incubated at 37 °C overnight.

## Phage spotting assays and EOP measurements

Phage stocks isolated from single plaques were propagated in *E. coli* MG1655 at 37 °C in LB. To titre phage, dilutions of stocks were mixed with *E. coli* MG1655 and melted LB + 0.5% agar and spread on LB + 1.2% agar plates and incubated at 37 °C overnight. For phage spotting assays, 40 μl of a bacterial strain of interest was mixed with 4 ml LB + 0.5% agar and spread on an LB + 1.2% agar + antibiotic plate. Phage stocks were then serially diluted in 1× FM buffer (20 mM Tris-HCl pH 7.4, 100 mM NaCl, 10 mM MgSO$_4$), and 2 μl of each dilution was spotted on the bacterial lawn. Plates were then incubated at 25 °C overnight before imaging. EOP was calculated by comparing the ability of the phage to form plaques on an experimental strain relative to the control strain. Experiments were replicated 3 times independently and representative images are shown.

For spotting phage T2 and T4 on strains producing CapRel$^{SJ46}$ variants, 40 μl of a bacterial strain of interest was mixed with 4 ml M9 + 0.5% agar and spread on an M9 + 1.2% agar + antibiotic plate. Phage were serially diluted and spotted as described above. Plates were then incubated at 30 °C overnight before imaging.

## Growth curves following phage infection in liquid culture

Single colonies of *E. coli* MG1655 pBR322 empty vector or pBR322-capRel$^{SJ46}$ or pBR322-capRel$^{SJ46}$(Y155A) were grown in LB overnight. Cultures were then back-diluted to $A_{600}$ = 0.1 in fresh LB and 100 μl of cells were added into each well of a 96-well plate. Ten microlitres of serial diluted T4 phage were added to each well at the indicated MOI and growth following phage infection was measured at 15 min intervals with orbital shaking at 25 °C on a plate reader (Biotek). Data reported are the mean and standard deviation of eight plate replicates and the growth curve experiment was replicated three times independently.

## One-step growth curves

Single colonies of *E. coli* MG1655 pBR322 empty vector or pBR322-capRel$^{SJ46}$ were grown overnight in LB. Overnight cultures were back-diluted to $A_{600}$ = 0.05 in 25 ml fresh LB and grown to $A_{600}$ -0.3 at 25 °C. Ten millilitres of each culture was infected with T4 phage at an MOI of 0.05 or SECΦ27 phage at an MOI of 0.01 in LB at 25 °C and phages were allowed to adsorb for 10 min before serial dilution in LB three times (1:100, 1:10, 1:10 serial dilution) to three flasks. Then, at indicated time points, 100 μl of infected cells from the corresponding dilution flask were mixed with 100 μl of indicator cells MG1655 pBR322 empty vector ($A_{600}$ -0.3), and the mixtures were mixed with 4 ml of LB + 0.5% agar and spread on LB + 1.2% agar plates. Plates were incubated overnight at 25 °C and plaques were enumerated the following day. PFU was calculated based on the dilution flask samples were taken from. Data reported are the mean and individual data points from 3 biological replicates.

## Adsorption assays

Single colonies of *E. coli* MG1655 pBR322-capRel$^{SJ46}$ or pBR322 empty vector were grown overnight in LB. Overnight cultures were back-diluted to $A_{600}$ = 0.05 in 25 ml of fresh LB and grown to $A_{600}$ = 0.2

at 25 °C. Cells or a control containing only media were infected with phage T4 or SECΦ27 at MOI = 0.1 and incubated at 25 °C for 10 min to allow for adsorption. Cells were pelleted at 9,000$g$ for 2 min and unadsorbed phages from the supernatant were serial diluted, mixed with indicator cells MG1655 pBR322 empty vector + top agar and plated. Plates were incubated overnight at 25 °C and plaques were enumerated the following day. Percent adsorption was calculated by normalizing unadsorbed phages from each sample to media control. Data reported are the mean and individual data points from 3 biological replicates.

## Western blot of CapRel$^{SJ46}$ and Gp57 after phage infection

For immunoblotting of CapRel$^{SJ46}$, single colonies of *E. coli* MG1655 pBR322-His$_6$-capRel$^{SJ46}$ or *E. coli* MG1655 genomic *His$_6$-capRel$^{SJ46}$* under its native promoter were grown overnight in LB. Overnight cultures were back-diluted to $A_{600}$ = 0.05 in 25 ml fresh LB and grown to $A_{600}$ = 0.2 at 25 °C. Cells were infected with phage SECΦ27 at MOI = 100, and incubated at 25 °C during the experiment. At each indicated time point (0, 10, 20, 40, 60 min), $A_{600}$ was measured and 1 ml of cells was pelleted at 21,000$g$ for 2 min at 4 °C. Supernatant was removed and pellets were flash-frozen in liquid nitrogen. Pellets were thawed and resuspended in 1× Laemmli sample buffer (Bio-Rad) supplemented with 2-mercaptoethanol with $A_{600}$ normalized. Samples were then boiled at 95 °C and analysed by 12% SDS–PAGE and transferred to a 0.45 μm PVDF membrane. Anti-His$_6$ antibody (Invitrogen) was used at a final concentration of 1:1000, and SuperSignal West Femto Maximum Sensitivity Substrate (Thermo Fisher) was used to develop the blots. Blots were imaged by a ChemiDoc Imaging system (Bio-Rad). Blots were stained with Coomassie stain and imaged as loading control. Image shown is a representative of two independent biological replicates.

For immunoblotting of Gp57, single colonies of *E. coli* MG1655 containing genomic *gp57-HA* under its native promoter were grown overnight in LB. Overnight cultures were back-diluted to $A_{600}$ = 0.05 in 25 ml of fresh LB and grown to $A_{600}$ = 0.2 at 25 °C. Cells were infected with phage SECΦ27 at MOI = 100, and incubated at 25 °C during the experiment. At each indicated time point (0, 10, 20, 30, 40, 50 min), $A_{600}$ was measured and 1 ml of cells was pelleted at 21,000$g$ for 2 min at 4 °C. Samples were processed as described above with $A_{600}$ normalized. Samples were then boiled at 95 °C and analysed by 12% SDS–PAGE and transferred to a 0.45 μm PVDF membrane. Anti-HA antibody (Cell Signaling Technology) was used at a final concentration of 1:1000, and SuperSignal West Femto Maximum Sensitivity Substrate (Thermo Fisher) was used to develop the blots. Blots were imaged by a ChemiDoc Imaging system (Bio-Rad), then stained with Coomassie stain and imaged as loading control. The experiment was performed three times independently and band intensities were quantified using Fiji. Relative band intensities were calculated by normalizing the summed intensity of both Gp57 bands to the intensity of total proteins by Coomassie stain.

## Western blot of CapRel$^{SJ46}$, CapRel$^{EB}$ or chimera expression levels

Single colonies of *E. coli* MG1655 pBR322-capRel$^{SJ46}$, pBR322-capRel$^{SJ46}$-Flag, pBR322-capRel$^{Ebc}$-Flag or pBR322-chimera-Flag were grown overnight in LB. Overnight cultures were back-diluted to $A_{600}$ = 0.05 in 5 ml fresh LB and grown to $A_{600}$ = 0.2 at 37 °C. $A_{600}$ was measured and 5 ml of cells was pelleted at 4,000$g$ for 5 min. Supernatant was removed and pellets were resuspended in 1× Laemmli sample buffer (Bio-Rad) supplemented with 2-mercaptoethanol with $A_{600}$ normalized. Samples were then boiled at 95 °C and analysed by 12% SDS–PAGE and transferred to a 0.45 μm PVDF membrane. Anti-Flag antibody (Cell Signaling Technology) and anti-GyrA antibody (Inspiralis) were used at a final concentration of 1:1,000, and SuperSignal West Femto Maximum Sensitivity Substrate (Thermo Fisher) was used to develop the blots. Blots were imaged by a ChemiDoc Imaging system (Bio-Rad). Image shown is a representative of 2 independent biological replicates.

## Error-prone PCR mutagenesis of CapRel[SJ46]

The C terminus of CapRel[SJ46] was mutagenized using error-prone PCR-based mutagenesis as described previously[47]. In brief, primers TZ-94 and TZ-95 were used to amplify the C terminus of CapRel[SJ46] using Taq polymerase (NEB) and 0.5 mM MnCl$_2$ was added to the reaction as the mutagenic agent. PCR products were treated with Dpn I, column purified, and inserted into pBR322-CapRel[SJ46] backbone amplified with primer TZ-96 and TZ-97 using Gibson assembly. Gibson products were transformed into DH5α and grown overnight in LB at 37 °C. Overnight cultures were miniprepped to obtain the mutagenized library. Individual colonies were Sanger sequenced to assess the number of mutations. To perform the selection, mutagenized library was electroporated into *E. coli* MG1655 pBAD33-gp57, and plated onto LB plates containing 0.2% arabinose to select for survivors. Colonies were picked and sequenced to identify mutations in CapRel[SJ46], and further validated by constructing plasmid with only single mutants.

## Isolation of phage escape mutants to infect CapRel[SJ46]

The phage evolution experiment was conducted as described previously[48]. In brief, 5 independent populations were evolved in a 96-well plate containing a sensitive host *E. coli* MG1655 pBR322 empty vector and a resistant host *E. coli* MG1655 pBR322-capRel[SJ46]. One control population was evolved with only the sensitive host. Overnight bacterial cultures were back-diluted to $A_{600}$ = 0.1 in LB and 100 μl were seeded into each well. Cells were infected with tenfold serial dilutions of SECΦ27 phage with MOI from $10^2$ to $10^{-4}$, with one well uninfected to monitor for contamination. Plates were sealed with breathable plate seals and incubated at 25 °C for 6 h in a plate shaker at 1,000 rpm. Cleared wells from each population were pooled, pelleted at 4,000g for 20 min to remove bacteria, and the supernatant lysates were transferred to a 96 deep-well block with 40 μl chloroform added to prevent bacterial growth. Lysates were spotted onto both sensitive and resistant hosts to check the defence phenotype. Thirteen rounds of evolution were performed to allow all five populations to overcome CapRel[SJ46] defence. Evolved clones from each evolved population were isolated by plating to single plaques on lawns of resistant host, and control clones from the control population were isolated on a lawn of the sensitive host. Two clones from each population were propagated using the corresponding host and sequenced as described below.

Bas8 escape mutants were isolated by plating a population of phage onto CapRel[SJ46]- or CapRel[EcHT]-containing cells. Twenty microlitres of $10^{11}$ PFU ml$^{-1}$ Bas8 phage mixed with 40 μl overnight culture of *E. coli* MG1655 pBR322-capRel[SJ46] or pBR322-capRel[EcHT] were added to 4 ml LB + 0.5% agar and spread onto LB + 1.2% agar. Plates were incubated at 25 °C overnight. Single plaques were isolated and propagated using the same strain in LB at 25 °C. Amplified phage lysates were pelleted to remove bacteria, and then plated to single plaques and propagated similarly for a second round of isolation to improve purity and sequenced.

## Phage DNA extraction and Illumina sequencing

To extract phage DNA, high-titre phage lysates (>$10^6$ PFU μl$^{-1}$) were treated with DNase I (0.001 U μl$^{-1}$) and RNase A (0.05 mg ml$^{-1}$) at 37 °C for 30 min. 10 mM EDTA was used to inactivate the nucleases. Lysates were then incubated with proteinase K at 50 °C for 30 min to disrupt capsids and release phage DNA. Phage DNA was isolated by ethanol precipitation. In brief, sodium acetate pH 5.2 was added to 300 mM followed by 100% ethanol to a final volume fraction of 70%. Samples were incubated at −80 °C overnight, pelleted at 21,000g for 20 min and supernatant removed. Pellets were washed with 100 μl isopropanol and 200 μl 70% (v/v) ethanol, and then aired dried at room temperature and resuspended in 25 μl 1× TE buffer (10 mM Tris-HCl, 0.1 mM EDTA, pH 8). Concentrations of extracted DNA were measured by NanoDrop (Thermo Fisher Scientific).

To prepare Illumina sequencing libraries, 100–200 ng of genomic DNA was sheared in a Diagenode Bioruptor 300 sonicator water bath for twenty 30 s cycles at maximum intensity. Sheared genomic DNA was purified using Ampure XP beads, followed by end repair, 3′ adenylation, and adapter ligation. Barcodes were added to both 5′ and 3′ ends by PCR with primers that anneal to the Illumina adapters. The libraries were cleaned by Ampure XP beads using a double cut to elute fragment sizes matching the read lengths of the sequencing run. Libraries were sequenced on an Illumina MiSeq at the MIT BioMicro Center. Illumina reads were assembled to the reference genomes using Geneious Prime 2021.2.2.

## Mass spectrometry of phages

Wild-type or mutant (L114P in Gp57, evolved clone 1 from population 3) SECΦ27 phage were propagated in *E. coli* MG1655 for high-titre stocks. In brief, *E. coli* MG1655 ($A_{600}$ = 0.2) in LB were infected with phages at MOI = 0.1 and incubated at 37 °C for 4 h. Cells were pelleted at 4,000g for 10 min and supernatant lysates were filtered through 0.2-μm filters. Five hundred microlitres of phage stocks ($10^{10}$ PFU μl$^{-1}$) were further concentrated with Amicon Ultra filter (MW 100 kDa) and washed twice with 1× FM buffer (20 mM Tris-HCl pH 7.4, 100 mM NaCl, 10 mM MgSO$_4$). Concentrated phage lysates were boiled to denature virions and run on 4–20% SDS–PAGE. Each lane from the gel was excised. Proteins were reduced with 10 mM dithiothreitol (Sigma) for 1 h at 56 °C and then alkylated with 20 mM iodoacetamide (Sigma) for 1 h at 25 °C in the dark. Proteins were then digested with 12.5 ng μl$^{-1}$ modified trypsin (Promega) in 50 μl 100 mM ammonium bicarbonate, pH 8.9 at 25 °C overnight. Peptides were extracted by incubating the gel pieces with 50% acetonitrile/5% formic acid then 100 mM ammonium bicarbonate, repeated twice followed by incubating the gel pieces with 100% acetonitrile then 100 mM ammonium bicarbonate, repeated twice. Each fraction was collected, combined, and reduced to near dryness in a vacuum centrifuge. Peptides were desalted using Pierce Peptide Desalting Spin Columns (Thermo) and then lyophilized. The tryptic peptides were separated by reverse phase HPLC (Thermo Ultimate 3000) using a Thermo PepMap RSLC C18 column over a 90 min gradient before nano-electrospray using an Exploris mass spectrometer (Thermo). Solvent A was 0.1% formic acid in water and solvent B was 0.1% formic acid in acetonitrile. Detected peptides were mapped to SECΦ27 protein sequences and the abundance of proteins were estimated by number of spectrum counts/molecular mass to normalize for protein sizes.

## Co-immunoprecipitation analysis

For immunoprecipitation of CapRel[SJ46] after phage infection, *E. coli* MG1655 containing pBR322-capRel[SJ46]-Flag were grown overnight in LB. Overnight cultures were back-diluted to $A_{600}$ = 0.05 in 175 ml of LB and grown to $A_{600}$ ~0.3 at 25 °C. Cells were infected with wild-type or mutant (L114P in Gp57, evolved clone 1 from population 3) SECΦ27 at MOI = 100 and incubated at 25 °C. At the indicated time points (15 min or 40 min), $A_{600}$ was measured and 50 ml of cells were pelleted at 6,000g for 5 min at 4 °C. Uninfected cells were collected at 0 min before phage infection. Supernatant was removed and cells were resuspended in 900 μl lysis buffer (25 mM Tris-HCl, 150 mM NaCl, 1 mM EDTA, 1% Triton X-100 and 5% glycerol) supplemented with protease inhibitor (Roche), 1 μl per ml Ready-Lyse Lysozyme Solution (Lucigen) and 1 μl per ml benzonase nuclease (Sigma). Samples were lysed by two freeze-thaw cycles, and lysates were normalized by $A_{600}$. Lysates were pelleted at 21,000g for 10 min at 4 °C, and 850 μl of supernatant were incubated with pre-washed anti-Flag M2 magnetic beads (Sigma) for 1 h at 4 °C with end-over-end rotation. Beads were then washed 3 times with lysis buffer containing 350 mM NaCl but free of detergent. On-bead reduction, alkylation and digestion were performed. Proteins were reduced with 10 mM dithiothreitol (Sigma) for 1 h at 56 °C and then alkylated with 20 mM iodoacetamide (Sigma) for 1 h at 25 °C in the dark. Proteins were then digested with modified trypsin (Promega) at an enzyme/substrate ratio of 1:50 in 100 mM ammonium bicarbonate, pH 8 at

25 °C overnight. Trypsin activity was halted by addition of formic acid (99.9 %, Sigma) to a final concentration of 5 %. Peptides were desalted using Pierce Peptide Desalting Spin Columns (Thermo) then lyophilized. The tryptic peptides were subjected to LC–MS/MS as described above. Experiments were performed two times independently and spectral counts are reported. Ratio of spectral counts between Gp57 and CapRel$^{SJ46}$ were calculated and plotted for normalization.

For co-producing CapRel$^{SJ46}$ and Gp57, *E. coli* MG1655 containing pBR322-capRel$^{SJ46}$ or pBR322-capRel$^{SJ46}$-Flag (wild type or mutants) and pBAD33-gp57-HA (wild type or mutants) were grown overnight in M9-glucose. Overnight cultures were back-diluted to $A_{600}$ = 0.05 in 50 ml of M9 (no glucose) and grown to $A_{600}$ ~0.3 at 37 °C. Cells were induced with 0.2% arabinose for 30 min at 37 °C, then $A_{600}$ was measured and cells were pelleted at 4,000 *g* for 10 min at 4 °C. Supernatant was removed and cells were resuspended in 900 µl lysis buffer as described above. Samples were lysed by two freeze-thaw cycles, and lysates were normalized by $A_{600}$. Lysates were pelleted at 21,000 *g* for 10 min at 4 °C, and 850 µl of supernatant were incubated with pre-washed anti-Flag M2 magnetic beads (Sigma) for 1 h at 4 °C with end-over-end rotation. Beads were then washed 3 times with lysis buffer containing 350 mM NaCl. Laemmli sample buffer (Bio-Rad) supplemented with 2-mercaptoethanol was added to beads directly to elute proteins. Samples were boiled at 95 °C and analysed by 12% SDS–PAGE and transferred to a 0.45 µm PVDF membrane. Anti-Flag and anti-HA antibodies (Cell Signaling Technology) were used at a final concentration of 1:1,000, and SuperSignal West Femto Maximum Sensitivity Substrate (Thermo Fisher) was used to develop the blots. Blots were imaged by a ChemiDoc Imaging system (Bio-Rad). Images shown are representatives of 3 independent biological replicates.

## Incorporation assays

For co-producing CapRel$^{SJ46}$ and Gp57, the SECΦ27 major capsid protein, single colonies of *E. coli* MG1655 containing pBR322-capRel$^{SJ46}$ and pBAD33-gp57 (wild-type or L114P variant) or corresponding empty vectors were grown overnight in M9-glucose. Overnight cultures were back-diluted to $A_{600}$ = 0.05 in 25 ml M9-glucose and grown to $A_{600}$ ~0.3 at 37 °C. Cells were pelleted at 4,000 *g* for 5 min at 4 °C and washed once with M9 (no glucose), and then back-diluted to $A_{600}$ = 0.1 in 15 ml M9 (no glucose) and recovered for 45 min at 37 °C. At the beginning of the experiment, cells were induced with 0.2% arabinose. At the indicated time points (0, 10, 20, 30, 40 min), $A_{600}$ was measured and an aliquot of 250 µl of cells was transferred to microcentrifuge tube containing [5,6-³H]uridine (PerkinElmer) (4 µCi ml⁻¹) for transcription measurements or EasyTag EXPRESS-³⁵S Protein Labeling Mix, [³⁵S] (PerkinElmer) at 44 µCi ml⁻¹ for translation measurements. Tubes were incubated at 37 °C for 2 min, then quenched by addition of nonradioactive uridine (1.5 mM) or cysteine and methionine (15 mM each) and incubated for an additional 2 min. Samples were then added to ice cold trichloroacetic acid (TCA) (10% w/v) and incubated at least 30 min on ice to allow for precipitation. Resulting samples were vacuum filtered onto a glass microfibre filter (Whatman, 1820-024) that had been pre-wetted with 5% w/v TCA. Filters were washed with 35× volume of 5% w/v TCA, then with 5× volume of 100% ethanol. Air dried filters were placed in tubes with scintillation fluid and measured in a scintillation counter (PerkinElmer). CPM (Counts Per Million) was normalized to $A_{600}$ and percent incorporation at each time point was calculated by normalizing to *t* = 0. Data reported are the mean and individual data points from three independent biological replicates.

For producing the CapRel$^{SJ46}$ N-terminal toxin domain, single colonies of *E. coli* MG1655 containing pBAD33-capRel$^{SJ46}$(1–272) or an empty vector were grown overnight in M9-glucose. Transcription and translation experiments were done as described above. Data reported are the mean and individual data points from three independent biological replicates.

For phage infection experiments, single colonies of *E. coli* MG1655 harbouring pBR322 empty vector or pBR322-capRel$^{SJ46}$ were grown overnight in LB. Overnight cultures were back-diluted to $A_{600}$ = 0.05 in 25 ml fresh LB and grown to $A_{600}$ ~0.3 at 25 °C. Cells were then diluted to $A_{600}$ = 0.1 in 10 ml LB and infected with wild-type or mutant (L114P in Gp57, evolved clone 1 from population 3) SECΦ27 at MOI = 100 and incubated at 25 °C. At the indicated time points (0, 15, 30, 45 and 60 min), $A_{600}$ was measured and an aliquot of 250 µl of cells was transferred to a microcentrifuge tube containing [5,6-³H]uridine (PerkinElmer) (32 µCi ml⁻¹) for transcription measurements or EasyTag EXPRESS-³⁵S Protein Labeling Mix, [³⁵S] (PerkinElmer) at 88 µCi ml⁻¹ for translation measurements. Tubes were incubated at 25 °C for 4 min, then quenched by addition of nonradioactive uridine (1.5 mM) or cysteine and methionine (15 mM) and incubated for an additional 2 min. Samples were then processed same as above. Data reported are the mean and individual data points from three independent biological replicates. Statistical significance was determined by unpaired, two-tailed Student's *t*-test ($P < 0.05$).

## Homology search, alignment, and conservation analysis

CapRel$^{SJ46}$ was identified in the sequence database from our previous bioinformatic survey of RSH proteins[24] that included gene neighbourhood analysis to identify toxin–antitoxin systems[49]. Bacterial strains containing CapRel$^{SJ46}$, CapRel$^{Ebc}$ or CapRel$^{Kp}$ with 100% amino acid identity were found on NCBI database. Local genomic regions (±10 kb of CapRel) were extracted and annotated for all coding sequences. Prophage genes and intact prophage regions were identified by PHASTER[50]. Additional homologues of CapRel$^{SJ46}$ were identified by ConSurf[51] using PSI-BLAST (default settings) to search UniRef90 database, yielding 44 homologues. For Extended Data Fig. 1, sequences were aligned with MAFFT L-INS-i v7.453 (ref. [52]) with manual curation of the C-terminal region guided by homology modelling of the stand-alone Phrann Gp30 antitoxin using Swiss-Model[53], and with our CapRel$^{SJ46}$ predicted structure as a template. For ConSurf analysis, 52 homologues were used to generate the multiple sequence alignment by MAFFT and used as input. Conservation scores were calculated using the Bayesian method and default settings. For gene neighbourhood analysis, previously identified CapRel and PhRel sequences[24] were reduced to representatives with 65% amino acid identity using MMSeqs v13.45111 easy-cluster (default settings[54]), while retaining proteins of interest. The associated accession numbers were then used as input to FlaGs v1.2.6 (one flanking gene either side of the query and otherwise default settings)[49].

Homologues of the major capsid proteins in BASEL phages were identified by BLASTp[55] searches against each phage genome. Homologues of Gp57 (Gp8$^{Bas4}$, Gp8$^{Bas5}$ and Gp8$^{Bas8}$) were aligned by MUSCLE[56].

## CapRel$^{SJ46}$ preparation for crystallization and HDX-MS

For the production of His$_{10}$–SUMO-tagged CapRel$^{SJ46}$ and CapRel$^{SJ46}$ variants, *E. coli* BL21 (DE3) cells were transformed with pET24d plasmids containing the gene of interest and grown in LB medium to $A_{600}$ of 0.6. Expression of the protein of interest was induced by addition of 0.5 mM IPTG, and cells were grown for 3 h at 30 °C. The culture was then centrifuged, and pellet was resuspended in resuspension buffer (50 mM Tris-HCl pH 8.0, 1.5 M KCl, 2 mM MgCl$_2$, 1 mM TCEP, 0,002% mellitic acid and 1 pastil of protease inhibitors cocktail (Roche)).

Cells were disrupted using a high-pressure homogenizer (Emulsiflex) and the supernatant was separated from the pellet by centrifugation and filtered through 0.45 µm filters. Protein extracts were loaded onto a gravity-flow column (Cytiva) packed with HisPur Nickel resin (Thermo Fisher Scientific), washed with buffer A (50 mM Tris-HCl pH 8, 500 mM NaCl, 500 mM KCl, 1 mM TCEP, 0.002% Melitic acid) and stepwise eluted in buffer A supplemented with 500 mM imidazole. To remove remaining contaminants and imidazole, the elution fraction was immediately transferred to a size-exclusion chromatography (SEC) column Superdex 200 pgcolumn (Cytiva), previously equilibrated in the SEC buffer (50 mM HEPES pH 7.5, 500 mM NaCl, 500 mM KCl, 2 mM MgCl$_2$, 1 mM TCEP, 0.002% mellitic acid). The fractions containing the protein were concentrated to around 1 mg ml⁻¹ and the His tag was removed

by incubating with UlpI protease (1:50 molar ratio) at 4 °C for 30 min. The $His_{10}$–SUMO tag and the protease were then removed by passing the samples over a gravity-flow column (Cytiva) packed with HisPur Nickel resin (Thermo Fisher Scientific). Purity of the sample preparation was assessed spectrophotometrically and by SDS–PAGE. For all the purified protein samples, $A_{260}/A_{280}$ ratio was below 0.6. Samples were stored at −20 °C or concentrated to 7 mg ml$^{-1}$ and used directly in crystallization experiments.

For the purification of the complex containing $His_{10}$–SUMO–CapRel$^{SJ46}$ and $His_{10}$–SUMO–Gp57, E. coli BL21 (DE3) strain containing freshly transformed pET24d-$His_{10}$-SUMO-capRel$^{SJ46}$(Y155A) and pET21a-$His_{10}$-SUMO-gp57 were grown in LB medium to $A_{600}$ of 0.2. This culture was then diluted in fresh LB media and grown until $A_{600}$ of 0.6. Expression of the protein of interest was induced by addition of 0.5 mM IPTG, and cells were grown for overnight at 16 °C. The subsequent purification, Sumo tag cleavage and purity assessment steps were identical to the workflow described above for the all the CapRel$^{SJ46}$ protein variants.

## Crystallization of CapRel$^{SJ46}$

The screening of crystallization conditions of CapRel$^{SJ46}$ was carried out using the sitting-drop vapour-diffusion method. The drops were set up in Swiss (MRC) 96-well two-drop UVP sitting-drop plates using the Mosquito HTS system (TTP Labtech). Drops of 0.1 µl protein and 0.1 µl precipitant solution were equilibrated to 80 µl precipitant solution in the reservoir. Commercially available screens LMB and SG1 (Molecular Dimensions) were used to test crystallization conditions. The condition resulting in protein crystals (LMB screen position C9 for CapRel$^{SJ46}$: 26% w/v PEG 2000 MME 0.1 M Bis-Tris 5.8) were repeated as 2 µl drops. The final crystallization drop was made by mixing in a 1:1 ratio the crystallization mother liquor at pH 5.8 and CapRel$^{SJ46}$ at pH 8.0. Crystals were collected using suitable cryo-protecting solutions (mother liquor supplemented with 20% glycerol) and vitrified in liquid $N_2$ for transport and storage before X-ray exposure. X-ray diffraction data was collected at the SOLEIL synchrotron (Gif-sur-Yvette, Paris, France) on the Proxima 1 (PX1) and Proxima 2A (PX2A) beamlines using an Eiger-X 16M detector. Because of the high anisotropic nature of the data from all the crystals we performed anisotropic cut-off and correction of the merged intensity data as implemented on the STARANISO server (http://staraniso.globalphasing.org/) using the DEBYE and STARANISO programs. The analysis of the data suggested a resolution of 2.31 Å (with 2.31 Å in $a^*$, 2.85 Å in $b^*$ and 2.72 Å in $c^*$).

## Structure determination

The data were processed with the XDS suite[57] and scaled with Aimless. In all cases, the unit cell content was estimated with the program MATTHEW COEF from the CCP4 program suite[58]. Molecular replacement was performed with Phaser[59]. The crystals of CapRel$^{SJ46}$ diffracted on average to ~2.3 Å. We used the coordinates of Rel$_{Tt}^{NTD}$ (PDB ID 6S2T) as a search model for the toxSYNTH domain[60]. The MR solution from Phaser was used in combination with Rosetta as implemented in the MR-Rosetta[61] suit from the Phenix package[62]. After several iterations of manual building with Coot[63] and maximum likelihood refinement as implemented in Buster/TNT[64], the model was extended to cover all the residues ($R/R_{free}$ of 21.5/26.0). Extended Data Table 1 details all the X-ray data collection and refinement statistics.

## Isothermal titration calorimetry

For all ITC measurements CapRel$^{SJ46}$ samples were prepared from the pET24d-$His_{10}$-SUMO-capRel$^{SJ46}$ as detailed above. In the case of Gp57, E. coli BL21 (DE3) cells were transformed with pET21a-$His_{10}$-SUMO-gp57 and grown in LB medium to $A_{600}$ of 0.2. This culture was then diluted in fresh LB media and grown until $A_{600}$ of 0.6. Expression of $His_{10}$–SUMO–Gp57 was induced by addition of 0.1 mM IPTG, and cells were grown for overnight at 16 °C. The subsequent purification, SUMO-tag cleavage

and purity assessment steps were identical to the workflow described above for the all the CapRel$^{SJ46}$ protein variants. After removing the SUMO tag, samples were concentrated to 10 µM and used directly for ITC immediately after purification.

All titrations were performed with an Affinity ITC (TA instruments) at 25 °C. For the titration, CapRel$^{SJ46}$ was loaded in the instrument syringe at 150 µM and Gp57 was used in the cell at 10 µM. The titration was performed in 50 mM HEPES pH 7.5; 500 mM KCl; 500 mM NaCl; 150 mM imidazole; 10 mM $MgCl_2$; 1 mM TCEP; 0.002% mellitic acid. Final concentrations were verified by the absorption using a Nanodrop One (ThermoScientific). All ITC measurements were performed by titrating 2 µl of CapRel$^{SJ46}$ (or a CapRel$^{SJ46}$ variant) into Gp57 (or a Gp57 variant) using a constant stirring rate of 75 rpm. All data were processed, buffer-corrected and analysed using the NanoAnalyse and Origin software packages.

## Multiwavelength light scattering

The sample of the CapRel$^{SJ46}$·Gp57 complex used for the MALS measurements was prepared in the same way as the samples used for hydrogen–deuterium exchange mass spectrometry (HDX-MS): 50 mM HEPES pH 7.5; 500 mM KCl; 500 mM; NaCl; 10 mM $MgCl_2$; 1 mM TCEP; 0.002% mellitic acid at a concentration of 5 mg ml$^{-1}$. The measurement was performed in an HPLC Alliance system (Waters) connected to a 2998 PDA detector (Waters), a TREOS II MALS detector (Wyatt Technology) and a RI-501 refractive index detector (Shodex). The data were analysed with Astra 7 suite (Wyatt technology).

## HDX-MS

HDX-MS experiments were performed on an HDX platform composed of a Synapt G2-Si mass spectrometer (Waters Corporation) connected to a nanoAcquity UPLC system. Samples of CapRel$^{SJ46}$ and CapRel$^{SJ46}$ complexed with Gp57 were prepared at a concentration of 20 to 50 µM. For each experiment 5 µl of sample (CapRel$^{SJ46}$ or CapRel$^{SJ46}$-Gp57) were incubated for 1 min, 5 min, 15 min or 60 min in 95 µl of labeling buffer L (50 mM HEPES, 500 mM KCl, 500 mM NaCl, 2 mM $MgCl_2$, 1 mM TCEP, 0.002% mellitic acid, pH 7.5) at 20 °C. The non-deuterated reference points were prepared by replacing buffer L by equilibration buffer E (50 mM HEPES, 500 mM KCl, 500 mM NaCl, 2 mM $MgCl_2$, 1 mM TCEP, 0.002% mellitic acid, pH 7.5). After labelling, the samples are quenched by mixing with 100 µl of pre-chilled quench buffer Q (1.2 % formic acid, pH 2.4). Seventy microlitres of the quenched samples are directly transferred to the Enzymate BEH Pepsin Column (Waters Corporation) at 200 µl min$^{-1}$ and at 20 °C with a pressure 8.5 kPSI. Peptic peptides were trapped for 3 min on an Acquity UPLC BEH C18 VanGuard Pre-column (Waters Corporation) at a 200 µl min$^{-1}$ flow rate in water (0.1% formic acid in HPLC water pH 2.5) before eluted to an Acquity UPLC BEH C18 Column for chromatographic separation. Separation was done with a linear gradient buffer (7–40% gradient of 0.1% formic acid in acetonitrile) at a flow rate of 40 µl min$^{-1}$. Peptides identification and deuteration uptake analysis was performed on the Synapt G2-Si in ESI ± HDMS$^E$ mode (Waters Corporation). Leucine enkephalin was applied for mass accuracy correction and sodium iodide was used as calibration for the mass spectrometer. HDMS$^E$ data were collected by a 20–30 V transfer collision energy ramp. The pepsin column was washed between injections using pepsin wash buffer (1.5 M guanidinium HCl, 4% (v/v) methanol, 0.8% (v/v) formic acid). A blank run was performed between each sample to prevent significant peptide carry-over. Optimized peptide identification and peptide coverage for all samples was performed from undeuterated controls (five replicates). All deuterium time points were performed in triplicate.

## Data treatment and statistical analysis of HDX-MS

The non-deuterated references points were analysed by PLGS (ProteinLynx Global Server 2.5.1, Waters) to identify the peptic peptides belonging to

CapRel[SJ46] or Gp57. Then, all the HDMS[E] data including reference and deuterated samples were processed by DynamX 3.0 (Waters) for deuterium uptake determination. We chose the following filtering parameters: minimum intensity of 1,000, minimum and maximum peptide sequence length of 5 and 20, respectively, minimum MS/MS products of 3, minimum products per amino acid of 0.27, minimum score of 5, and a maximum MH+ error threshold of 15 p.p.m. Data were analysed at peptidic and overall level and manually curated by visual inspection of individual spectra. The overall level is based on the relative fractional uptake (RFU$_{a,t}$), which can be calculated by the following formula:

$$\text{RFU}_{a,t}(\%) = \frac{\gamma_{a,t}}{\text{MaxUptake}_a \times D}$$

where $\gamma_{a,t}$ is the deuterium uptake for peptide a at incubation time $t$, and MaxUptake$_a \times D$ is the theorical maximum uptake in deuterium value that peptide a can take. The ΔRFU compared RFU value between two different experimental conditions, in this case, this is the comparison between CapRel[SJ46] and CapRel[SJ46] + Gp57. Heat maps have been generated in DynamX. All the raw data can be accessed at: https://doi.org/10.6084/m9.figshare.19745089.

## CapRel[SJ46] expression and purification for biochemical assays

Full-length capRel[SJ46] was overexpressed in freshly transformed *E. coli* BL21(DE3) pET24d-N-His$_{10}$-SUMO-capRel[SJ46] pMG25-paSpo co-transformed with the plasmid encoding PaSpo small alarmone hydrolase (SAH) from *Salmonella* phage SSU5, which has been shown to neutralize the toxicity of other toxSAS toxins[24]. Fresh transformants were used to inoculate 800 ml of LB medium (final $A_{600}$ of 0.03) supplemented with 50 μg ml$^{-1}$ kanamycin, 20 μg ml$^{-1}$ chloramphenicol and 0.2% arabinose. Bacterial cultures were grown at 37 °C until an $A_{600}$ of 0.4–0.5 and protein expression was induced with 0.1 mM IPTG (final concentration). Cells were grown for additional 1 h at 30 °C and the biomass was collected by centrifugation (10,000 rpm, for 5 min, JLA-10.500 rotor (Beckman Coulter)).

Cell mass was resuspended in buffer A (750 mM KCl, 500 mM NaCl, 5 mM MgCl$_2$, 40 μM MnCl$_2$, 40 μM zinc acetate, 1 mM mellitic acid, 20 mM imidazole, 10% glycerol, 4 mM β-mercaptoethanol and 25 mM HEPES:KOH pH 8) supplemented with 0.1 mM PMSF and 1 U ml$^{-1}$ of DNase I, and lysed by one passage through a high-pressure cell disrupter (Stansted Fluid Power, 150 MPa). Mellitic acid was added to buffers as it was earlier shown to stabilize *Thermus thermophilus* Rel stringent factor[65]. Cell debris was removed by centrifugation (25,000 rpm for 1 h at 4 °C, JA-25.50 rotor (Beckman Coulter)), the clarified lysate was filtered through a 0.22-μm syringe filter and loaded onto a HisTrap 5 ml HP column (Cytiva) pre-equilibrated in buffer A. The column was washed with 5 column volumes of buffer A, and the protein was eluted using a combination of stepwise and linear gradient (5 column volumes with 0–100% buffer B) of buffer B (750 mM KCl, 500 mM NaCl, 5 mM MgCl$_2$, 40 μM MnCl$_2$, 40 μM Zn(OAc)$_2$, 1 mM mellitic acid, 1 M imidazole, 10% glycerol, 4 mM β-mercaptoethanol, 25 mM HEPES:KOH pH 8). Fractions enriched in CapRel[SJ46] (approximately 40% buffer B) were pooled, totalling approximately 5 ml. The sample was loaded on a HiLoad 16/600 Superdex 200 pg column pre-equilibrated with a high-salt buffer (buffer C; 2 M NaCl, 5 mM MgCl$_2$, 10% glycerol, 4 mM β-mercaptoethanol, 25 mM HEPES:KOH pH 8). The fractions containing CapRel[SJ46] were pooled and applied on a HiPrep 10/26 desalting column (Cytiva) pre-equilibrated with storage buffer (buffer D; 720 mM KCl, 5 mM MgCl$_2$, 40 mM arginine, 40 mM glutamic acid, 10% glycerol, 4 mM β-mercaptoethanol, 25 mM HEPES:KOH pH 8). Fractions containing CapRel[SJ46] were collected (about 14 ml in total) and the His$_{10}$–SUMO tag was cleaved off by addition of 10 μg of His$_6$–Ulp1 per 1 mg CapRel[SJ46] followed by a 30-min incubation on ice. After the His$_{10}$–SUMO tag was cleaved off, the protein was passed through a 5 ml HisTrap HP pre-equilibrated with buffer D

supplemented with 20 mM imidazole. Fractions containing CapRel[SJ46] in the flow-through were collected and concentrated on an Amicon Ultra (Millipore) centrifugal filter device with a 10 kDa cut-off. The purity of protein preparations was assessed by SDS–PAGE. Protein preparations were aliquoted, frozen in liquid nitrogen and stored at –80 °C. Individual single-use aliquots were discarded after the experiment.

## Cell-free translation

Experiments with PURExpress in vitro protein synthesis kit (NEB, E6800) were performed as per the manufacturer's instructions. All reactions were supplemented with 0.8 U μl$^{-1}$ RNase Inhibitor Murine (NEB, M0314S). Purified CapRel[SJ46] protein was used at a final concentration of 250 nM, with *gp57*, *gp57(L114P)* or *gp57(I115F)* as template plasmid at 10 ng μl$^{-1}$. As a mock control CapRel[SJ46] was substituted for equal volume of HEPES:Polymix buffer[66], pH 7.5. After a 10-min incubation at 37 °C, a 1.34 μl aliquot of the reaction mixture was taken and quenched by addition of 13.66 μl of 2× sample buffer (100 mM Tris:HCl pH 6.8, 4% SDS, 0.02% bromophenol blue, 20% glycerol, 20 mM DTT and 4% β-mercaptoethanol), and DHFR template plasmid was added to the remaining reaction mixture at a final concentration of 20 ng μl$^{-1}$. After further incubation at 37 °C for 1 h, the reaction mixture was mixed with ninefold volume of 2× sample buffer, boiled at 98 °C for 5 min, and 5 μl of the mixture was resolved by 18% SDS–PAGE. The SDS–PAGE gel was fixed by incubating for 5 min at room temperature in 50% ethanol solution supplemented with 2% phosphoric acid, washed three times with water for 20 min at room temperature, and stained with 'blue silver' solution (0.12% Brilliant Blue G250 (Sigma-Aldrich, 27815), 10% ammonium sulfate, 10% phosphoric acid, and 20% methanol) overnight at room temperature. After washing with water for 3 h at room temperature, the gel was imaged on an Amersham ImageQuant 800 (Cytiva) imaging system. For tRNA pyrophosphorylation experiments, Gp57, Gp57(L114P), or Gp57(I115F) was produced in a similar reaction mixture without CapRel[SJ46] and DHFR template at 37 °C for 2 h.

## tRNA pyrophosphorylation by CapRel[SJ46]

The reaction mixture containing 5 μM tRNA from *E. coli* MRE600 (Sigma-Aldrich, 10109541001), 500 μM [γ$^{32}$P]ATP, 250 nM CapRel[SJ46] and 1/10 volume of either wild-type Gp57, Gp57(L114P), or Gp57(I115F) product from the PUREsystem in HEPES:Polymix buffer, pH 7.5 (5 mM Mg$^{2+}$ final concentration) supplemented with 1 mM DTT was incubated at 37 °C for 10 min. To visualize phosphorylated tRNA, the reaction sample was mixed in 2 volumes of RNA dye (98% formamide, 10 mM EDTA, 0.3% bromophenol blue and 0.3% xylene cyanol), tRNA was denatured at 37 °C for 10 min and resolved on urea-PAGE in 1× TBE (8 M urea, 8% PAGE). The gel was stained with SYBR Gold (Life technologies, S11494) and exposed to an imaging plate overnight. The imaging plate was imaged by a FLA-3000 (Fujifilm).

## Reporting summary

Further information on research design is available in the Nature Portfolio Reporting Summary linked to this article.

## Data availability

Structural data from this study are available from the Protein Data Bank (PDB) under accession 7ZTB. HDX raw data can be accessed at https://doi.org/10.6084/m9.figshare.19745089. Sequencing data are available in the Sequence Read Archive (SRA) under BioProject PRJNA837951. Materials including strains and plasmids are available upon reasonable request. Source data are provided with this paper.

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

**Acknowledgements** We thank A. Harms for sharing the BASEL phage collection; the MIT BioMicro Center and its staff for their support in sequencing; the MIT Biopolymers and Proteomics Core and its staff for their help in mass spectrometry experiments; W. Verseés for allowing the use of the biophysics facilities at the VUB; the Mass Spectrometry Laboratory at ULiège and Thomas Tilmant for support and assistance regarding MS data acquisition; K. Gozzi and B. Wang for comments on the manuscript; and all members of the Laub laboratory for helpful discussions. G.C.A. and V.H. were supported by the Swedish Research council (grant 2018-00956 within the RIBOTARGET consortium under the framework of JPIAMR, project grants 2017-03783 and 2021-01146 to V.H., project grant 2019-01085 to G.C.A.), the Knut and Alice Wallenberg Foundation (2020.0037 to G.C.A.), the Ragnar Söderberg Foundation (M23/14 to V.H.), the European Regional Development Fund through the Centre of Excellence for Molecular Cell Technology (V.H.), and the Estonian Science Foundation (project grant PRG335 to V.H.). A.G.-P. was supported by Fonds National de Recherche Scientifique (FRFS-WELBIO CR-2017S-03, FNRS CDR J.0068.19, FNRS-EQP UN.025.19 and FNRS-PDR T.0090.22), the European Research Council (CoG DiStRes, no. 864311), the Joint Programming Initiative on Antimicrobial Resistance (JPI-EC-AMR-R.8004.18), the Programme Actions de Recherche Concerté 2016-2021, Fonds Jean Brachet and the Fondation Van Buuren, Chargé de Recherches fellowship from the FNRS no. CR/DM-392 (H.T.). A.C. and K.C.W. are fellows of the FRIA, C.M. is supported as a Research Associate of the FRS–FNRS. C.M. was supported by grant F.4532.22 from the FRS–FNRS. The authors acknowledge the use of the PROXIMA 1 and 2A beamlines at the Soleil synchrotron (Gif-sur-Yvette, France). M.T.L. is an Investigator of the Howard Hughes Medical Institute.

**Author contributions** Experiments were conceived and designed by T.Z., T.K., G.C.A., V.H., A.G.-P. and M.T.L. Phage and bacterial experiments, as well as incorporation and co-immunoprecipitation assays, were done by T.Z. with assistance from M.L. and S.S. Metabolic labelling experiments were done by T.Z. and T.B. Cell-free translation and tRNA pyrophosphorylation assays were done by T.Z. and T.K. CapRel and Gp57 purification was done by T.Z., T.K., H.T., A.C. and A.T. ITC was performed by H.T. and A.C. MALS was done by A.C. and A.T. HDX-MS data acquisition and analysis was performed by C.M. and K.C.W. X-ray data collection and analyses was performed by H.T., A.T., A.C. and A.G.-P. Bioinformatic analyses were performed by T.Z. and G.C.A. Structural modelling was done by T.Z., A.T. and A.G.-P. Figure design, manuscript writing, and editing was done by T.Z., T.K., G.C.A., V.H., A.G.-P. and M.T.L. Project supervision and funding was provided by G.C.A., V.H., A.G.-P. and M.T.L.

**Competing interests** A.G.-P. is co-founder and stockholder of Santero Therapeutics. the other authors declare no competing interests.

**Additional information**
**Correspondence and requests for materials** should be addressed to Vasili Hauryliuk, Abel Garcia-Pino or Michael T. Laub.

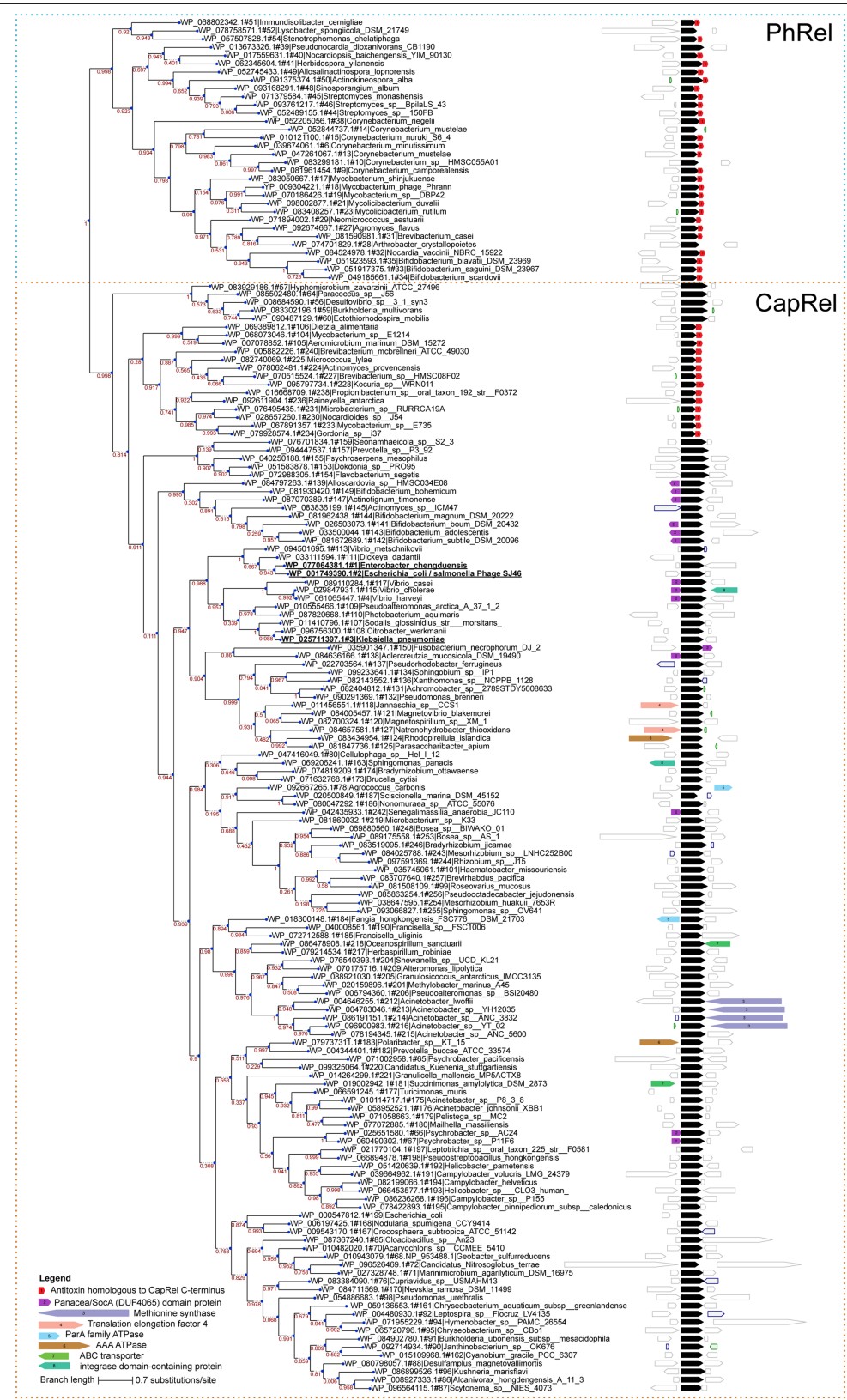

**Extended Data Fig. 1 | CapRel is broadly distributed in different bacteria, and is usually a fused TA system.** FlaGs output for CapRel and PhRel representatives (black arrows). Those proteins studied in this paper are in bold and underlined. Most CapRel systems are fused, with the exception of Actinobacteria where the TA pair is usually unfused. The unfused state is the most common form of the system in the closely related PhRel subfamily of toxSASs, and fusion and/or fission events appear to have occurred independently multiple times. Conserved flanking open reading frames are colored and

numbered by homologous clusters: 1 (red): antitoxin homologous to the CapRel C-terminus, and 2 (purple): Panacea/SocA (DUF4065) domain-containing proteins. For other coloured open reading frames see the legend on the lower left. Arrows with no fill colour and with blue and green outlines are, respectively, pseudogenes and non-coding RNA genes. Red numbers on branches show Maximum Likelihood bootstrap support on a scale of 0-1, where 1 is 100% support.

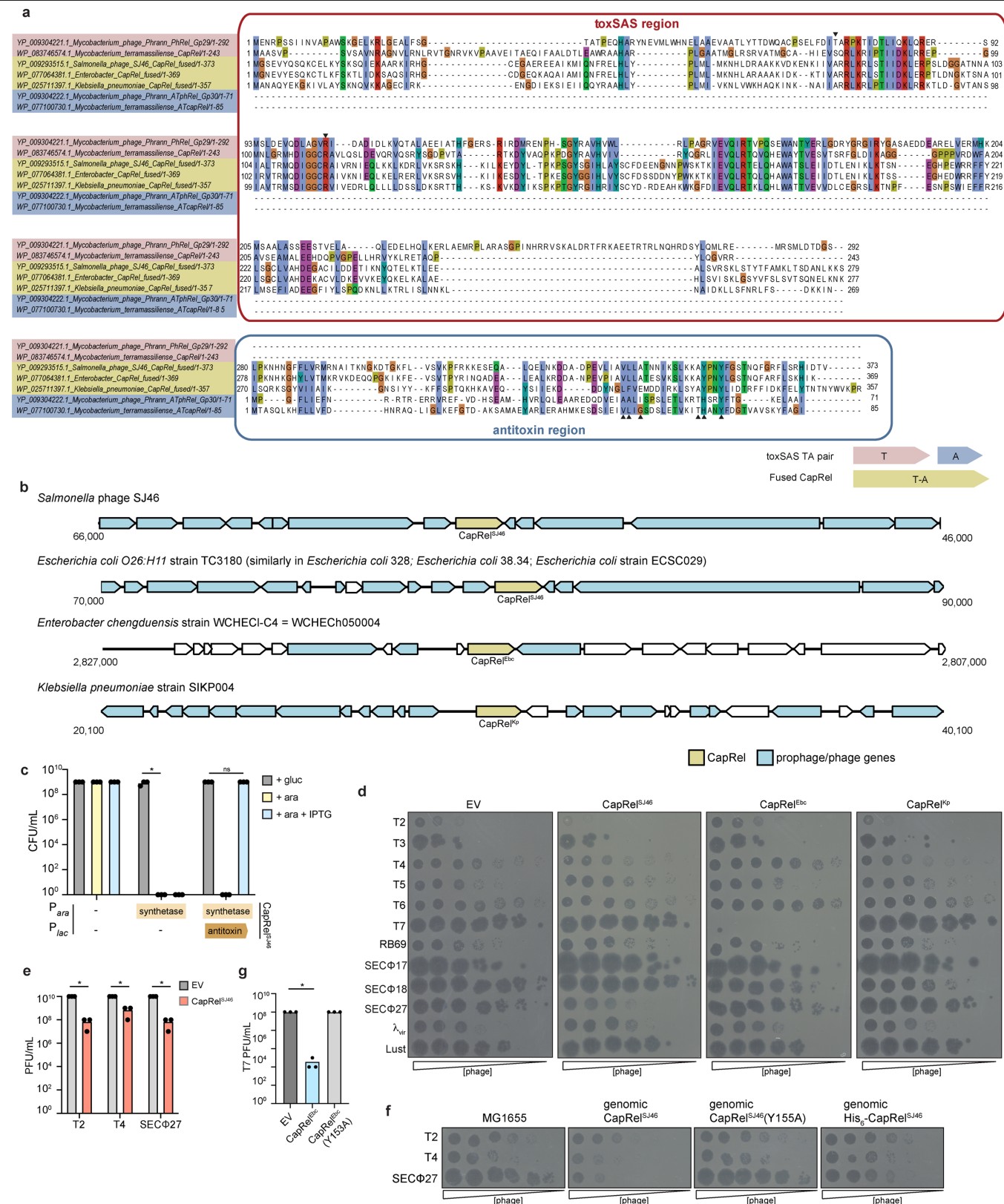

**Extended Data Fig. 2** | See next page for caption.

**Extended Data Fig. 2 | Analysis of CapRel homologs.** (**a**) Sequence alignment comparing fused CapRel systems with related, unfused systems. Alignment of toxSAS PhRel and ATphRel from the *Mycobacterium* phage Phrann, non-fused CapRel and ATcapRel from *Mycobacterium terramassilience,* and the three fused systems CapRel[SJ46], CapRel[Ebc] and CapRel[Kp]. The N-terminal region of fused CapRel systems is a toxSAS toxin domain, while the C-terminal region is homologous to the antitoxins of the PhRel and unfused CapRel TA systems. Substituted sites that rendered CapRel[SJ46] toxic (see Extended Data Fig. 3j) are indicated with black arrowheads. The inset diagram summarises the homologous regions of the bicistronic toxin-antitoxin and fused toxin-antitoxin systems considered here. (**b**) Genome maps of native locations of CapRel[SJ46], CapRel[Ebc] and CapRel[Kp] (+/− 10 kb) with predicted flanking prophage and phage genes. (**c**) Summary of 3 independent replicates of cell viability assay in Fig. 1b. Asterisks indicate $p = 0.007$ (unpaired two-tailed t-test). (**d**) Serial dilutions of the phages indicated spotted on lawns of cells producing CapRel[SJ46], CapRel[Ebc], or CapRel[Kp] or harboring an empty vector (EV). (**e**) Summary of 3 independent replicates of phage spotting assay in Fig. 1d. Asterisks indicate $p = 10^{-10}$ (T2, SECΦ27), $10^{-6}$ (T4) (unpaired two-tailed t-test). (**f**) Serial dilutions of the phages indicated spotted on lawns of cells containing genomic CapRel[SJ46], CapRel[SJ46] (Y155A) or His$_6$-CapRel[SJ46]. (**g**) Summary of 3 independent replicates of phage spotting assay in Fig. 1g. Asterisk indicates $p = 10^{-22}$ (unpaired two-tailed t-test).

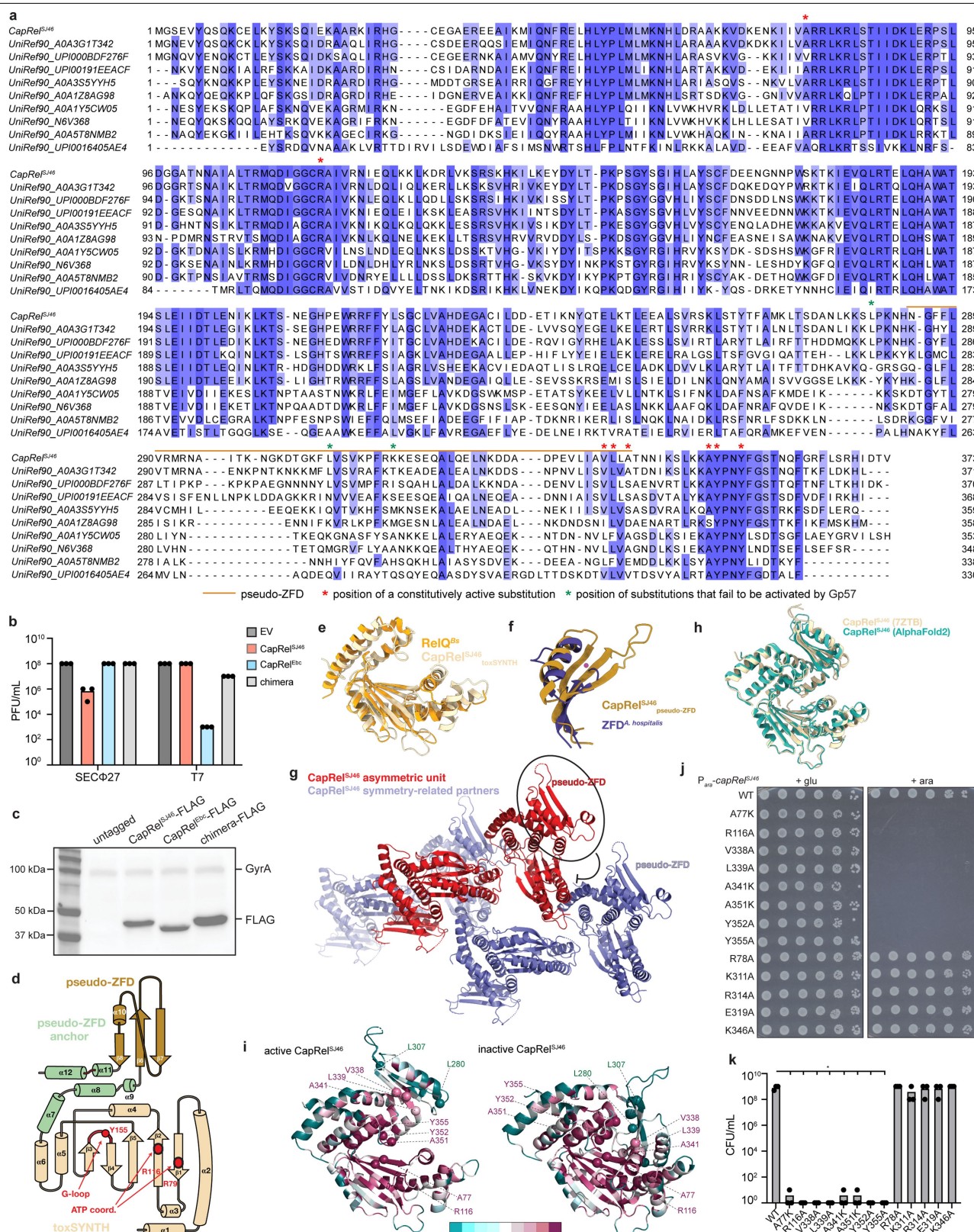

**Extended Data Fig. 3** | See next page for caption.

**Extended Data Fig. 3 | Structural analysis of CapRel$^{SJ46}$.** (**a**) Alignment of CapRel$^{SJ46}$ and diverse fused CapRel homologs, with labels indicating the pseudo-ZFD and location of substitutions that render CapRel$^{SJ46}$ constitutively active or unable to be activated by Gp57, the SECΦ27 major capsid protein. (**b**) Summary of 3 independent replicates of phage spotting assay in Fig. 2b. (**c**) Immunoblot of untagged CapRel chimera or FLAG-tagged CapRel$^{SJ46}$, CapRel$^{Ebc}$ or CapRel chimera. GyrA is included as a loading control. Image shown is a representative of 2 biological replicates. (**d**) Topology of CapRel$^{SJ46}$. The toxSYNTH domain is colored in light yellow, the pseudo-ZFD in dark gold and the regions that anchor pseudo-ZFD to toxSYNTH are in green. The adenine coordinating R79 and R116 are shown as red dots and the G-loop is colored in red. (**e**) Superposition of the toxSYNTH domain of CapRel$^{SJ46}$ (colored in light yellow) onto RelQ (PDB ID: 5DEC, colored in light orange) from *Bacillus subtilis*. (**f**) Superposition of the pseudo-ZFD of CapRel$^{SJ46}$ (colored in dark gold) onto the ZFD transcription factor of *Acidianus hospitalis* (2LVH, colored in purple). (**g**) Analysis of asymmetric unit and symmetry-related partners of CapRel$^{SJ46}$ crystal packing. Black arrow indicates steric clash that would arise if CapRel$^{SJ46}$ were in a closed conformation. (**h**) Superposition of the crystal structure of CapRel$^{SJ46}$ (colored in light yellow) onto the structure of the open state predicted by AlphaFold (colored in green). (**i**) Structures of the open (*left*; from crystal structure) or closed (*right*; AlphaFold prediction) conformations of CapRel$^{SJ46}$ color coded by the conservation score of each amino acid calculated by ConSurf. Substitutions that render CapRel$^{SJ46}$ constitutively active or unable to be activated by Gp57 are labeled as spheres. (**j**) Serial dilutions of cells expressing the indicated variant of CapRel$^{SJ46}$ from an arabinose-inducible promoter on media containing glucose (*left*) or arabinose (*right*). (**k**) Summary of 3 independent replicates of cell viability assay in Extended Data Fig. 3j under arabinose induction. Asterisks indicate p = 0.007 (unpaired two-tailed t-test).

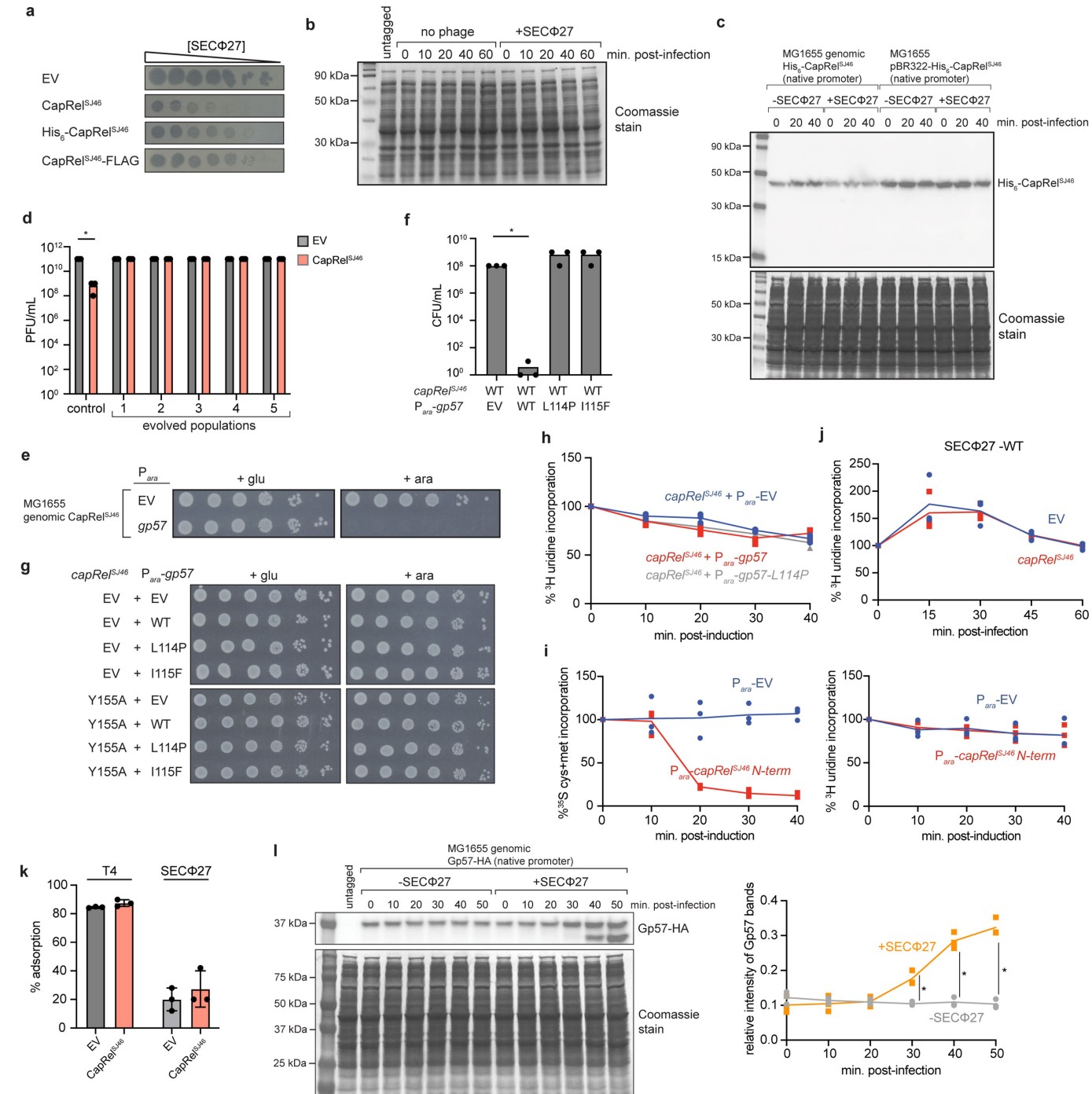

**Extended Data Fig. 4** | See next page for caption.

**Extended Data Fig. 4 | Gp57 from SECΦ27 triggers CapRel$^{SJ46}$ to inhibit translation, not transcription.** (**a**) Serial dilutions of phage SECΦ27 spotted on lawns of cells producing CapRel$^{SJ46}$, His$_6$-CapRel$^{SJ46}$, or CapRel$^{SJ46}$-FLAG, or harboring an empty vector (EV). (**b**) Loading control for Fig. 3a shown as total protein levels stained by Coomassie stain. Image shown is a representative of 2 biological replicates. (**c**) Immunoblot of His$_6$-CapRel$^{SJ46}$ expressed from the bacterial genome or a low-copy number plasmid following infection with SECΦ27 (MOI = 100) compared to an uninfected control. Total protein levels stained by Coomassie stain is included as a loading control. Images shown are representatives of 2 biological replicates. (**d**) Summary of 3 independent replicates of phage spotting assay in Fig. 3c. Asterisk indicates p = 10$^{-10}$ (unpaired two-tailed t-test). (**e**) Serial dilutions on media containing glucose (*left*) or arabinose (*right*) of cells expressing CapRel$^{SJ46}$ from its native promoter in the bacterial genome and expressing Gp57 from an arabinose-inducible promoter or an empty vector. (**f**) Summary of 3 independent replicates of cell viability assay in Fig. 3g under arabinose induction. Asterisk indicates p = 10$^{-30}$ (unpaired two-tailed t-test). (**g**) Serial dilutions on media containing glucose (*left*) or arabinose (*right*) of cells expressing CapRel$^{SJ46}$(Y155A) from its native

promoter or an empty vector and expressing the indicated variant of Gp57 from an arabinose-inducible promoter. (**h**) Cells harboring CapRel$^{SJ46}$ and producing the wild-type or L114P variant of Gp57 (expressed from an arabinose-inducible promoter) or harboring an empty vector were pulse-labeled with $^3$H-uridine at the times indicated post-addition of arabinose. (**i**) Cells producing the CapRel$^{SJ46}$ N-terminal toxin domain (expressed from an arabinose-inducible promoter) or harboring an empty vector were pulse-labeled with $^{35}$S-Cys/Met (*left*) or $^3$H-uridine (*right*) at the times indicated post-addition of arabinose. (**j**) Same as (**h**) but for cells carrying CapRel$^{SJ46}$ or an empty vector and at times post-infection with SECΦ27 at MOI = 100. (**k**) Adsorption of T4 and SECΦ27 with cells containing CapRel$^{SJ46}$ or an empty vector. Data reported are the mean +/− SD from 3 biological replicates. (**l**) Immunoblot of Gp57-HA expressed from its native promoter in the bacterial genome following infection with SECΦ27. Total protein level stained by Coomassie stain is included as a loading control (*left*). Quantification of the relative intensities of both Gp57-HA bands normalized to the loading control from 3 independent replicates (*right*). Asterisks indicate p = 0.004 (30 min), 0.0003 (40 min), 0.0002 (50 min) (unpaired two-tailed t-test).

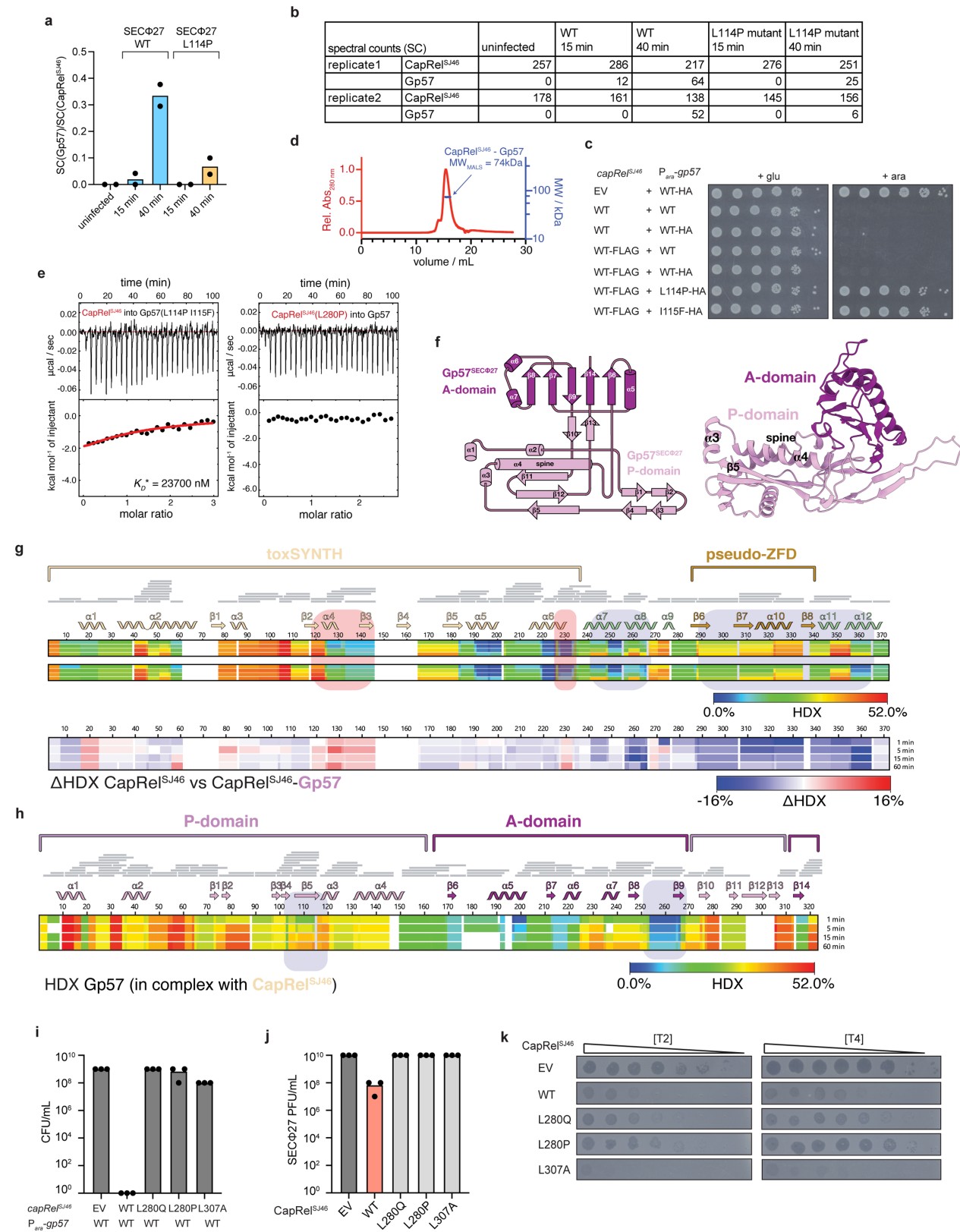

**Extended Data Fig. 5** | See next page for caption.

**Extended Data Fig. 5 | Characterization of the CapRel^SJ46-Gp57 interaction.**
(**a**) Immunoprecipitation of CapRel^SJ46-FLAG from cells infected with wild-type SECΦ27 or mutant phage that produces Gp57(L114P), followed by mass spectrometry. Spectrum counts (SC) of Gp57 that had co-precipitated with CapRel^SJ46 were normalized to the spectrum counts of CapRel^SJ46. Data reported are 2 biological replicates. (**b**) Same as in (**a**) but showing spectrum counts of CapRel^SJ46 and Gp57 in two independent replicates. (**c**) Serial dilutions on media containing glucose (*left*) or arabinose (*right*) of cells producing CapRel^SJ46 or CapRel^SJ46-FLAG, each expressed from its native promoter, and the indicated variant of untagged or HA-tagged version of Gp57, expressed from an arabinose-inducible promoter. (**d**) SEC-MALS analysis of CapRel^SJ46-Gp57 complex, revealing a molecular weight of 74 kDa. The monomers of CapRel^SJ46 and Gp57 are predicted to be 42 and 36 kDa, respectively. (**e**) Binding of CapRel^SJ46 to Gp57 (L114P I115F) *(left)* or CapRel^SJ46(L280P) to Gp57 *(right)*

monitored by isothermal titration calorimetry (ITC). (**f**) Topology and cartoon representation of SECΦ27 Gp57. The P-domain is colored in pink and the A-domain in violet. (**g**) Heat maps representing the HDX of CapRel^SJ46 (*top*) and CapRel^SJ46-Gp57 complex (*center*) and the ΔHDX (*bottom*). Regions involved in strong uptake such as residues 115-145 and 225-235 (which includes the active site β-strand β2 and the G-loop) are shaded in red and regions involved in strong protection 240-268 and 288-366 (which include both anchors and the pseudo-ZFD) are shaded in blue. (**h**) Heat map representing the HDX of Gp57 in the complex with CapRel^SJ46. Shaded regions highlight areas of variable HDX signal that indicate these regions are involved in the CapRel^SJ46-Gp57 interface. (**i**) Summary of 3 independent replicates of cell viability assay in Fig. 4f under arabinose induction. (**j**) Summary of 3 independent replicates of phage spotting assay in Fig. 4g. (**k**) Serial dilutions of T2 and T4 phage spotted on cells producing the indicated mutant of CapRel^SJ46 or harboring an empty vector.

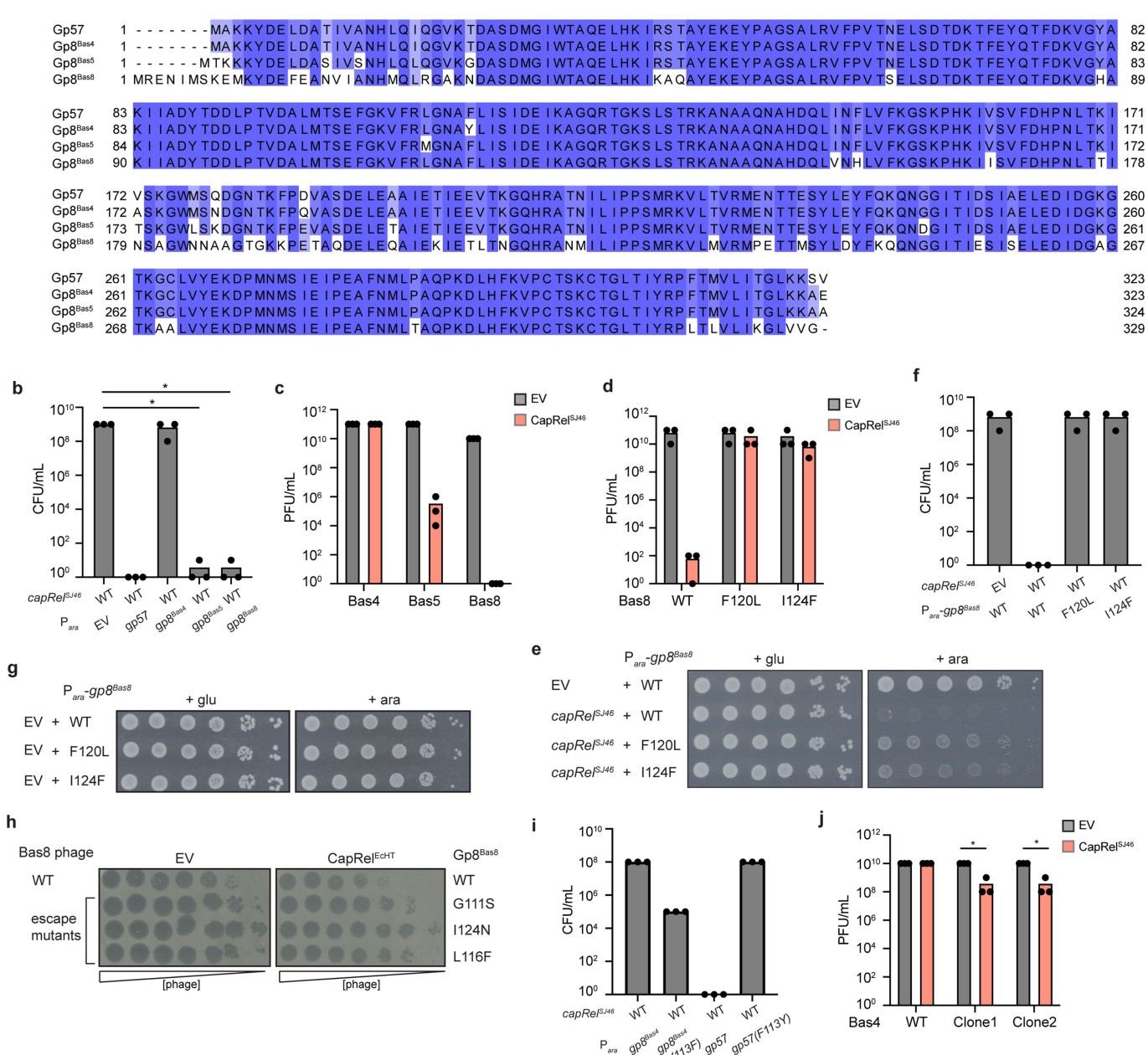

**Extended Data Fig. 6 | The major capsid proteins from multiple, related phages activate CapRel^{SJ46}. (a)** Multiple sequence alignment of the major capsid proteins from phages SECΦ27, Bas4, Bas5 and Bas8. **(b)** Summary of 3 independent replicates of cell viability assay in Fig. 5a under arabinose induction. Asterisks indicate p = 10^{-34} (unpaired two-tailed t-test). **(c)** Summary of 3 independent replicates of phage spotting assay in Fig. 5b. **(d)** Summary of 3 independent replicates of phage spotting assay in Fig. 5c. **(e)** Serial dilutions on media containing glucose (*left*) or arabinose (*right*) of cells expressing CapRel^{SJ46} from its native promoter or harboring an empty vector and producing the indicated variant of the Bas8 major capsid protein (Gp8^{Bas8})

from an arabinose-inducible promoter. **(f)** Summary of 3 independent replicates of cell viability assay in Extended Data Fig. 6e under arabinose induction. **(g)** Serial dilutions on media containing glucose (*left*) or arabinose (*right*) of cells containing an empty vector and producing the indicated variant of the Bas8 major capsid protein from an arabinose-inducible promoter. **(h)** Serial dilutions of the phages indicated spotted on lawns of cells harboring CapRel^{EcHT} or an empty vector. **(i)** Summary of 3 independent replicates of cell viability assay in Fig. 5e under arabinose induction. **(j)** Summary of 3 independent replicates of phage spotting assay in Fig. 5f. Asterisks indicate p = 10^{-6} (unpaired two-tailed t-test).

**Extended Data Table 1 | X-ray data collection and processing**

| Sample | CapRel[SJ46] |
|---|---|
| Diffraction source | Soleil PX1 |
| Wavelength (Å) | 0.9786 |
| Temperature (K) | 100.0 |
| Detector | Eiger-X 16M |
| Crystal-detector distance (mm) | 332.44 |
| Rotation range per image (°) | 0.01 |
| Exposure time per image (s) | 0.01 |
| Space group | $P2_1$ |
| $a, b, c$ (Å) | 49.6 136.2 57.8 |
| $\alpha, \beta, \gamma$ (°) | 90.0, 102.7, 90.0 |
| Mosaicity (°) | 0.20 |
| Resolution range (Å) | 68.10 − 2.31 |
| Total N°. of reflections | 147846 (7291) |
| N°. of unique reflections | 20877 (1044) |
| Completeness (ellipsoidal %) | 91.9 (63.1) |
| Redundancy | 7.1 (7.0) |
| $\langle I/\sigma(I) \rangle$ | 11.2 (1.5) |
| $CC_{1/2}$ | 0.998 (0.712) |
| $R_{pim}$ | 0.051 (0.531) |
| Overall $B$ factor / Wilson plot (Å$^2$) | 54.7 |
| R-factor (%) | 21.2 |
| $R_{free}$-factor (%) | 26.5 |
| Ramachandran profile (%) | |
| Core | 97.9 |
| Allowed | 2.1 |
| Outliers | 0.0 |
| R.m.s. deviations | |
| Bond lengths (Å) | 0.013 |
| Bond angles (°) | 1.52 |
| Number of atoms | 5534 |
| Macromolecules | 5304 |
| Solvent | 228 |
| Other | 3 |
| B-factors (Å$^2$) | |
| All atoms | 68.7 |
| Macromolecules | 69.2 |
| Solvent atoms | 55.8 |
| Other atoms | 63.6 |
| PDB ID | 7ZTB |

The $CC_{1/2}$ criterion was used to determine the resolution range. Values for the outer shell are given in parentheses.

**Extended Data Table 2 | Mass spectrometry analysis of SECΦ27 phage lysates (wild type and mutant producing Gp57(L114P))**

| Gene product | annotation | MW (kDa) | WT Gp57 | | | Gp57(L114P) mutant | | |
|---|---|---|---|---|---|---|---|---|
| | | | Spectrum count (SC) | SC/MW | coverage (%) | Spectrum count (SC) | SC/MW | coverage (%) |
| Gp57 | DUF2184 domain-containing protein | 36 | 878 | 24.39 | 86 | 548 | 15.22 | 92 |
| Gp55 | HtjA | 18 | 325 | 18.06 | 88 | 190 | 10.56 | 87 |
| Gp63 | tail protein/depolymerase | 25 | 333 | 13.32 | 79 | 220 | 8.80 | 78 |
| Gp66 | tail tape-measure protein | 98 | 197 | 2.01 | 58 | 168 | 1.71 | 56 |
| Gp81 | tail fiber protein | 72 | 121 | 1.68 | 44 | 88 | 1.22 | 44 |
| Gp52 | portal protein | 48 | 105 | 2.19 | 71 | 102 | 2.13 | 71 |
| Gp71 | tail fiber protein | 132 | 99 | 0.75 | 39 | 73 | 0.55 | 35 |
| Gp56 | scaffolding protein | 28 | 68 | 2.43 | 67 | 69 | 2.46 | 69 |
| Gp77 | recombinase | 24 | 40 | 1.67 | 65 | 15 | 0.63 | 36 |
| Gp20 | hypothetical protein | 16 | 11 | 0.69 | 43 | 13 | 0.81 | 43 |
| Gp78 | ssDNA-binding protein | 17 | 24 | 1.41 | 77 | 16 | 0.94 | 66 |
| Gp59 | head-tail adaptor | 16 | 11 | 0.69 | 26 | 10 | 0.63 | 17 |
| Gp62 | structural protein | 15 | 20 | 1.33 | 44 | 16 | 1.07 | 44 |
| Gp51 | terminase large subunit | 59 | 10 | 0.17 | 11 | 5 | 0.08 | 8.4 |
| Gp54 | major head subunit precursor | 41 | 6 | 0.15 | 6.2 | 15 | 0.37 | 26 |
| Gp68 | minor tail protein L | 28 | 15 | 0.54 | 33 | 9 | 0.32 | 21 |
| Gp60 | ribonucleoside-triphosphate reductase | 14 | 10 | 0.71 | 29 | 9 | 0.64 | 29 |
| Gp1 | DNA adenine methyltransferase | 28 | 4 | 0.14 | 12 | 6 | 0.21 | 26 |
| Gp79 | chaperone of endosialidase | 26 | 14 | 0.54 | 41 | 9 | 0.35 | 33 |
| Gp14 | DNA-cytosine methylase | 27 | 9 | 0.33 | 18 | 9 | 0.33 | 20 |
| Gp67 | minor tail protein | 13 | 13 | 1.00 | 49 | 8 | 0.62 | 26 |
| Gp75 | exodeoxyribonuclease | 41 | 5 | 0.12 | 12 | 7 | 0.17 | 18 |
| Gp39 | ATP-binding protein | 23 | 5 | 0.22 | 20 | 5 | 0.22 | 20 |
| Gp70 | tail assembly protein | 21 | 6 | 0.29 | 35 | 5 | 0.24 | 31 |
| Gp72 | hypothetical protein | 22 | 4 | 0.18 | 16 | 0 | 0.00 | 0 |
| Gp3 | hypothetical protein | 7 | 5 | 0.71 | 19 | 4 | 0.57 | 19 |
| Gp53 | head morphogenesis protein | 29 | 5 | 0.17 | 11 | 2 | 0.07 | 8 |
| Gp73 | outer membrane protein | 10 | 4 | 0.40 | 42 | 0 | 0.00 | 0 |
| Gp80 | hypothetical protein | 11 | 3 | 0.27 | 21 | 2 | 0.18 | 21 |
| Gp22 | hypothetical protein | 10 | 0 | 0.00 | 0 | 4 | 0.40 | 24 |
| Gp69 | C40 family peptidase | 28 | 3 | 0.11 | 9.5 | 0 | 0.00 | 0 |

# Reporting Summary

## Statistics

For all statistical analyses, confirm that the following items are present in the figure legend, table legend, main text, or Methods section.

| n/a | Confirmed | |
|---|---|---|
| ☐ | ☒ | The exact sample size (*n*) for each experimental group/condition, given as a discrete number and unit of measurement |
| ☐ | ☒ | A statement on whether measurements were taken from distinct samples or whether the same sample was measured repeatedly |
| ☐ | ☒ | The statistical test(s) used AND whether they are one- or two-sided<br>*Only common tests should be described solely by name; describe more complex techniques in the Methods section.* |
| ☒ | ☐ | A description of all covariates tested |
| ☒ | ☐ | A description of any assumptions or corrections, such as tests of normality and adjustment for multiple comparisons |
| ☐ | ☒ | A full description of the statistical parameters including central tendency (e.g. means) or other basic estimates (e.g. regression coefficient) AND variation (e.g. standard deviation) or associated estimates of uncertainty (e.g. confidence intervals) |
| ☐ | ☒ | For null hypothesis testing, the test statistic (e.g. *F*, *t*, *r*) with confidence intervals, effect sizes, degrees of freedom and *P* value noted<br>*Give P values as exact values whenever suitable.* |
| ☒ | ☐ | For Bayesian analysis, information on the choice of priors and Markov chain Monte Carlo settings |
| ☒ | ☐ | For hierarchical and complex designs, identification of the appropriate level for tests and full reporting of outcomes |
| ☒ | ☐ | Estimates of effect sizes (e.g. Cohen's *d*, Pearson's *r*), indicating how they were calculated |

*Our web collection on statistics for biologists contains articles on many of the points above.*

## Software and code

Policy information about availability of computer code

| Data collection | Biotek Gen5 v. 3.02 for growth curve data.<br>NCBI BLAST for homology searches.<br>PHASTER web server for searching prophage regions.<br>Consurf 2016 web server for identifying additional homologs.<br>MAFFT L-INS-i v7.453 for aligning sequences.<br>FlaGs v1.2.6 for gene neighborhood analysis. |
|---|---|
| Data analysis | Phenix 1.20.1-4487, Coot 0.9.8.1, BUSTER / TNT 2.10.4, Fiji 2.1.0/1.53c, PyMOL 2.4.2, Prism 9.4.0, Geneious 2022.0.2, MMSeqs v13.45111 |

For manuscripts utilizing custom algorithms or software that are central to the research but not yet described in published literature, software must be made available to editors and reviewers. We strongly encourage code deposition in a community repository (e.g. GitHub). See the Nature Portfolio guidelines for submitting code & software for further information.

## Data

Policy information about availability of data

All manuscripts must include a data availability statement. This statement should provide the following information, where applicable:

- Accession codes, unique identifiers, or web links for publicly available datasets
- A description of any restrictions on data availability
- For clinical datasets or third party data, please ensure that the statement adheres to our policy

Structural data in this study are available in PDB (7ZTB). HDX data raw data can be accessed at: doi.org/10.6084/m9.figshare.19745089. Sequencing data are available in the Sequence Read Archive (SRA) under BioProject PRJNA837951. All other data are available in the manuscript or the supplementary materials. Source

data for graphs in Fig. 1-5 and Extended Data Fig. 4 are available. Other previously published structures are available in PDB (5DEC, 6S2T, 2LVH). UniRef90 database is publicly available. Reference phage genomes are publicly available: SECΦ27(NC_047938.1), Bas04 (MZ501069.1), Bas08(MZ501059.1). Materials including strains and plasmids are available upon reasonable request.

# Field-specific reporting

Please select the one below that is the best fit for your research. If you are not sure, read the appropriate sections before making your selection.

☒ Life sciences ☐ Behavioural & social sciences ☐ Ecological, evolutionary & environmental sciences

For a reference copy of the document with all sections, see nature.com/documents/nr-reporting-summary-flat.pdf

# Life sciences study design

All studies must disclose on these points even when the disclosure is negative.

| Sample size | Sample sizes were chosen based on the number needed to reliably determine differences between groups. Given large effect sizes, we chose to replicate experiments in triplicate as is routine to simply indicate reproducibility. |
| --- | --- |
| Data exclusions | No data were excluded from analysis. |
| Replication | All experimental findings were repeated at least twice. All reported results were successfully reproduced. |
| Randomization | No experimental groups or control groups were subjectively chosen and there are no covariates to control for as experiments were done in isogenic strains. No randomization is required. |
| Blinding | Blinding was not required because all data were obtained objectively and had strong effect sizes over multiple independent replicates and raw data are reported in the manuscript. |

# Reporting for specific materials, systems and methods

We require information from authors about some types of materials, experimental systems and methods used in many studies. Here, indicate whether each material, system or method listed is relevant to your study. If you are not sure if a list item applies to your research, read the appropriate section before selecting a response.

## Materials & experimental systems

| n/a | Involved in the study |
| --- | --- |
| ☐ | ☒ Antibodies |
| ☒ | ☐ Eukaryotic cell lines |
| ☒ | ☐ Palaeontology and archaeology |
| ☒ | ☐ Animals and other organisms |
| ☒ | ☐ Human research participants |
| ☒ | ☐ Clinical data |
| ☒ | ☐ Dual use research of concern |

## Methods

| n/a | Involved in the study |
| --- | --- |
| ☒ | ☐ ChIP-seq |
| ☒ | ☐ Flow cytometry |
| ☒ | ☐ MRI-based neuroimaging |

## Antibodies

| Antibodies used | 6x-His Tag Monoclonal Antibody (HIS.H8) (Invitrogen, catalog # MA1-21315), HA-tag (C29F4) rabbit mAb (Cell Signaling Technology #3724), anti-GyrA polyclonal (Inspiralis, PA001), (FLAG) DYKDDDDK Tag (D6W5B) Rabbit mAb (Cell Signaling Technology, #14793) |
| --- | --- |
| Validation | Antibodies have been validated based on manufacturer's website:<br>anti-His: validated by "western blot analysis of His-tagged protein in E. coli lysates".<br>anti-HA: validated by "western blot analysis of extracts from HeLa cells, untransfected or transfected with either HA-FoxO4 or HA-Akt3".<br>anti-GyrA polyclonal: reference Ey, P.L., S.J., & Jenkin, C.R. (1978).<br>anti-FLAG: validated by "western blot analysis of extracts from 293T cells, mock transfected (-) or transfected with DYKDDDDK-GFP".<br>In addition, all western blotting experiments include controls with untagged proteins as internal validation for antibodies. |

