## [Peer Review File · Nature]

Manuscript Title: Direct activation of a bacterial innate immune system by a viral capsid protein

Reviewer Comments & Author Rebuttals

Reviewer Reports on the Initial Version:

Referees' comments:

Referee #1 (Remarks to the Author):

In this study, Zhang and co-authors uncover how the CapRelSJ46 TA system that protects *E. coli* against diverse phages is being activated upon phage infection and leads to abortive infection.

The approach taken and the careful experimental design of this study make the results convincing. This is a very sound contribution to the TA field with key findings: the characterization of the molecular mechanism triggering activation of a TA system upon phage infection; and the discovery of a proteolysis-independent mechanism of activation for a type II TA system.

Authors show that CapRelSJ46 is a fused TA system containing a C-terminal antitoxin domain that senses phage infection through direct binding with the Gp57 capsid protein from SECΦ27 phage. By crystallization, they show that CapRelSJ46 protein alone adopts a closed conformation where the toxic pyrophosphokinase catalytic site (N-ter) is sterically hindered by the C-terminal antitoxin domain. Their mutation experiment further supports that model given that disrupting the intramolecular interface led to constitutive toxicity of CapRelSJ46 even in the absence of phage infection. To nail down the molecular determinant of phage dependent activation of CapRelSJ46 toxicity, authors used an experimental evolution approach and isolated SECΦ27 phages that can infect CapRelSJ46-carrying cells. Authors later demonstrate that SECΦ27 mutants are mutated in a hypothetical gene encoding a capsid protein which prevent activation of CapRelSJ46 and therefore, abortive infection. Combination of their *in vitro* and *in vivo* data suggests a model where the phage Gp57 capsid protein bind and relieves CapRelSJ46 autoinhibition leading to pyrophosphorylation of the cell's tRNAs by the pyrophosphokinase toxic domain and results in translation arrest leading to abortive infection.

I only have minor comments/questions:

- Because the Salmonella SJ46 phage contains a fused CapRel, is its infection abortive? How do you explain it had been selected?
- About the *in vitro* transcription-translation assay presented in figure 3j: if I understood correctly, it's a Gp57-encoding template that is being added to the *in vitro* reaction. I got confused by the text page 6 line 212 that says "in the presence of the SECΦ27 major capsid protein Gp57" maybe it should state explicitly that it's a Gp57-encoding template that is being added to the reaction. Related to this experiment, it seems that during the last 60min incubation some extra Gp57wt is being translated even in presence of CapRelSJ46, when no DHFR is being made. Could some *cis* elements make Gp57 translation more robust to translation inhibition (transiently)?
- In figure 3, panels h and i: SECΦ27 infection and gp57 overexpression lead to different extents of translation shutdown in cells harboring CapRelSJ46, could you comment on this result?

Referee #2 (Remarks to the Author):

Zhang et al. present an elegant series of biochemical, structural, genetic, and in vivo analyses that demonstrate a toxin-antitoxin system in bacteria named CapRel functions through direct recognition of bacteriophage capsid protein. The experiments are thorough, well presented, and the new results are exceptionally exciting for multiple fields related to understanding mechanisms of phage defense and the emerging connections between antiviral immunity in bacteria and animals. Some control experiments are necessary to complete understanding of the proposed mechanism of phage detection. Otherwise, I have only minor comments to help improve the manuscript for a general audience.

1) A key open question necessary to understand the proposed model of capsid sensing is to define the stoichiometry of CapRel-Gp57 complex formation. The authors' model proposes 1:1 complex formation based solely on structural modeling. This question is particularly important as the authors seek to compare CapRel with Trim5a activation in mammalian cells that relies on recognition of the intact HIV capsid lattice. Does CapRel sense individual Gp57 monomers or perhaps initial stages of capsid lattice formation? Experimental evidence to support stoichiometry (SEC-MALS, electron microscopy analysis etc.) should be readily available as the authors already demonstrate purification of CapRel and Gp57 and co-complex formation in vitro with ITC and HDX. Although determining the structure of the CapRel-Gp57 complex would significantly enhance the paper, only biochemical or lower-resolution stoichiometry information are necessary.

2) Is there is a fitness cost associated with evasion of CapRel sensing? These experiments are particularly interesting as previous results with the defense system Pycsar demonstrated that T5 phages that acquire capsid mutations to escape defense are significantly less fit than wildtype viruses (Tal and Morehouse et al. Cell 2021 PMID 34644530).

Minor Points

3) It is surprising that CapRel crystallized in an apparently "active" conformation. Can the authors further comment on why this may be? Does analysis of packing in the CapRel crystal lattice provide further insight into conformational changes required for toxin activation?

4) The CapRel chimera data demonstrate that the CapRel SJ46-Ebc chimera is significantly less capable of defending against phage T7 compared to the parental CapRel-Ebc construct (Figure 2b). The text description that the chimeric CapRel "gained protection against T7, manifesting as decreased EOP and smaller plaques" doesn't appear to match the data well. Can the authors comment on why sensing is severely compromised and potentially clarify the text?

5) In Figure 4a soluble Gp57 mutant proteins appear to express to significantly higher levels than wildtype protein. Is this difference meaningful, or perhaps related to capsid lattice stability and the mechanism of CapRel escape?

6) While no further experiments related to cell death are necessary, it would be helpful if the authors discussed how CapRel activation and translation repression leads to abortive infection. This step of the process is not outlined in the model Figure 5h.

7) In the main text figure panels, it is confusing which data are experimentally derived and which models are created with AlphaFold. Especially in Figure 4c presentation of the modeled CapRel-Gp57 complex is potentially misleading to the reader. Although the text legends are clear, it would be helpful in there was an indication in the panels themselves which structures are experimental and which are models.

8) Text Comments:

- Line 52: The phrase "which are effectively in constant surveillance mode" as used to selectively describe RM systems is confusing, aren't CapRel, CRISPR, and other pathogen-sensing machineries also always in a constant surveillance mode?

- Line 88: To understand use of the CapRel-SJ46 system, it may help to explain that defense

systems are often encoded within phage genomes as part of phage competition.

- Line 710: Please list the composition of crystal solution LMB C9 and the solutions used for cryoprotection.

I hope the authors will find my comments useful, thank you for the opportunity to read this exciting manuscript.

Philip Kranzusch

Referee #3 (Remarks to the Author):

Phage defense systems have received a lot of attention in recent years, but for most, their mechanisms of action remain mysterious. For defensive Abi systems (which encompass many mechanistically distinct systems, including TA systems), phage infection must be specifically sensed for activation of defense. However, known mechanisms underlying the activation of such defenses are few and far between, with many relying on incomplete data to draw definitive conclusions (e.g. escape phages used as the only evidence to identify said activating cue). In this manuscript, Zhang et al convincingly demonstrate the direct activation of a defensive TA system by a phage capsid protein *in vivo* and *in vitro*. This manuscript was an absolute pleasure to read – outstanding science described in a very clear manner. I am extremely enthusiastic about this being published in *Nature*, as it really represents a milestone in our understanding of bacterial immune systems. I have a series of fairly minor suggestions, though I'll label some as 'major' to distinguish from truly minor spelling-type issues, none are truly major in the sense of significantly decreasing my enthusiasm for this outstanding work.

Major:

The authors use strong language about the broad applicability of their findings: 'anticipating that major capsid proteins may emerge as common, direct triggers for a diverse range of anti-phage defense systems.' Yet they haven't shown that major capsid proteins are common triggers for CapRel homologs. I am not suggesting the authors repeat all the *in vivo/in vitro* studies with different pairs, but given that the identity of the capsid protein is known for several of the phages tested here, it would be worth testing for CFUs during co-expression of the inhibitory CapRel and cognate inhibited phage capsid protein (e.g. for T7 capsid co-expressed with CapRelEBC as was done in Fig 3g for Gp57+CapRelSJ46). One would expect to see toxicity upon co-expression only, as was observed for Gp57+CapRelSJ46. Some of the major capsid proteins of phages tested are also solved, and these could be used in some structural modeling with an inhibitory CapRel to look for evidence of an interaction. This would strengthen the story significantly. If, however, the authors find no evidence of capsid triggering other CapRelS, this is a valuable discussion point and not a deal-breaker for the paper.

The bulk of the paper focuses on Secphi27. As such, it would be much more relevant to include the one-step growth curve for this phage +/- CapRelSJ46 in fig 1e (T4 data should be moved to the supplement); time-course data for CapRel and translation inhibition of Secphi27 is presented in Fig 3a/i but we have no baseline for the kinetics of this phage's replication.

Line 124 – the authors construct a chimera in which the C-term of CapRelSJ46 was replaced by the corresponding region of CapRelEbc. They observe that the chimeric CapRel no longer protected against Secphi27 but gained (some) capacity to restrict T7. The description of this data is stronger than what is shown (Fig 2B), which is a very modest reduction in T7 plaquing. Given that the chimera does not phenocopy CapRelEbc, I would like to see the reciprocal chimera (N term swaps), which, according to their model/conclusions, should not alter the specificity. The chimeras also may not be stable/well expressed, which may explain the results – I suggest the authors blot for the WT/chimeras at a minimum.

Figure 3a - is there a loading control here? There appears to be a slight decrease at 60 minutes – again, it would be valuable to know if, at this point, there is already evidence that CapRel is inhibiting phage production (referring to comment regarding missing one-step growth curve).

In figure 3J&K, it appears the authors switched to testing the double mutant, unclear why the single mutants weren't tested. Some explanation/pointing this out is necessary.

In Figure 5d - F120L/124F don't ablate toxicity and yet allow for apparent complete restoration of EOP – an explanation is warranted, and also, it should be confirmed that these alleles are not suddenly toxic on their own as was done in previous experiments.

My comments regarding the kinetics point to a missing piece in the discussion that I hope can be accommodated, though, of course, I understand there are word limits. The assumption is that Gp57 is produced late – but as far as I can tell, this is not known (or evaluated here). Gp57 should be blotted for (for example, in parallel with the CapRel blot in Fig 3a). Some discussion of why a late expressed phage gene product would be a valuable trigger is necessary – especially for those not immersed in the field. Logically it may be because it is a 'last line of defense' which usually operates in a cell with other defenses. Or does this phage produce capsid early, and thus triggering early makes more intuitive sense to more robustly inhibit phage. For T7, capsid is expressed late, and yet CapRelEBC provides near-complete protection against this phage, this is not intuitive. Perhaps CapRelEBC uses a different trigger (see above). In keeping in mind that the CapRel system is found on resident prophages, perhaps the later induction provides ample time for the prophage to enter the lytic cycle and escape the cell under attack by a different incoming phage.

In the discussion (line 307) I would like to see some additional discussion around the abundance of fused TA systems as proteolysis of the antitoxin for activation in such a context is difficult to imagine. Do the authors anticipate that direct binding of the antitoxin is more pervasive /more realistic for fused TA systems specifically? It is speculative, of course, but one added sentence would be nice. It could be valuable to have added the relative abundance of fused systems vs others in Figure 1a to this point.

Lastly, most presented data are representative images of experiments performed with replicates – I would not accept graphs generated from single data points, so I just want to ensure replicates are available/organized/included in a repository and a link to that be included in the 'Data and Materials Availability Statement.' I do not doubt the rigor of the data, but this standard should be applied to all and should be required for submission.

Minor:

Fig 3B - explain white vs brown wells in legend. Last well to the right I assume is no phage, this is not clear from the cartoon of the dilution series.

Line 184 – A supplementary table of all mutations found should be provided. I am convinced by the follow-up data, even if others were observed, but those data should be made transparent.

Line 324 – perhaps a bit strong to say all domains of life unless known for Archaea

“Uptake” is misspelled on line 763 (as upatke).

Fig S2a legend 'indicating that pseudo-ZFD'. Should be "indicating the pseudo-ZFD"

Fig 3H & K – the choice of colors may not be the most color blind friendly, please check.

Referee #4 (Remarks to the Author):

The work by Zhang et al. reported that the C-terminus of a fused TA can bind to the major coat protein of phages and demonstrated that this binding could activate the toxicity of the N-terminus of the fused TA by relieving autoinhibition. The authors used several in vitro evidence to draw the main conclusion that the phage capsid proteins stimulate the innate immune system of bacteria. Although many new anti-phage elements have been reported over the last three years, how bacteria sense phage attack and initiate 'altruistic' cell death via activating toxins in the population remains largely unclear.

Although the finding that the MCP can bind to the antitoxin of the TA systems is intriguing and novel, the robustness of the binding and the outcome of the binding (conformational change of the fused TA) are not verified by in vivo assays, and the resolved structural data by X-ray crystal diffraction and the predicted structures by AlphaFold which they used for the proposed model is contradictory.

The "Direct activation", as the authors stated in the title, is not fully supported by the presented assays. At least two direct pieces of evidence are needed to support the finding that the C-terminus of CapRel serves as the real sensor for phage attack by binding directly to the phage MCP. Firstly, most of the presented assays are based on the overexpression of TA in plasmid. In order to show this CapRel functions as a sensor and player in anti-phage, it is crucial to have a high baseline expression of the fused TA. However, the author only showed that the CapRel protein level is not changed when phage SECΦ27 is infected by western blot analysis (Fig. 3a). What is the expression level of CapRel in the host bacteria with a chromosomally-encoded CapRel that has the anti-phage activity? Is there enough active state of CapRels to halt host translation when MCP binds to the C-terminus?

Secondly, in the model proposed here, the toxin activation relies on the conformational change of the fused CapRel. Regarding the activation mechanism, there is an apparent contradiction between the crystal structure of CapRelSJ46 solved in this paper (an open state without the presence of phage major capsid protein) and the mechanism proposed by the authors based on AlphaFold prediction: CapRelSJ46 should be a closed state without the presence of phage major capsid protein (on lines 301-306). If so, the major capsid protein is not the main trigger for the transition between the two conformations. Small-angle scattering or other experiments are needed to prove the conformation of CapRelSJ46 in solution with and without MCP.

There are many overreaching statements regarding the proven implications of the work. For example, another primary concern is the claim of the "innate immune system." As the authors illustrate in the beginning (on lines 88-92), the fused TA is encoded by phage or prophages. Why do the authors name them as the "innate immune system of bacteria"? Many genes in prophages of E. coli are kept at relatively deficient levels or silenced. If CapRels are encoded by phage or prophage, the results presented in this paper is phage-phage interaction instead of host-phage interaction.

Appropriate use of statistics and treatment of uncertainties:

The weaknesses are the low replications and qualitative nature of the results. For example, Fig 1d,

1f, 2b, 2g, 3c, 3g, 4f, 4g, and 5 lack solid statistical analysis.

Author Rebuttals to Initial Comments:

We thank the reviewers for their enthusiasm about our work as well as their constructive suggestions for improving it. Below we respond to each query, noting how the text and/or figures have been revised accordingly. Reviewer comments are in black; our responses follow in blue.

Referees' comments:

Referee #1 (Remarks to the Author):

In this study, Zhang and co-authors uncover how the CapRelSJ46 TA system that protects *E. coli* against diverse phages is being activated upon phage infection and leads to abortive infection.

The approach taken and the careful experimental design of this study make the results convincing. This is a very sound contribution to the TA field with key findings: the characterization of the molecular mechanism triggering activation of a TA system upon phage infection; and the discovery of a proteolysis-independent mechanism of activation for a type II TA system.

Authors show that CapRelSJ46 is a fused TA system containing a C-terminal antitoxin domain that senses phage infection through direct binding with the Gp57 capsid protein from SECΦ27 phage. By crystallization, they show that CapRelSJ46 protein alone adopts a closed conformation where the toxic pyrophosphokinase catalytic site (N-ter) is sterically hindered by the C-terminal antitoxin domain. Their mutation experiment further supports that model given that disrupting the intramolecular interface led to constitutive toxicity of CapRelSJ46 even in the absence of phage infection. To nail down the molecular determinant of phage dependent activation of CapRelSJ46 toxicity, authors used an experimental evolution approach and isolated SECΦ27 phages that can infect CapRelSJ46-carrying cells. Authors later demonstrate that SECΦ27 mutants are mutated in a hypothetical gene encoding a capsid protein which prevent activation of CapRelSJ46 and therefore, abortive infection. Combination of their in vitro and in vivo data suggests a model where the phage Gp57 capsid protein bind and relieves CapRelSJ46 autoinhibition leading to pyrophosphorylation of the cell's tRNAs by the pyrophosphokinase toxic domain and results in translation arrest leading to abortive infection.

I only have minor comments/questions:

- Because the *Salmonella* SJ46 phage contains a fused CapRel, is its infection abortive? How do you explain it had been selected?

CapRel^{SJ46} does appear to function through an abortive infection mechanism based on the finding in Fig. 1h showing that the system provides defense at a low MOI (0.001) but not a high MOI (10). The *Salmonella* SJ46 phage likely carries, and was selected to harbor, this CapRel system because it enables its host to ward off infection by other phages. Indeed, a growing body of recent work points to temperate phages, and other mobile genetic elements, as major reservoirs of anti-phage defense elements (see, for example, Dedrick et al *Nature Micro* 2017, Owen et al *Cell Host Microbe* 2021, Vassallo et al *Biorxiv* 2022, Rousset et al *Cell Host Microbe* 2022). Importantly, SJ46 itself has a capsid protein quite different from that of SECΦ27 and the other phages examined in our study, so it is very unlikely to intoxicate itself, or its host.

- About the in vitro transcription-translation assay presented in figure 3j: if I understood correctly, it's a Gp57-encoding template that is being added to the in vitro reaction. I got confused by the text page 6 line 212 that says "in the presence of the SECΦ27 major capsid protein Gp57" maybe it should state explicitly that it's a Gp57-encoding template that is being added to the reaction. Related to this experiment, it seems that during the last 60min incubation some extra Gp57wt is

being translated even in presence of CapRel^{SJ46}, when no DHFR is being made. Could some cis elements make Gp57 translation more robust to translation inhibition (transiently)?

We have modified the text as suggested (line 236-238 of the revised manuscript). Additionally, in response to a query from Reviewer 3, we have replaced the former Fig. 3j with a version in which we examine the individual amino acid substitutions identified in the escape mutant variants of Gp57. These single substitutions prevent the Gp57-mediated activation of tRNA pyrophosphorylation by CapRel^{SJ46} and the consequent abrogation of DHFR and Gp57 production.

• In figure 3, panels h and i: SECΦ27 infection and gp57 overexpression lead to different extents of translation shutdown in cells harboring CapRel^{SJ46}, could you comment on this result?

This is almost certainly because SECΦ27 does not adsorb well to *E. coli* K12 cells such that not all cells are likely infected, even at high MOIs, with variable timing of genome injection for those cells that are infected. In contrast, when overproducing Gp57 using arabinose induction, cells will generally all activate CapRel^{SJ46} in a relatively synchronous manner. We have now, in response to a related query on infection kinetics from Reviewer 3, added data to show that SECΦ27 does not adsorb well (new Extended Data Fig. 4k), leading to asynchronous infection dynamics (new Fig. 1f). We also comment specifically on the point raised by this Reviewer following the presentation of Fig. 3h and 3i (see lines 226-228 of the revised manuscript).

Referee #2 (Remarks to the Author):

Zhang et al. present an elegant series of biochemical, structural, genetic, and in vivo analyses that demonstrate a toxin-antitoxin system in bacteria named CapRel functions through direct recognition of bacteriophage capsid protein. The experiments are thorough, well presented, and the new results are exceptionally exciting for multiple fields related to understanding mechanisms of phage defense and the emerging connections between antiviral immunity in bacteria and animals. Some control experiments are necessary to complete understanding of the proposed mechanism of phage detection. Otherwise, I have only minor comments to help improve the manuscript for a general audience.

1) A key open question necessary to understand the proposed model of capsid sensing is to define the stoichiometry of CapRel–Gp57 complex formation. The authors' model proposes 1:1 complex formation based solely on structural modeling. This question is particularly important as the authors seek to compare CapRel with Trim5a activation in mammalian cells that relies on recognition of the intact HIV capsid lattice. Does CapRel sense individual Gp57 monomers or perhaps initial stages of capsid lattice formation? Experimental evidence to support stoichiometry (SEC-MALS, electron microscopy analysis etc.) should be readily available as the authors already demonstrate purification of CapRel and Gp57 and co-complex formation in vitro with ITC and HDX. Although determining the structure of the CapRel–Gp57 complex would significantly enhance the paper, only biochemical or lower-resolution stoichiometry information are necessary.

The ITC data previously shown for CapRel^{SJ46}-Gp57 binding had indicated a 1:1 stoichiometry with an experimentally determined n-value of 0.91; we had not initially indicated this in Fig. 4b, but now do so. In addition, we have performed SEC-MALS, as suggested by the Reviewer, which further supports the notion of a 1:1 complex, with an experimentally determined molecular weight of the complex being 74 kDa, compared to the theoretical molecular weight of 78 kDa (see new Fig. 4c). Thus, we favor a model in which Gp57 monomers or perhaps early-stage capsid lattice components are able to activate CapRel^{SJ46}. Consistent with that model, we found that CapRel^{SJ46} could be activated by Gp57 alone both *in vivo* and *in vitro*, in the absence of any additional phage proteins, e.g. minor capsid proteins, protease processing enzymes, or chaperones.

2) Is there is a fitness cost associated with evasion of CapRel sensing? These experiments are particularly interesting as previous results with the defense system Pycsar demonstrated that T5 phages that acquire capsid mutations to escape defense are significantly less fit than wildtype viruses (Tal and Morehouse et al. Cell 2021 PMID 34644530).

We have tested exactly this idea, but did not detect any substantial fitness cost, at least with respect to infection of *E. coli* MG1655. However, the escape mutants isolated could, in principle, be less effective at infecting other hosts, or they could trigger other CapRels or other phage defense systems. This is something we intend to explore in the future.

Minor Points

3) It is surprising that CapRel crystallized in an apparently “active” conformation. Can the authors further comment on why this may be? Does analysis of packing in the CapRel crystal lattice provide further insight into conformational changes required for toxin activation?

Indeed, analysis of the crystal lattice suggests that the open state is favored by crystal packing constraints. As shown in the new Extended Data Fig. 3g, the closed state would produce substantial clash with symmetry-related partners. We now mention this on lines 154-156 of the revised manuscript. We comment further on the support for our model of an open, active state and a closed, inactive state in response to a related query from Reviewer 4 below.

4) The CapRel chimera data demonstrate that the CapRel SJ46–Ebc chimera is significantly less capable of defending against phage T7 compared to the parental CapRel–Ebc construct (Figure 2b). The text description that the chimeric CapRel “gained protection against T7, manifesting as decreased EOP and smaller plaques” doesn’t appear to match the data well. Can the authors comment on why sensing is severely compromised and potentially clarify the text?

We have now modified the text (lines 136-139) to emphasize that the chimera leads to increased protection against T7, but not to the levels seen with wild-type CapRel^{Ebc}. We also note that this likely arises because the intramolecular interface formed in the chimera is not optimized, particularly compared to the two different wild-type CapRels.

5) In Figure 4a soluble Gp57 mutant proteins appear to express to significantly higher levels than wildtype protein. Is this difference meaningful, or perhaps related to capsid lattice stability and the mechanism of CapRel escape?

We think this likely occurs because CapRel^{SJ46} activation by wild-type Gp57 inhibits translation, whereas the mutant proteins do not. Thus, the mutants can accumulate to higher levels, as seen in Fig. 4a.

6) While no further experiments related to cell death are necessary, it would be helpful if the authors discussed how CapRel activation and translation repression leads to abortive infection. This step of the process is not outlined in the model Figure 5h.

We now note in the text (see lines 348-350) that the translation inhibition driven by CapRel^{SJ46} must be enough to block mature virion production, but precisely what step, or steps, of virion production or release that becomes limiting is not yet clear, but again something we will pursue in future studies.

7) In the main text figure panels, it is confusing which data are experimentally derived and which models are created with AlphaFold. Especially in Figure 4c presentation of the modeled CapRel–

Gp57 complex is potentially misleading to the reader. Although the text legends are clear, it would be helpful if there was an indication in the panels themselves which structures are experimental and which are models.

We now explicitly note this in the revised versions of Fig. 2c-d and Fig. 4d (formerly 4c).

8) Text Comments:

- Line 52: The phrase “which are effectively in constant surveillance mode” as used to selectively describe RM systems is confusing, aren't CapRel, CRISPR, and other pathogen-sensing machineries also always in a constant surveillance mode?

We have now clarified this line of the text to indicate the RM and CRISPR systems do not require specific triggers and instead rely on the ability to distinguish self from non-self with no intrinsic change in activity of the effector (in each case a nuclease), whereas CapRel and other anti-phage defense systems must recognize specific features of an infecting phage to trigger activation of their effector proteins.

- Line 88: To understand use of the CapRel-SJ46 system, it may help to explain that defense systems are often encoded within phage genomes as part of phage competition.

We have now added just such a comment to the manuscript (lines 93-94).

- Line 710: Please list the composition of crystal solution LMB C9 and the solutions used for cryoprotection.

The composition of these solutions has been added to the Methods section.

I hope the authors will find my comments useful, thank you for the opportunity to read this exciting manuscript.

Philip Kranzusch

Indeed - many thanks for the useful and constructive comments that we feel have improved the manuscript.

Referee #3 (Remarks to the Author):

Phage defense systems have received a lot of attention in recent years, but for most, their mechanisms of action remain mysterious. For defensive Abi systems (which encompass many mechanistically distinct systems, including TA systems), phage infection must be specifically sensed for activation of defense. However, known mechanisms underlying the activation of such defenses are few and far between, with many relying on incomplete data to draw definitive conclusions (e.g. escape phages used as the only evidence to identify said activating cue). In this manuscript, Zhang et al convincingly demonstrate the direct activation of a defensive TA system by a phage capsid protein in vivo and in vitro. This manuscript was an absolute pleasure to read – outstanding science described in a very clear manner. I am extremely enthusiastic about this being published in Nature, as it really represents a milestone in our understanding of bacterial immune systems. I have a series of fairly minor suggestions, though I'll label some as 'major' to distinguish from truly minor spelling-type issues, none are truly major in the sense of significantly decreasing my enthusiasm for this outstanding work.

Major:

The authors use strong language about the broad applicability of their findings: ‘anticipating that major capsid proteins may emerge as common, direct triggers for a diverse range of anti-phage defense systems.’ Yet they haven’t shown that major capsid proteins are common triggers for CapRel homologs. I am not suggesting the authors repeat all the in vivo/in vitro studies with different pairs, but given that the identity of the capsid protein is known for several of the phages tested here, it would be worth testing for CFUs during co-expression of the inhibitory CapRel and cognate inhibited phage capsid protein (e.g. for T7 capsid co-expressed with CapRelEBC as was done in Fig 3g for Gp57+CapRelSJ46). One would expect to see toxicity upon co-expression only, as was observed for Gp57+CapRelSJ46. Some of the major capsid proteins of phages tested are also solved, and these could be used in some structural modeling with an inhibitory CapRel to look for evidence of an interaction. This would strengthen the story significantly. If, however, the authors find no evidence of capsid triggering other CapRelS, this is a valuable discussion point and not a deal-breaker for the paper.

We have now screened additional CapRel homologs for their anti-phage defense capabilities, finding that all provided protection against at least one of the phages tested. We then selected a CapRel from a different strain of *E. coli* (HT2012018) than that harboring CapRel^{SJ46}. This CapRel^{E_cHT} defends against Bas8 phage, albeit with more modest protection than CapRel^{SJ46}. Nevertheless, we selected for escape mutants of Bas8 and found mutations in its major capsid protein that produce substitutions G111S, L116F, and I124N, each of which is in the same corresponding region of Gp57 from SECΦ27. These data support the notion that major capsid proteins may be common, direct triggers for CapRel systems and we have added these new data to the revised manuscript (Extended Data Fig. 6g), with accompanying text on lines 325-329. We also tested, as suggested, whether the T7 capsid would activate the CapRel^{E_bc} system, but it did not. **[Redacted]** In a related vein, we also now include a reference to a paper on *bioRxiv* indicating that mutants in a *Pseudomonas aeruginosa* phage that enable it to evade defense by a CBASS system map to its major capsid gene. Whether this capsid directly interacts with the CBASS system or activates it is not known, but this example, along with the others already cited (Gol-Lit, PifA, Pyscar), point to a likely major role for capsid proteins as the triggers of diverse phage defense elements.

The bulk of the paper focuses on Secphi27. As such, it would be much more relevant to include the one-step growth curve for this phage +/- CapRelSJ46 in fig 1e (T4 data should be moved to the supplement); time-course data for CapRel and translation inhibition of Secphi27 is presented in Fig 3a/i but we have no baseline for the kinetics of this phage’s replication.

We now present a one-step growth curve with SECΦ27 in Fig. 1f. However, as noted above in a response to Reviewer 1, this phage adsorbs quite poorly to *E. coli* K12 strains, as now documented in Extended Data Fig. 4k. Consequently, it is difficult to achieve complete and synchronous infection with SECΦ27, which results in the more gradual accumulation of PFUs following infection, compared to T4 where bursting occurs in a narrow time range. Nevertheless, the one-step growth curve for SECΦ27 indicates an approximate eclipse period of ~55-60 min, which we think will help readers interpret the translation inhibition data in Fig. 3, as noted by the Reviewer.

Line 124 – the authors construct a chimera in which the C-term of CapRelSJ46 was replaced by the corresponding region of CapRelEbc. They observe that the chimeric CapRel no longer protected against Secphi27 but gained (some) capacity to restrict T7. The description of this data is stronger than what is shown (Fig 2B), which is a very modest reduction in T7 plaquing. Given that the chimera does not phenocopy CapRelEbc, I would like to see the reciprocal chimera (N term swaps), which, according to their model/conclusions, should not alter the specificity. The chimeras also may not be stable/well expressed, which may explain the results – I suggest the authors blot for the WT/chimeras at a minimum.

As noted above in response to a related point from Reviewer 2, we have revised the text describing the results in Fig. 2b (lines 136-139 of the revised manuscript). Additionally, we tried to construct the reciprocal chimera, as suggested, but found that it was toxic, likely indicating that the intramolecular interface is not optimal and leads to constitutive activity. As also suggested by this Reviewer, we blotted for the chimera in Fig. 2b and found that it is not expressed any less than the wild-type CapRel^{SJ46} and CapRel^{Ebc} supporting the notion that, while the phage specificity has been altered, the intramolecular interface is not optimized for activation. The new blotting data are shown in the revised Extended Data Fig. 3c.

Figure 3a - is there a loading control here? There appears to be a slight decrease at 60 minutes – again, it would be valuable to know if, at this point, there is already evidence that CapRel is inhibiting phage production (referring to comment regarding missing one-step growth curve).

We now include a stain for total protein levels for the blot shown in Fig. 3a (see revised Extended Data Fig. 4b), which indicates comparable loading at all time points. We also, as noted above, added the one-step growth curve for SECΦ27, indicating an eclipse period of ~55-60 min, which coincides with the slight decrease noted by the Reviewer and is consistent with our measurement of translation inhibition in Fig. 3i showing translation is inhibited in CapRel^{SJ46} at 45 min.

In figure 3J&K, it appears the authors switched to testing the double mutant, unclear why the single mutants weren't tested. Some explanation/pointing this out is necessary.

We have now revised these figure panels to show data for the single mutants, which behave indistinguishably from the double mutant.

In Figure 5d - F120L/124F don't ablate toxicity and yet allow for apparent complete restoration of EOP – an explanation is warranted, and also, it should be confirmed that these alleles are not suddenly toxic on their own as was done in previous experiments.

This is a very astute observation and great suggestion. We have now overexpressed each in the absence of CapRel^{SJ46} and verified that neither has a detectable effect on growth (Extended Data Fig. 6f). So the simplest and most likely explanation is that these variants reduce binding to CapRel^{SJ46}, but can still stimulate CapRel^{SJ46} (as shown in Fig. 5d) because they accumulate over the extended periods of time involved in these plating experiments. In contrast, in the context of phage defense (Fig. 5c), there is a very limited window of time in which CapRel^{SJ46} must sense capsid protein and then act to abort infection. If the binding is reduced enough, CapRel^{SJ46} won't get activated at all or enough to block translation before the infection process is over and new, mature virions are released. We now provide this explanation in brief on lines 318-322 of the revised manuscript.

My comments regarding the kinetics point to a missing piece in the discussion that I hope can be accommodated, though, of course, I understand there are word limits. The assumption is that Gp57 is produced late – but as far as I can tell, this is not known (or evaluated here). Gp57 should

be blotted for (for example, in parallel with the CapRel blot in Fig 3a). Some discussion of why a late expressed phage gene product would be a valuable trigger is necessary – especially for those not immersed in the field. Logically it may be because it is a ‘last line of defense’ which usually operates in a cell with other defenses. Or does this phage produce capsid early, and thus triggering early makes more intuitive sense to more robustly inhibit phage. For T7, capsid is expressed late, and yet CapRelEBC provides near-complete protection against this phage, this is not intuitive. Perhaps CapRelEBC uses a different trigger (see above). In keeping in mind that the CapRel system is found on resident prophages, perhaps the later induction provides ample time for the prophage to enter the lytic cycle and escape the cell under attack by a different incoming phage.

This is also a great point that we have now addressed. We are unable to directly modify the SECΦ27 genome, so instead we put *gp57* under the control of its native promoter into the *E. coli* genome and then blotted for Gp57 levels with or without phage infection (see new Extended Data Fig. 4I). From this experiment, we see that Gp57 levels increased above the baseline during phage infection after ~30 minutes (note that Gp57 also gets processed to a shorter form during infection, as with many major capsid proteins). Translation inhibition by CapRel^{SJ46} was detected by ~45 min and, as noted above, the eclipse period for SECΦ27 is difficult to measure precisely, but at least 55-60 min. [Redacted]

In the discussion (line 307) I would like to see some additional discussion around the abundance of fused TA systems as proteolysis of the antitoxin for activation in such a context is difficult to imagine. Do the authors anticipate that direct binding of the antitoxin is more pervasive /more realistic for fused TA systems specifically? It is speculative, of course, but one added sentence would be nice. It could be valuable to have added the relative abundance of fused systems vs others in Figure 1a to this point.

We now note in the text (lines 85-91), when introducing Fig. 1a, that CapRel systems are frequently found as fused systems and we now include Extended Data Fig. 1 to illustrate the broad distribution of fused CapRel systems. In short, CapRel, which is the the largest toxSAS subfamily, is widespread in Proteobacteria, Actinobacteria and Cyanobacteria along with isolated examples in other phyla. With the exception of Actinobacterial CapRels, the prevalent form is fused. We have also now added two sentences to the discussion (lines 352-357) about the potential for direct binding of triggering proteins like Gp57 as a potential, and likely, mechanism of action for fused and non-fused TA systems.

Lastly, most presented data are representative images of experiments performed with replicates – I would not accept graphs generated from single data points, so I just want to ensure replicates are available/organized/included in a repository and a link to that be included in the ‘Data and Materials Availability Statement.’ I do not doubt the rigor of the data, but this standard should be applied to all and should be required for submission.

For all experiments in which representative images are shown (*i.e.* plating and plaquing data), we now show the quantified data in bar graph form in the Extended Data figures (with data points for each independent replicate) and also provided as Source Data.

Minor:

Fig 3B - explain white vs brown wells in legend. Last well to the right I assume is no phage, this is not clear from the cartoon of the dilution series.

We have updated the legend to indicate that white wells represent those cleared by phage.

Line 184 – A supplementary table of all mutations found should be provided. I am convinced by the follow-up data, even if others were observed, but those data should be made transparent.

This table has been generated and is now provided as Supplementary Table 2.

Line 324 – perhaps a bit strong to say all domains of life unless known for Archaea

A fair point - we have modified this line.

“Uptake” is misspelled on line 763 (as upatke).

Typo fixed.

Fig S2a legend ‘indicating that pseudo-ZFD’. Should be “indicating the pseudo-ZFD’

Fixed (now Extended Data Fig. 3a).

Fig 3H & K – the choice of colors may not be the most color blind friendly, please check.

The green has been changed to grey and brown respectively.

Referee #4 (Remarks to the Author):

The work by Zhang et al. reported that the C-terminus of a fused TA can bind to the major coat protein of phages and demonstrated that this binding could activate the toxicity of the N-terminus of the fused TA by relieving autoinhibition. The authors used several *in vitro* evidence to draw the main conclusion that the phage capsid proteins stimulate the innate immune system of bacteria. Although many new anti-phage elements have been reported over the last three years, how bacteria sense phage attack and initiate ‘altruistic’ cell death via activating toxins in the population remains largely unclear.

Although the finding that the MCP can bind to the antitoxin of the TA systems is intriguing and novel, the robustness of the binding and the outcome of the binding (conformational change of the fused TA) are not verified by *in vivo* assays, and the resolved structural data by X-ray crystal diffraction and the predicted structures by AlphaFold which they used for the proposed model is contradictory. The “Direct activation”, as the authors stated in the title, is not fully supported by the presented assays. At least two direct pieces of evidence are needed to support the finding that the C-terminus of CapRel serves as the real sensor for phage attack by binding directly to the phage MCP. Firstly, most of the presented assays are based on the overexpression of TA in plasmid. In order to show this CapRel functions as a sensor and player in anti-phage, it is crucial to have a high baseline expression of the fused TA. However, the author only showed that the CapRel protein level is not changed when phage SECΦ27 is infected by western blot analysis (Fig. 3a). What is the expression level of CapRel in the host bacteria with a chromosomally-encoded CapRel that has the anti-phage activity? Is there enough active state of CapRel to halt host translation when MCP binds to the C-terminus?

We think the conclusion of direct activation is warranted based on at least eight mutually corroborating lines of evidence, encompassing both our *in vivo* and *in vitro* studies: (i) escape

mutants of SECΦ27 specifically and only map to *gp57*, which encodes the major capsid protein, (ii) purified capsid protein binds directly to CapRel^{SJ46} (measured by ITC and MALS) and triggers its pyrophosphorylation activity *in vitro*, (iii) Gp57 can stimulate CapRel^{SJ46} to inhibit translation in an *in vitro* transcription-translation assay, but the escape variants of Gp57 cannot stimulate CapRel^{SJ46} and have substantially lower affinity for CapRel^{SJ46} than wild-type Gp57, (iv) purified capsid protein leads to substantial protection (as measured by hydrogen-deuterium exchange) of the pseudo-ZFD of CapRel^{SJ46}, (v) the escape mutations in Gp57 and the Gp57-insensitive mutations in CapRel^{SJ46} map to the interaction interface predicted by the independent HDX-MS studies, (vi) point mutations in the pseudo-ZFD domain of CapRel^{SJ46} were isolated that prevent activation by the capsid protein *in vivo*, (vii) expressing the major capsid protein in the absence of phage infection is sufficient to activate CapRel^{SJ46} *in vivo* to inhibit translation, (viii) mutations informed by the AlphaFold-predicted 'closed' state of CapRel^{SJ46} lead to constitutive activity.

We have, however, also now performed the experiments suggested by the Reviewer. We introduced CapRel^{SJ46} onto the *E. coli* chromosome under the control of its native promoter and then compared levels by Western blot analysis (see new Extended Data Fig. 4c). We find that the levels are, as expected, reduced relative to the plasmid-borne CapRel^{SJ46}. However, the chromosomally-encoded CapRel^{SJ46} still provides robust protection against SECΦ27 (see new Extended Data Fig. 2f), and can be activated to cause the same (when compared to plasmid-borne CapRel^{SJ46}) level of toxicity when Gp57 is co-produced (Extended Data Fig. 4e), indicating that there is enough to halt translation.

We appreciate the concern about the levels of active toxin required to promote abortive infection. However, the combined effect of the high affinity between CapRel^{SJ46} and Gp57 and the catalytic effect of the active toxin ensures that even if CapRel^{SJ46} is expressed at low levels the tight-binding complex will be formed and remain stable, enabling it to catalytically modify the cellular pools of uncharged tRNA (which then become dead-end products) and arrest of translation.

Secondly, in the model proposed here, the toxin activation relies on the conformational change of the fused CapRel. Regarding the activation mechanism, there is an apparent contradiction between the crystal structure of CapRelSJ46 solved in this paper (an open state without the presence of phage major capsid protein) and the mechanism proposed by the authors based on AlphaFold prediction: CapRelSJ46 should be a closed state without the presence of phage major capsid protein (on lines 301-306). If so, the major capsid protein is not the main trigger for the transition between the two conformations. Small-angle scattering or other experiments are needed to prove the conformation of CapRelSJ46 in solution with and without MCP.

First, we would highlight our response to a related point from Reviewer 2 above where we indicate that the particular crystallization conditions used promote a packing in the lattice that favors the open state of CapRel^{SJ46}, but does not rule out that the enzyme could be crystallized in the closed state under alternative conditions. Because of the very high protein concentration of a crystal lattice, this type of "packing-favored" structure is a common feature of proteins that explore multiple conformations; the inward open vs outward conformations of multidrug transporters are just but one example. For this family of transporters, many members have been crystallized in only one of the two states even in the presence of mutations or inducers that would trigger the other, with the capricious crystal lattice conditions as the main driver of the conformational selection. This is also why using SAXS to study CapRel^{SJ46} could be quite cumbersome. The high protein concentrations required to do SAXS could actually mask the actual state of the enzyme in a more physiological environment. Consequently, we opted for hydrogen-deuterium exchange to study the system. Those results strongly support the notion of a Gp57-induced conformational change from the closed to open state: the data indicate both that Gp57 leads to significant protection of the C-terminal pseudo-ZFD and that residues near the active site become less protected, which is

consistent with the opening of the enzyme upon activation by Gp57 (see Fig. 4e-f and Extended Data Fig. 5f-g). Further, as noted in our first response above to this Reviewer, there are many independent, mutually corroborating lines of investigation already presented in the paper that support Gp57 as the main trigger for the conformational change.

There are many overreaching statements regarding the proven implications of the work. For example, another primary concern is the claim of the “innate immune system.” As the authors illustrate in the beginning (on lines 88-92), the fused TA is encoded by phage or prophages. Why do the authors name them as the “innate immune system of bacteria”? Many genes in prophages of *E. coli* are kept at relatively deficient levels or silenced. If CapRels are encoded by phage or prophage, the results presented in this paper is phage-phage interaction instead of host-phage interaction.

Prophages are, of course, a major component of many bacterial genomes and contribute in important ways to their hosts, including their immunity against phages, blurring the line between phage-phage and host-phage interactions. Also, we would note that although many genes in the prophages of *E. coli* and other bacteria are silenced, as noted by the Reviewer, there is a growing body of evidence that some genes (beyond those encoding for phage repressors) are, in fact, expressed in the lysogenic state. For instance, the *rexAB* system of phage lambda is expressed during lysogeny and, notably, can help defend against infection by T4 *rII* mutants. Similarly, Dedrick et al *Nature Micro* 2017 showed in mycobacteria that lysogenized mycophages often express genes other than the repressor, and that many of these genes function in anti-phage defense.

Appropriate use of statistics and treatment of uncertainties:

The weaknesses are the low replications and qualitative nature of the results. For example, Fig 1d, 1f, 2b, 2g, 3c, 3g, 4f, 4g, and 5 lack solid statistical analysis.

We now show individual replicate data points for all of the panels in bar graphs in the Extended Data figures, with the quantitative data also provided in Source Data. For all relevant comparisons, we have performed and report statistical tests and p-values, as defined in the figure legends.

Reviewer Reports on the First Revision:

Referees' comments:

Referee #1 (Remarks to the Author):

Very nice revision of a beautiful work. In my opinion this work is now ready for Nature.

Referee #2 (Remarks to the Author):

The authors' revised manuscript is significantly improved. All of my reviewer points have been addressed with new experimental data and analysis of the CapRel crystal structure, and I congratulate the authors on a very exciting story. I have two minor points for the text, but otherwise recommend the manuscript for publication.

1) The authors present new analysis of the CapRel crystal lattice to support that a proposed inactive conformation is incompatible with crystal packing. While the authors' model is reasonable, a minor concern is that relatively little evidence is available to define what is actually the "inactive" CapRel conformation and what is the "active" CapRel conformation. The authors may want to be more cautious in the text to indicate that the current captured structural state by crystallography is consistent with a conformational change leading to activation but that determination of what is the fully active state will require future structural understanding.

2) The authors comment that capsid recognition may be common across different defense systems. Recent, elegant work by Gao, Zhang and colleagues demonstrates that phage portal and terminase proteins are conserved PAMPs recognized by STAND/Avs defense systems (Gao, Wilkinson, and Strecker et al. Science 2022 PMID 35951700). The authors should consider citing this work and perhaps amend their discussion to be less specifically focused on phage capsid recognition. The two best characterized systems CapRel and STAND/Avs together argue that structural proteins in general are likely conserved PAMPs for defense activation.

Referee #3 (Remarks to the Author):

I thank the authors for the careful consideration of my comments. I am however not satisfied with the lack of changes to the manuscript in response to my first major comment (regarding the narrative that capsid proteins are common triggers without even acknowledging that even among CapRel homologs there may be a diversity of triggers). I generally agree - capsid seems like a great trigger to evolve a response too (assuming the kinetics do not pose a challenge being 'too late' to establish a protective response). However, I think the authors need to add an explicit caveat to the generalizability of their manuscript given that they do not provide evidence that capsid is a common trigger among CapRel homologs.

[Redacted]

Another small comment: the brief explanation added in now lines 318-322 to explain the difference in toxicity/restoring EOP does not reflect the much better explanation in the rebuttal. I recognize the need to be brief here, but the idea of binding affinity should be included in the main text (not just the accumulation of protein).

Lastly, for extended data figure 1 please include a legend defining the corresponding color in the figure rather than relying on a description of colors written in the legend.

Congratulations to the authors again on an exciting manuscript.

Referee #4 (Remarks to the Author):

The authors have fully addressed my previous concerns. Specifically, they provided explanations for the crystallized structural of the fused TA being the active state even without MCP, and the difficulties of using SAXS to solve these structures. The chromosomally-encoded CapRel performed similarly in *E. coli* MG1655 as the plasmid-borne CapRel, combined with other evidences, it is reasonable to draw the conclusion that this process can occur in vivo during phage attack.

Author Rebuttals to First Revision:

We thank the reviewers for their enthusiasm about our work as well as their suggestions for improving it. Below we respond to each query, noting how the text and/or figures have been revised accordingly. Reviewer comments are in black; our responses follow in blue.

Referees' comments:

Referee #1 (Remarks to the Author):

Very nice revision of a beautiful work. In my opinion this work is now ready for Nature.

Referee #2 (Remarks to the Author):

The authors' revised manuscript is significantly improved. All of my reviewer points have been addressed with new experimental data and analysis of the CapRel crystal structure, and I congratulate the authors on a very exciting story. I have two minor points for the text, but otherwise recommend the manuscript for publication.

1) The authors present new analysis of the CapRel crystal lattice to support that a proposed inactive conformation is incompatible with crystal packing. While the authors' model is reasonable, a minor concern is that relatively little evidence is available to define what is actually the "inactive" CapRel conformation and what is the "active" CapRel conformation. The authors may want to be more cautious in the text to indicate that the current captured structural state by crystallography is consistent with a conformational change leading to activation but that determination of what is the fully active state will require future structural understanding.

We have now modified the text (line 263) to indicate that future structural studies are needed to validate the open state of CapRel^{SJ46} captured by crystallography and predicted complex with Gp57.

2 The authors comment that capsid recognition may be common across different defense systems. Recent, elegant work by Gao, Zhang and colleagues demonstrates that phage portal and terminase proteins are conserved PAMPs recognized by STAND/Avs defense systems (Gao, Wilkinson, and Strecker et al. Science 2022 PMID 35951700). The authors should consider citing this work and perhaps amend their discussion to be less specifically focused on phage capsid recognition. The two best characterized systems CapRel and STAND/Avs together argue that structural proteins in general are likely conserved PAMPs for defense activation.)

We now discuss (lines 371-373) that other structural components of phages can also serve as triggers and cite the STAND/Avs paper as suggested.

Referee #3 (Remarks to the Author):

I thank the authors for the careful consideration of my comments. I am however not satisfied with the lack of changes to the manuscript in response to my first major comment (regarding the narrative that capsid proteins are common triggers without even acknowledging that even among CapRel homologs there may be a diversity of triggers). I generally agree - capsid seems

like a great trigger to evolve a response too (assuming the kinetics do not pose a challenge being 'too late' to establish a protective response). However, I think the authors need to add an explicit caveat to the generalizability of their manuscript given that they do not provide evidence that capsid is a common trigger among CapRel homologs. **[Redacted]**

We now discuss (lines 374-375) the possibility that although capsid proteins and other structural proteins can be effective triggers, the intense arms race between bacteria and phage may drive bacterial defense systems to also rely on alternative triggers in some cases.

Another small comment: the brief explanation added in now lines 318-322 to explain the difference in toxicity/restoring EOP does not reflect the much better explanation in the rebuttal. I recognize the need to be brief here, but the idea of binding affinity should be included in the main text (not just the accumulation of protein).

We have now added discussion of capsid protein mutants having reduced binding affinity for CapRel^{SJ46} in line 322-323.

Lastly, for extended data figure 1 please include a legend defining the corresponding color in the figure rather than relying on a description of colors written in the legend. Congratulations to the authors again on an exciting manuscript.

A legend defining the corresponding colors has been added to Extended Data Fig. 1.

Referee #4 (Remarks to the Author):

The authors have fully addressed my previous concerns. Specifically, they provided explanations for the crystallized structural of the fused TA being the active state even without MCP, and the difficulties of using SAXS to solve these structures. The chromosomally-encoded CapRel performed similarly in *E. coli* MG1655 as the plasmid-borne CapRel, combined with other evidences, it is reasonable to draw the conclusion that this process can occur in vivo during phage attack.